

# Development of CarbonTracker Europe-CH₄ – Part 2: global methane emission estimates and their evaluation for 2000-2012.

Aki Tsuruta[1], Tuula Aalto[1], Leif Backman[1], Janne Hakkarainen[2], Ingrid T. van der Laan-Luijkx[3], Maarten C. Krol[3,4,5], Renato Spahni[6], Sander Houweling[4,5], Marko Laine[2], Ed Dlugokencky[7], Angel J. Gomez-Pelaez[8], Marcel van der Schoot[9], Ray Langenfelds[9], Raymond Ellul[10], Jgor Arduini[11,12], Francesco Apadula[13], Christoph Gerbig[14], Dietrich G. Feist[14], Rigel Kivi[15], Yukio Yoshida[16], Wouter Peters[3,17]

[1]Climate Research, Finnish Meteorological Institute, Helsinki, Finland
[2]Earth Observation, Finnish Meteorological Institute, Helsinki, Finland
[3]Meteorology and Air Quality, Wageningen University, Wageningen, the Netherlands
[4]SRON Netherlands Institute for Space Research, Utrecht, the Netherlands
[5]Institute for Marine and Atmospheric Research, Utrecht University, Utrecht, the Netherlands
[6]Climate and Environmental Physics, Physics Institute, and Oeschger Centre for Climate Change Research, University of Bern, Bern, Switzerland
[7]NOAA Earth System Research Laboratory, Boulder, Colorado, USA
[8]Izaña Atmospheric Research Center, Agencia Estatal de Meteorología (AEMET), Tenerife, Spain
[9]CSIRO Oceans and Atmosphere, Aspendale, Australia
[10]Atmospheric Research, Department of Geosciences, University of Malta, Msida, Malta
[11]Departement of Pure and Applied Sciences, University of Urbino, Urbino, Italy
[12]National Research Council, Institute of Atmospheric Sciences and Climate, Bologna, Italy
[13]Ricerca sul Sistema Energetico – RSE SpA, Milano, Italy
[14]Max Planck Institute for Biogeochemistry, Jena, Germany
[15]Arctic Research, Finnish Meteorological Institute, Sodankylä, Finland
[16]Center for Global Environmental Research, National Institute for Environmental Studies, Tsukuba, Ibaraki, Japan
[17]University of Groningen, Centre for Isotope Research, Groningen, the Netherlands

*Correspondence to*: Aki Tsuruta (Aki.Tsuruta@fmi.fi)

**Abstract.** Gobal methane emissions were estimated for 2000-2012 using the CarbonTracker Europe-CH₄ (CTE-CH₄) data assimilation system. In CTE-CH₄, the anthropogenic and biosphere emissions of CH₄ are simultaneously constrained by global atmospheric in-situ methane mole fraction observations. We use three configurations developed in Tsuruta *et al.* (2016) to assess the sensitivity of the CH₄ flux estimates to (a) the number of unknown flux scaling factors to be optimized which in turn depends on the choice of underlying land-ecosystem map, and (b) on the parametrization of vertical mixing in the atmospheric transport model TM5. The posterior emission estimates were evaluated by comparing simulations to surface in-situ observation sites, to profile observations made by aircraft, to dry air total column-averaged mole fractions (XCH₄) observations from the Total Carbon Column Observing Network (TCCON), and to XCH₄ retrievals from the Greenhouse gases Observing SATellite (GOSAT). Our estimated posterior mean global total emissions during 2000-2012 are 516±51 Tg CH₄ yr⁻¹, and emission estimates during 2007-2012 are 18 Tg CH₄ yr⁻¹ greater than those from 2001-2006, mainly driven by an increase in emissions from the south America temperate region, the Asia temperate region and Asia tropics. The



sensitivity of the flux estimates to the underlying ecosystem map was large for the Asia temperate region and Australia, but not significant in the northern latitude regions, i.e. the north American boreal region, the north American temperate region and Europe. Instead, the posterior estimates for the northern latitude regions show larger sensitivity to the choice of convection scheme in TM5. The Gregory *et al.* (2000) mixing scheme with faster interhemispheric exchange leads to higher estimated $CH_4$ emissions at northern latitudes, and lower emissions in southern latitudes, compared to the estimates using Tiedtke (1989) convection scheme. Our evaluation with non-assimilated observations showed that posterior mole fractions were better matched with the observations when Gregory *et al.* (2000) convection scheme was used.

# 1 Introduction

Methane ($CH_4$) is a greenhouse gas, whose a 100-year time horizon Global Warming Potential (GWP) is 28 times that of carbon dioxide ($CO_2$) (IPCC 2013: chapter 8.7). Methane is emitted by both anthropogenic activities and natural biogenic processes. The main anthropogenic emission sources are fugitive emission from solid fuel, leaks from gas extraction and distribution, agriculture, and waste management. Anthropogenic methane emissions are assumed to account for more than half of the total methane emissions from land and ocean (Kirschke *et al.*, 2013). Anthropogenic methane emissions have increased significantly since pre-industrial times largely due to heavy use of fossil fuels, but also due to the increase in ruminants, landfills and rice fields corresponding to the increase in human population (Ghosh *et al.*, 2015), which resulted in a steep increase in atmospheric methane concentration. Previous studies suggest that anthropogenic methane emissions did not increase significantly, or even decreased, during the 1980s and 1990s (Bousquet *et al.*, 2006; Dlugokencky *et al.*, 1998), which may have been the cause of stabilization of the atmospheric methane burden from 1999-2006. Changes in methane emissions during more recent years have not been satisfactorily explained. After some years of 'pause', atmospheric methane started to increase again in 2007 (Rigby *et al.*, 2008; Dlugokencky *et al.*, 2009). The growth rate of global average atmospheric methane from 2007 to 2012 averaged 5.7 ppb per year, which represents a significant change to the global $CH_4$ budget; assuming no trend in $CH_4$ lifetime, atmospheric $CH_4$ was approaching steady state from 1983-2006, with no trend in emissions (Dlugokencky *et al.*, 2003). Reasons for this increase are still under discussion (Heiman, 2011; Dlugokencky *et al.*, 2011; Dalsøren *et al.*, 2016). Methane emissions from natural wetlands account for around 30% of total methane emissions (Kirschke *et al.*, 2013). Wetlands and peatlands are the major sources of natural biosphere methane emissions, and most peatland is located in high northern latitudes, whereas large wetland areas are located in the tropics. Emissions from natural biosphere sources have strong seasonal and interannual variability (Spahni *et al.*, 2011), contributing substantially to seasonal and interannual variability in atmospheric methane burden (Meng *et al.*, 2015). Additionally, photochemical reaction with OH in the troposphere, the major sink of methane, has strong effects on the annual cycle of the atmospheric methane. Variations in emissions of methane of both anthropogenic and biogenic sources are not sufficiently understood, which makes it difficult to attribute the observed changes in $CH_4$ burden to changes emission sources.



Several inverse models have been developed to estimate methane emissions and their contribution to the atmospheric methane burden (e.g. Bousquet *et al.*, 2006; Bruhwiler *et al.*, 2014; Houweling *et al.*, 2014; Fraser *et al.*, 2013; Meilink *et al.*, 2008). Emission estimates vary among models and the way they are set-up (e.g. Kirschke *et al.*, 2013; Locatelli *et al.*, 2013; Bergamaschi *et al.*, 2015; Tsuruta *et al.*, 2016) because these inverse systems rely on assumptions about emission

magnitudes and patterns, and on specific choices in the design of the inverse problem. Inputs, such as prior emission fields, and observations, and the transport model used in inversions play a major role in regional and continental emission estimates. Depending on the optimization method and available information, small spatial-scale optimization may not be possible. For example, the computational cost in adjoint models (Bergamaschi *et al.*, 2015; Belikov *et al.*, 2016; Houweling *et al.*, 2015; Meirink *et al.*, 2008) does not depend much on the number of scaling factors, used to 'scale' the prior – first

guess of emission estimates – to get optimized (posterior) emissions, i.e. such models have the ability to do grid-scale optimization globally. The computational cost in some other methods, e.g. Thompson and Stohl (2014) and Zhao *et al.* (2009), depends on the number of scaling factors because their covariance matrix is calculated directly. In this approach, grid-scale optimization is possible without any asymptotic assumption, but only for the regional domains because the dimensions of the covariance matrix for a global domain become too large even for present computational capability.

Ensemble Kalman filter (EnKF) based models (Bruhwiler *et al.*, 2014; Tsuruta *et al.*, 2016) have much less computational limitation related to the number of scaling factors. By representing the state covariance matrix with some samples of the state (ensemble members), the computational cost depends mostly on the number of ensemble members (Tsuruta *et al.*, 2016). Together with dimension reduction based on regional information, these models are able to optimize emissions globally, but still approximate the cost function minimum.

Simultaneously estimating biospheric and anthropogenic contributions to the $CH_4$ budget requires complex inverse approaches especially when both emissions are in the same location. To estimate biospheric emissions, information from an underlying ecosystem distribution map is useful, which defines the location of the sources and can help distribute larger regions over which the atmospheric signals integrate (Peters *et al.*, 2010; van der Laan-Luijkx *et al.*, 2015). In the case of

methane, where emissions depend on soil properties (Spahni *et al.*, 2011), the distribution of wetlands and their inundation extent can be used. This approach has the advantage that emission estimates from difference source categories and ecosystem types can be optimized separately by applying different scaling factors. However, it has a disadvantage that the spatial distribution relies much on prior estimates, and that emissions cannot be assigned to regions outside of the predefined source regions, i.e. if the distribution in the prior or the ecosystem map is wrong, the emission estimates would not be

optimized appropriately. The application of land-ecosystem distribution maps and their effect on methane emission inversions was discussed by Tsuruta *et al.* (2016) for a short period during summer 2007. They concluded that the contributions of methane emission sources, to the regional methane budget, are sensitive to the prior, observations, transport, and the number of scaling factors per region. Sensitivities to some aspects of model set-up are further evaluated in this study with multiple-year inversions. Inverse models rely on how the emissions are transported in the atmosphere, which is



determined by the transport model. The differences between observed and simulated atmospheric mole fractions are often the quantity to be minimized, and depending on the transport model, the regional flux estimates can be different by up to 150% on grid-scale (Locatelli *et al.*, 2013). One important feature is the inter-hemispheric (IH) exchange rate, which has strong effects on the north-south gradient (Locatelli *et al.*, 2013). It was shown by Olivie *et al.* (2004) that the convection

scheme influences transport from the NH to the SH.

One way to evaluate the different model configurations is to compare simulated atmospheric mole fractions to observations. The evaluations at sites used in the inverse problem are useful for checking the statistical consistency of the $CH_4$ emission estimates in relation to the observations. Furthermore, observations not used in the inversions, often called independent

observations, give information about model performance at additional locations and time. Observations such as aircraft profiles and ground- and satellite-based dry air total column-averaged mole fractions ($XCH_4$) are useful independent data to evaluate the vertical and long-range transport. It should be noted that observations uncertainties also have to be taken into account in the model, although these are generally smaller than the uncertainties in the model estimates.

In this study, we examine the emission estimates from CarbonTracker Europe-$CH_4$ (CTE-$CH_4$) for 2000-2012 with three configurations developed in Tsuruta *et al.* (2016). CTE-$CH_4$ is a version of the European branch of CarbonTracker data assimilation systems (Peters *et al.*, 2010, van der Laan-Luijkx *et al.*, 2015). Two configurations were applied to examine (1) the influence of the underlying land-ecosystem map (configuration C1) and (2) the influence of the transport model (configuration C2). These configurations complement the tests presented in Tsuruta *et al.* (2016) because a long-term

simulation will allow us to examine whether the finding from a short-period study is systematic or due to the choice of a season. For the configuration C1, we vary the shapes and sizes of the land-ecosystem map to assess whether optimizing one scaling factor is enough for the regions where we expect $CH_4$ flux errors to be strongly correlated. Using smaller regional divisions leads to more unknowns in the inverse problem, but also allows greater freedom to fit the observations by adjusting the fluxes with additional scaling factors to be optimized. For the configuration C2, we investigate how the differences in

vertical mixing by convection influences the derived methane emissions by applying two variants of the convection scheme in the atmospheric chemistry transport model TM5 (Krol *et al.*, 2005). For the evaluation, simulated atmospheric mole fractions are compared with data from in-situ observation sites, and with aircraft profile observations in Europe. The aircraft profiles are from a campaign within the European project CarboEurope, which is a part of EU funded IA (Integrating Activity) project within the Integrated non-$CO_2$ Greenhouse gas Observation Systems (InGOS), and Infrastructure for

Measurement of the European Carbon Cycle (IMECC). Additionally, the $XCH_4$ model estimates will be compared to ground-based $XCH_4$ observations from the Total Carbon Column Observing Network (TCCON; Wunch *et al.*, 2011), and satellite-based $XCH_4$ retrievals from TANSO-FTS instrument on board Greenhouse gases Observing SATellite (GOSAT; Kuze *et al.*, 2009).



## 2. Methods and Datasets

### 2.1 CTE-CH$_4$

CTE-CH$_4$ is an atmospheric inverse model that optimizes global surface methane emissions region-wize (Tsuruta *et al.*, 2016) based on ensemble Kalman filter (EnKF; Evensen, 2003). In this study, we optimize anthropogenic and biospheric emissions, and assume emissions from other sources (fire, termites, and ocean) constant (see Section 2.2). Two version of CTE-CH$_4$ were used: v1.0 and v1.1. The two versions differ by number of scaling factors, which are associated with the regional definition. The CTE-CH$_4$ v1.0 optimizes either anthropogenic or biospheric emissions per region, whereas v1.1 optimizes both emissions per region. The regional definitions are a combination of modified TransCom (mTC) regions (Fig. 1) and land-ecosystem regions (Fig. 2). The land-ecosystem map is defined based on Prigent *et al.* (2007) and Wania *et al.* (2010), as in the LPJ-WHyMe vegetation model (Spahni *et al.*, 2011), and described in Tsuruta *et al.* (2016). In CTE-CH$_4$ v1.0, biospheric emission is optimized if a region is defined as either inundated wetland/peatland or wet mineral soils; elsewhere, anthropogenic emission is optimized. Combining the two regional definitions, we have 62 regions for CTE-CH$_4$ v1.0, and 78 regions for CTE-CH$_4$ v1.1 to be optimized.

The prior covariance structures are based on Tsuruta *et al.* (2016). Variance of the scaling factors was set to 0.8 for all regions, except for 'Ice' region (Fig. 2). For CTE-CH$_4$ v1.0, an informative covariance matrix was used; the scaling factors for biospheric and anthropogenic emissions were assumed to be independent, and biospheric scaling factors were assumed to be correlated among mTC regions based on distance between the centers of the regions (Tsuruta *et al.*, 2016). For CTE-CH$_4$ v1.1, a non-informative covariance matrix was used, i.e. all regions were assumed to be independent.

The atmospheric chemistry transport model TM5 (Krol *et al.*, 2005) was used as an observation operator (Tsuruta *et al.*, 2016). Global total annual mean atmospheric chemical loss, i.e. the integrated OH, Cl and O($^1$D) losses during 2000-2012 , were calculated based on Houweling *et al.* (2014) and Brühl and Crutzen (1993), and estimated to be 516±11 Tg CH$_4$ yr$^{-1}$. We did not apply interannual variability in the monthly concentrations and removal rates of the sinks, although Ghosh *et al.* (2015) and Dalsøren *et al.* (2016) obtained a decrease in the CH$_4$ lifetime in their simulations, and on the other hand, Rigby *et al.* (2008) showed that the decrease in tropospheric OH concentration could be one of the reasons for the increase in atmospheric methane after around 2007. In this study, two versions of convection schemes in TM5 were used: Tiedtke (1989) and Gregory *et al.* (2000). The two versions differ mainly in vertical mixing in the troposphere: mixing is faster, and atmospheric methane mole fractions at the surface in the Northern Hemisphere (NH) are expected to be lower with Gregory *et al.* (2000) compared to Tiedtke (1989). Gregory *et al.* (2000) produces faster vertical mixing around the surface and also shorter IH exchange time compared to Tiedtke (1989).



## 2.2 Prior CH₄ emissions

Five prior emission fields were used in this study, representing $CH_4$ release from anthropogenic, biosphere, fire, termites, and oceanic sources. Anthropogenic emissions accounted for c.a. 50% of global total annual $CH_4$ emissions during 2000-2012. For prior anthropogenic emissions, the Emissions Database for Global Atmospheric Research version 4.2 FT2010
(EDGAR v4.2 FT2010) inventory was used. The original inventory data coverage extends to 2010; for 2011-2012, emission fields were assumed to be same as for 2010. Tuner *et al.* (2016) suggested that a large increase in United States anthropogenic emissions contributed significantly to global growth in $CH_4$ emission during 2002-2014. Although we did not include the 2010-2012 increase in the prior, we expect CTE-CH₄ to be able to optimize such changes. A seasonal cycle is not included in the EDGARv4.2 FT2010 inventory. Emission estimates from the biogeochemistry model LPX-Bern v1.0
(Spahni *et al.*, 2013) were used as prior biosphere emissions. However, emission estimates from rice fields were excluded from the prior biosphere emissions because they were already included in the prior anthropogenic emissions. Additionally to the emissions, consumption of methane by methanotrophic bacteria in soils is accounted in LPX-Bern estimates, which are taken into account in CTE-CH₄ as negative fluxes. For prior emission estimates from fires, GFEDv3.1 (Randerson, 2013; van der Werf *et al.*, 2010) was used, and 2011-2012 emission fields were assumed unchanged, as the original data coverage
is up to 2011. Compared to 2011, global fire emission in 2012 was about 2 Tg $CH_4$ yr$^{-1}$ higher, mainly due to increase in emissions in northwest Russia during summer (GFEDv4.1; Giglio *et al.*, 2013). Therefore, we must be aware of an additional uncertainty in a spatial distribution of the emission sources, especially for 2012. For emission estimates from large scale biomass burning, GFEDv3.1 is used rather than the estimates in EDGAR v4.2 FT2010. For prior termite emissions, the estimates from Ito and Inatomi (2012) were used for 2000-2006. The 2006 estimate was also used for 2007-2012. Estimates
by Ito and Inatomi (2012) are about 10 Tg $CH_4$ yr$^{-1}$ smaller compared to Sanderson (1996) used in e.g. Bergamaschi *et al.* (2007). Prior emission estimates from 'natural' open ocean are calculated assuming a supersaturation of $CH_4$ in the seawater of 1.3 (Lambert & Schmidt, 1993). The ECMWF ERA-Interim sea surface temperature, sea ice concentration, surface pressure and wind speed (Dee *et al.*, 2011) were used for calculating the solubility and the transfer velocity (Bates *et al.*, 1966; Tsuruta *et al.*, 2015). No special treatment was applied to coastal emissions of the 'natural' ocean. In addition to the
'natural' ocean emission estimate, 'anthropogenic' ocean emission estimate from EDGAR v4.2 FT2010 were added to the prior. Sources of anthropogenic ocean emissions are mainly from ship and other 'non-road' transportation. This includes emissions around coastlines. Prior emission estimates from land and ocean anthropogenic sources, and from land biosphere sources, were optimized. Estimates from fire, termites and natural ocean sources were held constant and not optimized.

## 2.3 Atmospheric methane observations

Atmospheric observations of $CH_4$ dry-air mole fractions from the World Data Centre for Greenhouse Gases (WDCGG) were assimilated in CTE-CH₄. The set of observations consists of discrete air sample and continuous measurements from several cooperative networks (Table 2). The observations were filtered in order to avoid inversions to be influenced by strong local



signals, and based on observation uncertainty and representatively of transport model, model-data-mismatches (mdm) are defined (Tsuruta *et al.*, 2016). During assimilation, rejection thresholds were set as three times mdm, except for sites with mdm=4.5 ppb. For these sites, rejection thresholds were set as 20 times because assimilation of these observations is important to characterize the background atmospheric $CH_4$ well.

**2.4 Aircraft profiles for validation**

Aircraft profiles provide information about atmospheric $CH_4$ in general, but specifically concerning vertical transport. For validation we used aircraft data from regular profiling operated within the European CarboEurope project at Orléans (France), Bialystok (Poland), Hegyhatsal (Hungary) and Griffin (U.K.) during 2006-2012 (Table 3). In addition, we used data from an aircraft campaign performed within the Infrastructure for Measurement of the European Carbon Cycle (IMECC) project. The IMECC campaign deployed a Learjet 35a with multiple vertical profiles close to the surface up to 13km nearby several TCCON sites in central Europe. For details on the airborne $CH_4$ measurements the reader is referred to Geibel *et al.* (2012). The aircraft observations were not assimilated in the inversions.

**2.5 XCH4 dataset for validation**

In addition to the aircraft profiles and the mole fractions at in-situ stations, column-averaged dry-air mole fractions (XCH4) from the TCCON network and the TANSO-FTS instrument on board the GOSAT spacecraft were used for validation, as those provide additional information about the long-range transport and help to assure the quality of the simulations globally. The TCCON observations from the GGG2014 release (Wunch *et al.*, 2015) were used, and the daily mean was compared to the simulated XCH4 at each site. For GOSAT retrievals, the product reported by Yoshida *et al.* (2013) was used, and regional daily mean was compared to the corresponding simulations. The XCH4 datasets were not assimilated in the inversions.

For the validation using GOSAT and TCCON retrievals, the posterior total column dry air mole fractions (XCH4) were calculated from global 4°×6°×25 (latitude, longitude, vertical levels) daily 3-dimensional (3D) mole fraction fields from the TM5 chemistry transport model. For each observation, the 3D global daily mean gridded mole fraction estimates were horizontally (latitude, longitude) interpolated to the location of the observations to create the vertical profile of the simulated mole fractions. For both the GOSAT and TCCON retrievals, the averaging kernels (AK) were applied to model estimates based on Rodgers and Connor (2003):

$$\hat{C} = c_a + (\boldsymbol{h} \circ \boldsymbol{a})^T (\boldsymbol{x} - \boldsymbol{x_a}), \tag{3}$$

where $\hat{C}$ is the quantity for comparison, i.e. XCH4. $c_a$ (a scalar) is a prior column dry-air mole fraction of each observation, $\boldsymbol{h}$ is a vertical summation vector, $\boldsymbol{a}$ is an absorber-weighted averaging kernel of each observation, $\boldsymbol{x}$ is a model profile, and $\boldsymbol{x_a}$ is the prior profile of the observation. For the TCCON observations, one prior profile is provided each day, which is scaled to get the observed profiles that optimizes the spectral fit (Wunch *et al.*, 2011). Prior profiles of GOSAT retrievals are



provided for each observation (Yoshida *et al.*, 2013). For the comparison with TCCON XCH$_4$, model estimated XCH$_4$ were calculated for each site, and with the GOSAT retrievals, the spatial mean of XCH$_4$ for each mTC region was compared.

### 2.6 Inversion setups

In this study, three inversions were performed: (M1) using CTE-CH$_4$ v1.0 with Tiedtke (1989) convection scheme, (M2) using CTE-CH$_4$ v1.1 with the Tiedtke (1989) convection scheme, and (M3) using CTE-CH$_4$ v1.0 with the Gregory *et al.* (2000) convection scheme (Table 1). Prior and posterior CH$_4$ mole fractions were estimated with TM5 using prior and posterior emission estimates, respectively. M1 and M2 posterior mole fractions were estimated with the convection schemes that was applied to the inversions and forward runs for each inversion.

## 3. Results

### 3.1 Atmospheric mole fractions

Agreement of posterior atmospheric mole fractions with observations was significantly improved compared to the prior (Fig. 3). Atmospheric CH$_4$ from the prior increased continuously during 2000-2012 (Fig. 4), and became much higher than the observations, especially in the NH (Fig. 3). This is improved by the inversion by decreasing emissions compared to the prior. The phase of CH$_4$'s seasonal cycle calculated with prior emissions agreed poorly with the observations with a positive bias in CH$_4$ mole fractions from winter to summer in the NH, and around the end of each year in the SH (Fig. 3).

Furthermore, prior mole fractions were biased low compared to the observations in the SH around 2002-2004. This could be due to underestimation in the prior emission estimates in some regions, likely in the Southern Hemisphere (SH). The posterior mole fractions generally matched the observations well. Some seasonal bias remains (especially in M1), and the decrease in CH$_4$ mole fractions in the SH around 2002-2004 remains in the posterior mole fractions, but the period with large negative bias is shorter and the magnitude is smaller than the prior mole fractions (Fig. 3). This is an indication that the inversion successfully adjusted the prior emission estimates to match the observations. However, some negative biases were found in the posterior mole fractions around the equator, which remained unresolved throughout the study period in all inversions. Strong negative bias was found in Bukit Koto Tabang, Indonesia (BKT) (-25 to -27 ppb) and Mt. Kenya, Kenya (MKN) (-18 to -23 ppb), such that the posterior mole fractions were especially low during June-October. This suggests that measurements at those latitudes are not representative of large regions optimized in the model. Furthermore, there could be some 'missing' emissions in some regions, which were not constrained well by the inversion, at least not without deteriorating agreement at other measurement stations. Posterior emission estimates for south American tropics (mTC3) were similar to the prior, and the inversion could not significantly decrease the uncertainty of the prior emission estimates in the region (see Section 3.4.4 and 4.2). If the prior uncertainty is too small, emission estimates cannot change much from the prior, but it is probably not the reason here; the prior emission uncertainty was defined to be 80% of the prior emission



estimates, which should be a large enough to allow the scaling factors to change. When the prior biosphere emissions in the south American tropics are increased, agreement near the equator and in the SH during 2002-2004 improved, although the regional emission estimates elsewhere in the SH were decreased to compensate for the increase (not shown). We cannot conclude whether it is due to biosphere or anthropogenic emissions and is actually missing in the south American tropics, but
these results suggest that some underlying emission in the tropics could have been missing in this study.

Agreement between simulated $CH_4$ and surface observations was better for M2 and M3 than for M1 (Fig. 3) such that the root mean squared error (RMSE) is about 0.5 ppb smaller, and the biases in annual amplitude are about 1-2 ppb smaller. The negative bias from 2002 to 2004 is mainly seen in M1, while in M2 and M3, it is prominent only in 2002. Although the
difference in the average RMSE is small, it is significant considering that it is calculated from all the observations assimilated in the study period. The lowest global total emission estimates were found in M1 around 2002-2004, which created the largest growth rate in the global total emission estimates from 2001-2006 (B2007) to 2007-2012 (A2007). Based on previous studies, we found that the increasing rate of emission estimates in M2 and M3 are more reasonable (see Section 3.4.1). The differences in RMSE and bias between M2 and M3 are small at around 30°N, where many observations are
located. However, RMSE and bias in M2 are about 1 ppb and 2 ppb smaller, respectively, at high northern latitudes (60°N-75°N), and about 3 ppb and 6 ppb larger, respectively, around the equator (EQ-15°N) than in M3. Note that the 2002-2004 low mole fractions in the SH are not as prominent in the prior when using Gregory *et al.* (2000) convection scheme (Fig. 3) probably due to enhanced transport between the NH and SH in M3.

The growth rate (GR) of atmospheric methane mole fractions showed that the posterior estimates are closer to the observations than the prior, as expected (Fig. 4). The GR of the prior $XCH_4$ increased almost continuously in B2007, i.e. the increase started faster than the posterior which started to increase in around 2007 (Fig. 4). On the other hand, the GR of the posterior $XCH_4$ did not change much B2007, but increased in A2007 where all inversions show an increase in $XCH_4$ by about 5 ppb in A2007, with some seasonal and interannual variations (Fig. 4). The posterior estimates are in line with the GR
calculated from global network of marine boundary layer observations (Dlugokencky *et al.*, 2011), indicating that the GR in the prior is probably overestimated throughout 2000-2012. Note that the observations compared in Fig. 4 is calculated from surface observations, i.e. the GR of estimated $XCH_4$ is expected to fluctuate less.

**3.2 Validation with aircraft measurements**

Validation using $CH_4$ measurements from aircraft showed that the posterior mole fractions generally agree well with
independent observations which were not assimilated in the model (Fig. 5), and average RMSE is decreased from 80 ppb in the prior to 24 ppb in the posterior (Fig. S1). The RMSE between posterior and observed mole fractions showed that the posterior match best for GRI (<12.9 ppb), and worst for ORL (>37.4 ppb) (Fig. 5). For GRI, model performance at nearby



sites is good, i.e. the correlations between assimilated observations and posteriors are high, and the RMSE is equal to or smaller than mdm (Fig. 6). This suggests that emission estimates are well constrained and the emission distribution in the prior is good. On the other hand, model performance of sites near ORL is poor. Indeed, the bias in ORL profiles extends up to 2km, indicating background $CH_4$ concentration may not be as well constrained as at other sites. The $CH_4$ mole fractions at

other sites agree well for altitudes higher than 1.5km. The comparison with IMECC observations from central Europe shows the effect of the convection scheme on the profile at altitudes above 2km. Negative biases are seen in M1 and M2 estimates from 2-10km. Bias in the M3 estimates are small around 2-10km, but it is positive in the upper troposphere and lower stratosphere, where M1 and M2 match the observations better. Note that we used TM5 with only 25 vertical layers – using a higher vertical resolution of TM5 may resolve part of the discrepancy.

**3.3 Validation with TCCON and GOSAT XCH₄**

The total column-averaged dry-air mole fractions (XCH₄) provide additional information about the atmospheric methane distribution. TCCON and GOSAT XCH₄ retrievals are not assimilated in the inversions, so the following comparisons also allow us to assess model performance at independent locations and times globally.

Posterior XCH₄ at TCCON sites generally agreed well with the observations (Table 5, Fig. 7, Fig. S2). The increasing trend was much stronger than the observations in the prior (not shown), but it was corrected in the posterior. When the averaging kernel was applied, posterior XCH₄ became slightly lower than without it, at all sites. Agreements with the observations was better (RMSE was smaller) when the averaging kernel was not applied at most of the sites (especially for M1 and M2), and bias in the north-south gradient was larger when the averaging kernel was applied (Supplementary Table S2-S3). However,

the posterior XCH₄ values with the averaging kernel applied should be more comparable to the observations, therefore we applied it for the comparisons. The RMSE between observations and posterior XCH₄ showed that agreement was better in the NH than in the SH. For many sites in the NH, M1 and M2 XCH₄ were slightly lower than observed, especially when the averaging kernel was applied, but the trend and seasonal variability are well captured. For Izana (Spain), posterior XCH₄ increased much stronger than the observations (Fig. 7). However, this could be due to some missing spring observations with low XCH₄ in 2007-2008. On the other hand, $CH_4$ mole fractions are much smaller in the stratosphere than in the troposphere.

Therefore, XCH₄ is very sensitive to the tropopause altitude. The subtropical jet is located near the Izaña latitude. A small mistake in the location of this jet might have a large impact on the XCH₄ estimated by the model for this station. For SH sites (Darwin, Wollongong and Lauder), a strong negative bias was found in all inversions (Fig. 7). For Darwin, the estimated posterior surface mole fractions at the nearest in-situ site, Gunn Point, Australia (GPA) had negative biases of -25

to -30 ppb and were very weakly correlated (<0.1) in all inversions. This finding is in line with Monteil *et al.* (2013) and Locatelli *et al.* (2015). Data were available only after mid-2010, and the mdm was set high (75 ppb), so the inversion probably did not benefit much from these observations. Agreement was especially bad for Wollongong, with the highest RMSE of all inversions, more than 30 ppb (Table 5). The nearest in-situ site to Wollongong is Cape Grim, Australia (CGO).



The estimated posterior surface mole fractions at CGO showed some negative bias (-6 to -11 ppb), but the correlation with the observations were high (>0.85), indicating that the seasonal cycle at the surface site was well captured in the model. For Lauder, New Zealand (LAU), some negative bias (-12 to -15 ppb) was found at the surface sites, but the correlation was strong (>0.85) in all inversions. The trend, interannual variability and seasonal variability of posterior $XCH_4$ were also captured well, and the bias was not as strong as in Darwin and Wollongong. Since the negative bias increases in absolute value towards the equator, i.e. is larger at Wollongong than at Dawin and LAU, so it could indicate emissions are too low around the equator. Indeed, when the prior emission estimates in the south American tropics (mostly in 15°S-15°N) was increased (see Section 3.1), the negative bias decreased (not shown). Our estimates for the tropics (30°S-30°N) are about 10-20 Tg $CH_4$ yr$^{-1}$ lower the estimated by e.g. Houweling *et al.* (2014), and the agreement at Darwin, Wollongong and Gunn Point became better when the prior emissions in the south American temperate region were increased (not shown).

Differences in M1 and M2 $XCH_4$ were small. For M3, agreement with the observations was best among the inversions. The RMSE between the estimates and observations were the smallest in M3 at all sites, except Garmisch, Germany (Table 5). Garmisch is a mountain site (altitude 734m a.s.l.), and the mean of observed $XCH_4$ was lower than at near-by sites, e.g. Karlsruhe, Germany, and Bialystok, Poland. However, the mean posterior $XCH_4$ at Garmisch was similar to near-by sites; M3 gave the highest values of $XCH_4$ and the largest differences with the observations.

The comparison with GOSAT $XCH_4$ showed that all inversions have a negative bias in the SH regions (south American tropics (mTC3), the south American temperate region (mTC4), south Africa (mTC6), and Australia (mTC10)). This is in agreement with the TCCON comparison. For the NH, M1 and M2 have slight (about 2 ppb) positive biases, but agreed well with the retrievals, whereas M3 $XCH_4$ has a positive bias of about 10 ppb (Fig. 8), i.e. the north-south gradient of the GOSAT $XCH_4$ was better captured in M1 and M2 than in M3. The seasonal amplitude is significantly larger in the posterior than in the GOSAT $XCH_4$, especially in the SH (Fig. 8). This may suggest that the seasonality in the emission estimates is not well optimized, or the seasonal cycle of vertical mixing in the transport model is not well represented. The seasonal variability and amplitude are driven by both anthropogenic and biospheric emissions, and by the OH sink. Since anthropogenic emissions dominate in many mTC regions, seasonal variability in the posterior biosphere emission estimates often stayed similar to the prior. The optimized seasonal variability in the anthropogenic emission estimates could also be incorrect. However, these are not in line with the comparison with the TCCON $XCH_4$, which showed slight negative bias in posterior $XCH_4$ in the NH, and no significant differences in the seasonal amplitudes. This could be partly due to the bias in GOSAT $XCH_4$ (Yoshida *et al.*, 2013) or validation of the retrievals, such that the TCCON validation dataset is different from the observations presented in this study. Nevertheless, an important model feature about the spring increase in $XCH_4$ was found in the comparison with GOSAT $XCH_4$. The seasonal cycle of global and Ocean (mTC16-20) $XCH_4$ were better captured by the M3 inversion pointing to an important role for vertical mixing. The spring peak in the global mean, Asian tropics and Ocean were well captured in M3, but not in M1 and M2 (Fig. 8). Monthly emission estimates in M3 were





generally higher than in M1 and M2 during winter and early spring (November-April), especially in northern latitude temperate regions (35°N-60°N, Fig. S6). In previous inversion studies using satellite observations, a spring increase in estimated $XCH_4$ has also been reported (e.g. Fraser *et al.*, 2013). This suggests that winter emissions in northern latitude temperate regions, enhanced by faster vertical mixing around surface, play an important role in the $XCH_4$ seasonal cycle in

the tropics.

It must be noted that there are surface in-situ observational sites often located near the TCCON sites that are assimilated in the model. Furthermore, the GOSAT $XCH_4$ are evaluated against the TCCON observations, so retrieved $XCH_4$ at locations far from the TCCON sites may be biased. Therefore, the disagreements in the north-south gradient and seasonal amplitudes

may not only be due to the problem in the inversions.

### 3.4 Emission estimates

### 3.4.1 Global

Average global total emission during 2000-2012 is estimated to be $517\pm45$ Tg $CH_4$ yr$^{-1}$ with an increasing trend of 3 Tg $CH_4$ yr$^{-1}$ (Table 4, inversion M3). Average posterior global total emissions for 2000-2012 are about 29 Tg $CH_4$ yr$^{-1}$ lower than the

prior in inversion M3 (Table 4), although the posterior estimates are within the range of prior uncertainties ($\pm93$ Tg $CH_4$ yr$^{-1}$). The average posterior global total emission estimates from inversions M1, M2 and M3 agree well with each other (Table 4), and the estimates are in line with previous studies, e.g. Bousquet *et al.* (2006) and Fraser *et al.* (2013). However, anthropogenic annual mean emission estimates in M2 are more than 10 Tg $CH_4$ yr$^{-1}$ larger than in M1 and M3. Further, the posterior biospheric annual mean emissions in M2 are smaller than in the other two inversions (Fig. 9).

All inversions showed an increase in posterior mean global total emissions from B2007 to A2007 by about 18-19 Tg $CH_4$ yr$^{-1}$ (Table 4), which is somewhat smaller than the increase in the prior estimates of 33 Tg $CH_4$ yr$^{-1}$. The increase in posterior emissions during 2000-2010 was 15-16 Tg $CH_4$ yr$^{-1}$; this agrees well with previous studies by e.g. Bergamaschi *et al.* (2013) and Bruhwiler *et al.* (2014), who estimated an increase to be about 16-20 Tg $CH_4$ yr$^{-1}$.

The increase in global total emission was dominated by the increase in anthropogenic sources in both posterior and prior, but the increase in the posterior (15-28 Tg $CH_4$ yr$^{-1}$) was much weaker than the prior (37 Tg $CH_4$ yr$^{-1}$) (Table 4). The increase in the posterior estimates from 2003-2005 to 2007-2010 was 15-23 Tg $CH_4$ yr$^{-1}$, which agrees well with Bergamaschi *et al.* (2013) who estimated the increase to be smaller than the EDGAR v4.2 inventory, and about 14-22 Tg $CH_4$ yr$^{-1}$. However,

30  increase in anthropogenic emission estimates was smaller in Bruhwiler *et al.* (2014) who found an increase of about 10 Tg $CH_4$ yr$^{-1}$. Differences in the inversions are partly due to different observations and priors. Bergamaschi *et al.* (2013) used with satellite-based retrievals from SCIAMACHY and NOAA observations, whereas Bruhwiler *et al.* (2014) used in-situ



NOAA discrete and Environmental Canada (EC) continuous observations. Our study is also based on in-situ observations, but we have more discrete and continuous observations globally than the previous two studies, so the estimates from our study could potentially contain important additional information from observations other than NOAA and EC. For the prior estimates, this study and Bergamaschi *et al.* (2013) used EDGAR v4.2 inventory estimates (although slightly different versions, the estimates are similar), while Bruhwiler *et al.* (2014) used a constant prior from EDGAR v4.2 for 2000. Bergamaschi *et al.* (2013) found that although a significant increase in anthropogenic emissions was seen in constant-prior inversion, the increase was slightly lower than in their inversions with the trend included in the prior. The biosphere emission estimates for A2007 were found to be smaller than that of B2007 in the M1 and M3 inversions (-5 to -2 Tg $CH_4$ $yr^{-1}$), following the prior (-1 Tg $CH_4$ $yr^{-1}$), whereas the M2 inversion showed an increase (+7 Tg $CH_4$ $yr^{-1}$). The biosphere emission estimates in the M2 inversion for B2007 was much lower than in the other two, mainly due to unreasonably low estimates in the Asian temperate region (discussed in Section 3.4.3). The decrease in biospheric emissions in M1 and M3 agree with the finding in Bergamaschi *et al.* (2013), although our estimates have a stronger decreasing rate. This could be due to an overestimation of global anthropogenic emissions. Unreasonably low anthropogenic emission estimates in the Asian temperate region (discussed in Section 3.4.3) could have created a strong increase in the global anthropogenic emission estimates, which then led to strong decrease in the biospheric emission estimates. Note that we did not include interannual variability or trends in the methane sink in the atmosphere.

### 3.4.2 Northern boreal regions and Europe

In this section, we present results for the following regions: north American boreal region (mTC1smaller), Eurasian boreal region (mTC7), and Europe (mTC11-14).

Posterior anthropogenic emissions for Europe as a whole (mTC11-14) were similar to the prior (M1, M2) (Table 4). However, the relative contribution of different emission regions for posterior emissions was different than the prior. Posterior emissions were higher than the prior in southern Europe (south west Europe (mTC11) and south east Europe (mTC12)), where they were smaller than the prior in north east Europe (mTC14) in all inversions (Table S1). Most of the increase in southern Europe and the reduction in north east Europe came from anthropogenic sources. The observed mole fractions during winter at many of the in-situ sites in northern Europe can be good indicators of anthropogenic signals, because emissions from biogenic sources are small during winter. Winter mean posterior mole fractions at these sites agree well with the observations, indicating that the posterior anthropogenic emissions are reasonable. Southern Europe is a small source of biosphere emissions, so most of the atmospheric signals captured at the in-situ sites in this region are from anthropogenic sources. In southern Europe, posterior mole fractions at some sites in France, Spain and Italy have strong positive bias (> 10 ppb), although the correlations are good (about 0.8 or larger). The posterior mole fractions at other sites in south east Europe are not overestimated, but the correlations are often worse. This may imply that the inversion could not find a solution that matches all the observations equally, because of an incorrect distribution in the prior within the region or



some measurements having too much local influence or too small mdm. Validation with aircraft observations showed that transport in Europe is generally good, but validation data are only from central Europe, i.e. mixing in the atmosphere elsewhere can be also a part of the problem. Anthropogenic emission estimates for north west Europe were similar to the prior. This finding is in line with Bergamaschi *et al.* (2015), who estimated the anthropogenic emissions in north west

European countries to be similar to the EDGARv4.2 estimates and higher than the emissions reported in UNFCCC (2013).

For biospheric emission estimates, differences between prior and posterior estimates were negligible in southern Europe, whereas the reduction in the posterior was clear in northern Europe (north west and north east Europe) (Fig. 10, Fig. Sx). A reduction in biospheric emission estimates was also seen in the north American boreal region (Fig. 10)). This suggests that

the prior biosphere emissions in Boreal regions are too high, which gives higher prior $CH_4$ mole fractions than observed. The interannual variability in the posterior emissions also does not follow the prior. For 50°N-90°N, we found an increase in the biosphere emission estimates in 2006, followed by a decrease until 2010. The estimates increased again in 2011, and decreased in 2012. Most of the increase in estimated biosphere emissions in 2006 in our study was from the north American boreal region. The increase in 2006 does not agree with previous studies, e.g. Bousquet *et al.* (2011), who found little

increase from high northern latitude wetland emissions in 2006, but a significant increase in emissions there in 2007. Although their representativeness of a regional scale signal is questionable, site level emission observations support the findings for the year-to-year variability in the regional scale posterior emission estimates. Moore *et al.* (2011) showed that 2006 was a warm and wet year, and during 2004-2008, the highest autumn $CH_4$ emissions were observed in 2006 in Mer Bleue bog in Canada (45.41°N, 75.48°W). In the prior, the peak in biospheric emission in 2006 was also seen in northern

Europe, but the posterior estimates were higher in 2007 than in 2006. The posterior biospheric emission estimates for north east Europe in 2006 were about 60% lower than the prior estimate in all inversions. Monthly emission estimates showed that the main differences were found during summer and autumn. Drewer *et al.* (2010) found that $CH_4$ emissions in September in Lompolojänkkä fen in Finland (67.60°N, 24.12°E) were higher in 2006 than in 2007 due to heavy rain. However, summer 2006 was dry with low emissions and snow started to fall already at the end of September, cutting the emission season short

with freezing temperatures, and thus mean annual $CH_4$ emission was lower than in 2007. The high emission in September-October in 2006 in the prior could be due to a bias in precipitation (excluding snow) and temperature in CRU meteorological data (Mitchell and Jones, 2005), which was used as an input for the LPX-Bern model. CRU precipitation and temperature were too high during autumn 2006 at Lompolojänkkä and for the mTC14 region on average, which could have caused emission estimates in LPX-Bern that were too large. On the other hand, the posterior summer biospheric emissions in 2007

were nearly twice the prior. LPX-Bern estimates were low during the whole summer and autumn at Lompolojänkkä and in mTC14, although posterior showed high emission in July. This could be due to problems in wetland fraction or problems in the precipitation dependence. CRU precipitation in 2007 was high in early summer and extremely heavy in July at Lompolojänkkä and in mTC14 on average, which is in line with Drewer *et al.* (2010). Although the seasonal cycle of the precipitation is well captured in CRU, if peatland soil was already saturated with water in early summer, methane emission



would not be increased by high summer precipitation in LPX-Bern. For north west Europe, similar differences in the seasonal cycles of the prior and posterior emissions were found: the prior biospheric emissions were high in summer-autumn in 2006 and low in summer in 2007 compared to the posterior. The CRU meteorology again agreed well with the measurements e.g. at Stordalen. Emissions measured at Stordalen mire in northern Sweden (68.20°N, 19.03°E) (Jackowicz-Korczyński *et al.*, 2010) again support the posterior estimates more than the prior. In this case too, CRU meteorology agrees well with observations at Stordalen mire.

The difference between the Tiedtke (1989) and Gregory *et al.* (2004) convection schemes was prominent in all northern boreal regions and Europe. The posterior emissions in M3 were higher than in M1 and M2 throughout 2000-2012. The estimated prior surface atmospheric $CH_4$ mole fractions in these regions are lower when using the Gregory *et al.* (2000) scheme than using the Tiedtke (1989) scheme. Prior mole fractions agreed better observations when using the Gregory *et al.* (2004) scheme than when using the Tiedtke (1989) scheme; mean bias and RMSE for the study period are more than 20 and 10 ppb smaller, respectively. This indicates that the stronger vertical transport, present when using the Gregory *et al.* (2004) scheme, lowers the surface mole fractions faster than the Tiedtke (1989) scheme. Therefore, posterior emissions are greater when the Gregory *et al.* (2000) convection scheme is used. We cannot conclude which convection scheme is better for northern boreal regions and Europe based on the posterior mole fractions of those regions. Posterior mole fractions in M1 and M2 inversions agreed better with the in-situ observations and the GOSAT retrievals than in the M3 inversion, whereas the agreement with the aircraft and TCCON observations were better in the M3 inversion than in the M1 and M2 inversions. Note that the number of available GOSAT retrievals at northern Europe is limited. These comparisons may indicate that atmospheric mole fractions at the surface is better optimized using the Tiedtke (1989) convection scheme, whereas vertical mixing is better optimized using the Gregory *et al.* (2000) scheme for northern Europe.

### 3.4.3 Northern temperate

In this section, we present results for the north American temperate (mTC2) and the Asian temperate regions (mTC8).

Posterior total emissions for the north American temperate region were greater than prior emissions in all inversions (Fig. 11, Table S1). The main contribution to the increase in regional total emissions came from the anthropogenic emission estimates. Posterior mean anthropogenic emissions for 2000-2001 are closer to the prior, and are nearly 10 Tg $CH_4$ $yr^{-1}$ greater than prior mean estimates for 2004-2012 (Fig. 5). The prior showed no significant trend during 2000-2012. Posterior emissions grew by 0.5 Tg $CH_4$ $yr^{-2}$ during 2000-2012, but the growth is not significant. The estimated growth rate is similar to the estimates reported by Bruhwiler *et al.* (2014), but only about one third of that reported by Turner *et al.* (2016). Although no significant increase was found in the posterior emissions for the north American temperate region in this study, the validation showed that the trend in the regional mean posterior $XCH_4$ matched the trend in GOSAT retrievals, as well as the trend in $XCH_4$ at TCCON sites in the USA, e.g. Park Falls and Oklahoma. Furthermore, posterior $XCH_4$ also matched





TCCON and GOSAT retrievals well (Fig. 7, Fig. 8). Because we only optimized emissions per region, and there was only one scaling factor for anthropogenic emission estimates in the north American temperate region, we were not able to study the differences in the emission trend on the east and west side of America, as in Tuner *et al.* (2016). However, our study suggests that a large increase in local emissions is not necessary needed to reproduce the increasing trend in $XCH_4$. Long-range transport played more important role than the local emissions in the $XCH_4$ trend.

A negative correlation was found between posterior anthropogenic and biosphere emissions for the north American temperate region, i.e. anthropogenic emissions increased when biosphere emissions decreased. This is an effect of the inversion not being able to separate biosphere and anthropogenic emissions based on the current observational network. The in-situ observation sites in this area are mostly located close to anthropogenic emission sources, i.e. the interannual variability found in biosphere emission estimates may not be representative of the real variability.

The Asian temperate region has large anthropogenic and biospheric emissions. Anthropogenic emissions are responsible for most of the increase in the prior regional and global total emission estimates for A2007. However, anthropogenic emission estimates in this region were reduced by more than half in the inversions (Fig. 11, Table 4). Moreover, the increase in posterior anthropogenic emissions for 2000-2012 is not as strong as in the prior (Fig. 11, Table 4). The reduction in anthropogenic emissions from prior to posterior estimates for 2002-2010 was driven by observations from two continental sites in Korea, Anmyeon-do (AMY, data available for 2000-2012) and Gosan (GSN, data available for 2002-2011), which small values of the mdm were initially assigned and thus had a large impact on the regional flux estimates. When the mdms for those sites were set to 1000 ppb, reducing their influence in the inversion (M1a, M2a), the estimated total emission in this region was about 30 Tg $CH_4$ $yr^{-1}$ larger and in better agreement with e.g. Bruhwiler *et al.* (2014) and Bergamaschi *et al.* (2013). In the latter simulations, the global total $CH_4$ budget did not change (Supplementary Fig. S1) as the increased emission estimates in the Asian temperate region were mainly compensated by reduced fluxes in the Asian Tropical region (about 10 Tg $CH_4$ $yr^{-1}$ in M1), as well as in the Eurasian boreal region, Europe, and Ocean (about 5 Tg $CH_4$ $yr^{-1}$ each in M1). Only small changes were found in the emission estimates elsewhere, but the increase in the 2009-2012 anthropogenic ocean emission estimates became smaller (M1a; M2a is run only for 2000-2006 as a test case). When excluding the two Korean sites from the inversion, the posterior biosphere emissions remained close to the prior and M1, and the interannual variability in total emissions for the Asian temperate region in M1a and M2a was smaller. We consider it unrealistic that the regional anthropogenic emission could change by more than 30 Tg $CH_4$ $yr^{-1}$ in one to two years as is now the case in M1, M2, and M3. Economically fast growing countries such as China and India are located in the Asian temperate region, and there is no evidence that the anthropogenic emissions decreased significantly during 2002-2010 in that region. Total emission estimates in M1a and M2a were higher and more reasonable than in M1 and M2, and the ratio of anthropogenic to biospheric emission estimates of M1a and M2a were more consistent with each other compared to M1 and M2. This suggests that the posterior emissions in M1a and M2a are more reasonable, i.e. the M1 and M2 posterior anthropogenic emission estimates and the M2



posterior biosphere emissions are probably unreasonably low. Nevertheless, the posterior emissions were lower than the EDGAR v4.2 FT2010 emissions, which is in agreement with previous inversion studies (Pandy *et al.*, 2016; Thompson *et al.*, 2015). The effect of the changes in the emission estimates (M1a and M2a) to XCH$_4$ was small, although a slight increase was found globally. The agreements with GOSAT and TCCON XCH$_4$ in M1a and M2a were slightly better in regions and at

5 sites where negative biases were found in M1 and M2 (not shown).

### 3.4.4 Tropics

In this section, we present results for the following regions: south American tropical (mTC3) and Asian tropical (mTC9).

Tropical Asia is another region with high anthropogenic and biospheric emissions. Prior estimates from both sources are

10 about 30 Tg CH$_4$ yr$^{-1}$ each. The posterior mean total emissions for A2007 are lower than the prior in all inversions. Posterior estimates for both anthropogenic and biospheric emissions in M3 were lower than the prior, while in M1 only biospheric emissions were lower, and in M2 only anthropogenic emissions were lower. The estimates in M3 were lower than in M1 throughout 2000-2012 for both anthropogenic and biospheric emissions. Posterior estimates for biospheric and anthropogenic emissions are lower than in the previous study by Bruhwiler *et al.* (2014), who estimated the anthropogenic

emissions are even higher than our prior, and biospheric emissions are similar to our prior. The M2 anthropogenic emission estimates were lower than the prior estimates due to enhanced, and probably unrealistic, interannual variability compared to the M1 and M3 estimates (Fig. 12). This partly correlates with the strong interannual variability in the emissions in the Asian temperate region. For example, the increase in anthropogenic emissions in M2 around 2002-2005 is due to a strong decrease in emissions in the Asian temperate region. In the test case M1a, the interannual variability in both the Asian temperate

region and Asian tropics was not as large (Fig. S4). For both anthropogenic and biospheric emissions, the estimates were lower in M3 than in M1, i.e. the Gregory *et al.* (2000) convection scheme leads to higher atmospheric CH$_4$ mole fractions at the surface than the Tiedtke (1989) convection scheme. Validation with surface in-situ observations showed that the inversions had difficulties reproducing observed CH$_4$ mole fractions at BKT, such that strong negative bias was found in all inversions, but M3 estimates agreed best with the observations. Nevertheless, large uncertainty remains in the estimates, so

further information is needed to better quantify emissions in this region.

The estimates for the south American tropical region are very similar to each other (Table S1). All posterior estimates are close to the prior, and the uncertainty in the posterior was hardly reduced compared to the prior. This is because there were no observations assimilated within the region. Three stations (MEX, KEY, RPB) near the edge of the region were included

in the assimilation, but due to strong vertical transport these observations do not capture signals from tropical wetlands, which is the main methane source from this region. Moreover, after filtering, mostly those observations of well-mixed air samples that represents large volume of atmosphere. Therefore, the inversions could not constrain the emissions in this region well.



### 3.4.5 Africa and southern mid-latitudes

In this section, we present results for the following regions: south American temperate region (mTC4), north Africa (mTC5), south Africa (mTC6) and Australia (mTC10).

Posterior total emissions in the south American temperate region increased during 2006-2009 in all inversions (Fig. 13), which does not correspond to a decreases in other regions, e.g. the Asian temperate region. All inversions point in the same direction, but the results are still questionable. Observations assimilated within the region before 2006 are Tierra del Fuego (TDF) and Ushuaia (USH) in Argentina. Due to their locations (latitude 54°S) and having little local emission sources, the purpose of the sites are to sample well-mixed air that represents large volumes of atmosphere. Observations at Arembepe, Brazil (ABP) station were available during 2006-2009, and from Natal, Brazil (NAT) during 2010-2012. These sites capture the well-mixed air in the tropics better than TDF and USH, although most of the signals are from the Atlantic Ocean and not from the land. Interannual variability in the tropics is probably better represented with ABP and NAT observations assimilated, but whether it is of the south American tropics is questionable. Since the observations are located near the region, it is attractive for the inversion to change the emission estimates over land, where emissions are higher than the ocean. Similar interannual variability was reported by Bruhwiler *et al.* (2014), where ABP observations were assimilated (NAT observations were out of their study period), although the trend was not as significant as in this study. The increase in average total emission estimates from B2007 to A2007 is significant and the estimates agree well in all inversions in this region. Also all inversions showed significant increase in average emission estimates for both anthropogenic and biosphere sources, and 12 Tg $CH_4$ $yr^{-1}$ increase in total (Table 4). As the region is mostly within the tropics (30°S-30°N), and most of the emissions are located in the northern part of this region, the estimates agree with Houweling *et al.* (2014) who found most of the increase in the global total emissions in tropics and extra tropics. The increase in emissions during 2005-2008 and the decrease afterwards was also found in Basso *et al.* (2016), although they estimated biospheric emissions from the east part of the Amazon basin, whereas south American tropics is the main contributor to the interannual variability according to our results. In a study by Dlugokencky *et al.* (2011) using isotope observations, they also found the emissions from tropics to be an important contribution to the significant growth in the atmospheric methane after around 2007, although the isotope observations showed a decrease in the isotopic signature, indicating that the increased emission is probably from biogenic sources. The inversions in this study had difficulty changing the ratio of anthropogenic to biospheric emissions from the prior, which could be a reason why the interannual variability of the total emission was optimized by changing the dominant emissions, i.e. anthropogenic. Therefore, interannual variability of the posterior estimates was dominated by the contributions from anthropogenic sources.

All inversions showed larger posterior anthropogenic emissions in north and south Africa compared to the prior, with somewhat different interannual variability in the north and south (Fig. 13). Validation with in-situ observations in north



Africa showed that there is only a small bias in the posterior mole fractions (<1 ppb in M3). Also, the comparison with GOSAT and TCCON XCH$_4$ at Izaña agreed well with the posterior XCH$_4$. Although slight negative bias (about 10 ppb) was found at Izaña, it is probably not due to near-by emissions because negative bias of about 10 ppb was also found elsewhere in the NH. For south Africa, agreement with the in-situ observations is again, good, except for MKT, where a strong negative bias was found (see Section 3.1). The correlation between the posterior and observed mole fractions at MKT is strong (≥0.8), and the site is located at high altitude (>3000m a.s.l.), which implies that the bias may not be due to too small local emission. On the other hand, vertical transport in the tropics is strong, and MKT is the nearest site to the biogenic source region in central Africa. Therefore, the negative bias could also be due to missing emissions from wetlands in central Africa. Bruhwiler *et al.* (2014) also reported an increase in the posterior estimates compared to prior in Africa, but the increase was mainly in the biospheric emissions. Our interannual variability of anthropogenic emissions in north Africa is similar to their variability in central African biospheric emission estimates. This may partly be due to differences in the prior: our prior estimates are higher for anthropogenic emissions and lower for biospheric emissions than those used in Bruhwiler *et al.* (2014). Indeed, the ratios of prior anthropogenic to biospheric emissions in this study and in Bruhwiler *et al.* (2014) are nearly reciprocals of each other. We cannot conclude from this study which is closer to the truth because the region is not well constrained by the observations, i.e. i, in both studies.

Posterior emissions for Australia in M2 were systematically higher than in M1 and M3 throughout 2000-2012. The southern-most coast of Australia and much of New Zealand is defined as 'biosphere' land in M1 and M3 (Fig. 2), i.e. anthropogenic emissions in that area were not optimized in M1 an M3. Since the biosphere emissions are minor source and the posterior emissions did not change much from the prior in M2, we may need to revise the land-ecosystem map in this region to be able to optimize anthropogenic emissions better in M1 and M3.

### 3.4.6 Ocean

Prior anthropogenic emissions over ocean are mainly located in Tropical Ocean (mTC20), and the main differences between prior and posterior emissions are also located in this region (Fig. S5). All inversions showed 5-10 Tg CH$_4$ yr$^{-1}$ higher estimates in the posterior compared the prior, especially before 2006 and during 2011-2012 (Fig. 14). However, it is questionable whether the results are reasonable, since there is no indication that the non-road transportation and coastal anthropogenic emissions estimates varied interannually as the inversion results show. It is more likely that the ocean mTC regions were used to compensate for missing tropical land emissions. Indeed, the estimates for ocean were sensitive to the estimates in other regions (not shown). Further investigation without optimizing anthropogenic ocean emissions or using only natural ocean emissions as prior, i.e. excluding non-road transport (ship and aircraft) emissions, would help us to better understand the anthropogenic emission estimates over the ocean. Note that the uncertainty for the prior biospheric emission estimates in our mTC16-20 regions were set very small. Prior biospheric emissions around the cost were not zero, partly due



to differences in the definition of the coast line in our mTC regions and the prior. Only limited information is available about biospheric emissions around coast lines, and it is a minor source, so we did not assume the inversion is able to optimize it well.

## 4. Discussion

### 4.1 Differences between inversions

Interannual variability of emission estimates was often stronger in M2 than in M1 and M3. Differences were mainly seen in the Asian temperate region, where the proportion of biosphere emissions to the total emissions was much higher in M2 than in M1 and M3. Anthropogenic emission estimates for the Asian tropics in M2 showed strong interannual variability, but the biosphere emission estimates in M2 were similar to the M1 and M3 estimates. The ratio of biosphere to anthropogenic emission estimates in the Asian temperate region and Asian tropics changed from year to year. The dominant sources are similar in M1 and M3, but sometimes different in M2. For example, in the Asian temperate region, biosphere emissions were higher than anthropogenic emissions during 2003-2005 in M1 and M3, but lower in M2. This confirms the findings in our previous study (Tsuruta *et al.*, 2016) that the effect of the land-ecosystem map is strong in that region. Only small differences were found in the posterior values of $XCH_4$ in M1 and M2. However, agreement with in-situ $CH_4$ observations was better in M2 than in M1, i.e. the negative bias in the SH was less pronounced in M2. The emission estimates in the SH are often higher in M2 than in M1, where the differences were mainly seen in the anthropogenic emission estimates. This means that the land-ecosystem distribution used in this study generally represents the division of the source regions well, although some revision may be needed for Asia and SH regions e.g. Australia.

The interannual variability of M1 and M3 emissions were similar, as expected. This confirms that differences between the convection schemes do not have a large effect on the interannual variability of the emission estimates. The north-south gradient of emissions in M3 was different from that in M1 and M2, as expected. M3 emissions are about 10 Tg $CH_4$ $yr^{-1}$ higher in the NH, and about 10 Tg $CH_4$ $yr^{-1}$ lower in the SH compared to the M1 and M2 estimates (Table 4, Table S1). In all regions, estimates of emissions from the dominant sources (either biosphere or anthropogenic) were more strongly affected by the differences in convection schemes than the estimates of another source (M1 and M3). For the set-up in M2, we did not assess the effects of the convection schemes, but a similar result could be expected (Tsuruta *et al.*, 2016). Although the emission estimates for the SH were smaller in M3 than in M1, the SH posterior surface mole fractions and $XCH_4$ were higher in M3 than in M1, due to faster mixing and higher emission estimates in the NH. Agreement with observations was best in M3 among the inversions. The NH surface mole fractions in M3 were in good agreement with observations at in-situ stations, and M3 agreed best with the TCCON $XCH_4$ globally. However, compared to GOSAT retrievals, the $XCH_4$ from M3 posterior estimates were too high. This suggests that CTE-$CH_4$ performs better using Gregory *et al.* (2000) convection scheme (M3) than Tiedtke (1989) convection scheme (M1 and M2) in TM5. It could be assumed that if GOSAT retrievals




were assimilated in CTE-CH$_4$, the emission estimates would decrease in the NH and increase in the SH compared to this study, probably in all inversions. Also, assimilating satellite-based observations may reduce differences in the estimates between the M1 and M3 set-ups. However, previous studies (Houweling *et al.*, 2014; Pandey *et al.*, 2016; Bergamaschi *et al.*, 2013) have shown that the biases in the GOSAT XCH$_4$ products could mislead the distribution and seasonal cycle of the optimized surface emissions.

## 4.2 Uncertainties in emission estimates

CTE-CH$_4$ inversions successfully reduced uncertainties in the emission estimates in all regions. The smallest uncertainties in the posterior annual total emissions were generally seen in M1, and the largest in M2. It was expected that M2 would have larger uncertainties, as the prior uncertainties are larger in M2. The prior uncertainties in M2 are the sum of both anthropogenic and biosphere emissions for each region, whereas the uncertainty in M1 and M3 is from either emission sources. Although the differences were small (<0.1%), uncertainties of the emission estimates in M3 are slightly larger than those in M1 in most of the regions and years for anthropogenic, biosphere and total emissions. In a previous study (Tsuruta *et al.*, 2016), we found that the posterior uncertainties were larger when using the Gregory *et al.* (2000) convection scheme than the Tiedtke (1989) in TM5. This study with long-term simulations confirmed that the uncertainty estimates were often higher in M3 compared to M1. It could be because there is more mixing of the surface signals in Gregory *et al.* (2000), producing a wider range of ensemble mole fractions, and thus M3 have less flux sensitivity at surface sites. The number of observations assimilated did not explain this. The uncertainty was largest in M3, while the number of assimilated observations was also the largest in M3. The anthropogenic emission uncertainty estimates and their reduction (1-$\sigma2\_posterior/\sigma2\_prior$) for the Eurasian boreal region were larger than for north east Europe, which cannot also be explained purely by the number of observations within the region.

The annual uncertainty reduction was generally higher for the major emission source than the minor source. For most of the mTC regions, anthropogenic emission estimates were larger than biosphere emission estimates, and the uncertainty reduction was also larger for anthropogenic emissions as a whole, and the uncertainty reduction was equally high for anthropogenic and biospheric emission estimates (M1, M3). Especially for north east Europe, the uncertainty reduction rates were slightly higher for biosphere emission estimates, although the anthropogenic emission estimates were higher than the biosphere emissions. This is partly the effect of the land-ecosystem map. Much of north east Europe is defined as 'biosphere' land, i.e. inversions M1 and M3 can constrain the biosphere estimates more than the anthropogenic estimates. Uncertainty reduction in M2 is not affected by the land-ecosystem map. Uncertainty reduction for biospheric and anthropogenic emission estimates in M2 were similar in north east Europe because the estimates are not affected by the land-ecosystem map. Although the posterior uncertainties were generally the largest in the M2 estimates, uncertainty reduction for annual emissions is generally largest in the M2 estimates.





Emissions in the Eurasian boreal region are difficult to constrain because of the sparse observation network. The only observation site within the region used in this study is Tiksi (Russia), where observations started in 2010. Although Tiksi is a good reference site for biosphere signals during summer and autumn, one station is not enough to constrain the emissions in the whole Eurasian boreal region. Additional observations from e.g. the National Institute for Environmental Studies (NIES) tall tower network (Sasakawa *et al.*, 2012) and Zotino Tall Tower Observatory (ZOTTO) (Winderlich *et al.*, 2010), would be useful to better understand the emissions from this region. In future studies, we will include those observations. In this study, due to limited observations, the Eurasian boreal region has an extra degree of freedom. Nevertheless, the uncertainties for anthropogenic emissions were reduced by about 20% probably due to some influence of observations located near-by.

The covariance structure of the posterior estimates was similar to the prior in all inversions. This could mean that the assumption in the prior covariance is good, or the inversions are not able to change much from the prior due to e.g. too small prior variation or luck of observations. For regions such as south American tropics, M1 and M3 have strong prior correlation, but M2 has no correlation. Since the posterior correlations were similar to the prior in all inversions, i.e. M1 and M3 posterior have strong correlation, whereas M2 has nearly zero correlation, where the inversions could not optimize the dependencies well. On the other hand, the inversions showed similar posterior correlations regardless of the prior assumption in anthropogenic emissions between anthropogenic and water regions in the Asian temperate region. M1 and M3 had prior correlation of about 0.5, but the correlation was reduced to less than 0.1. M2 had prior correlation of zero, and the posterior correlation did not increase significantly, supporting the M1 and M3 posterior correlation. This suggests that the prior correlation for those regions in M1 and M3 was probably too strong. In the prior covariance, no negative correlation was assumed between any scaling factors. However, some regions showed weak but negative correlation in the posterior estimates. For example, anthropogenic emission in the Asian temperate region and Atlantic Ocean showed negative correlation in all inversions, which is one of the reasons why the ocean emissions were sensitive to the estimates of other regions (see Section 3.4.6). The inversions did not turn positive correlation to negative correlation.

## 5. Summary and Conclusions

We presented global methane emission estimates for 2000-2012 estimated using the CarbonTracker Europe-$CH_4$ (CTE-$CH_4$) data assimilation system. The estimates were evaluated against assimilated in-situ atmospheric $CH_4$ mole fraction observations and model-independent atmospheric measurements from aircraft campaigns, as well as $XCH_4$ observations from the TCCON and GOSAT. Three inversions were performed to evaluate the effect of simultaneously optimising two scaling factors per region, and the choice of convection scheme used in the TM5 atmospheric chemistry transport model: CTE-$CH_4$ v1.0 with Tiedtke (1989) convection scheme, CTE-$CH_4$ v1.1 with Tiedtke (1989) convection scheme, and CTE-$CH_4$ v1.0 with Gregory *et al.* (2000) convection scheme. The two versions of CTE-$CH_4$ differs by the number of scaling factors which depends whether to optimize either biospheric or anthropogenic emissions (v1.0) or both (v1.1) per region.



Our inversions indicate that global total posterior emissions for 2000-2012 were 515-517±44-62 Tg $CH_4$ $yr^{-1}$, and increased by about 18-19 Tg $CH_4$ $yr^{-1}$ from 2001-2006 to 2007-2012. The increase was mainly driven by the emissions in the modified TransCom (mTC) regions of the south American temperate region, the Asian temperate region and Asian tropics. This estimated growth in the posterior global total emissions was more than 10 Tg $CH_4$ $yr^{-1}$ smaller than the growth in the prior, which was based on anthropogenic emission estimates from EDGAR v4.2 FT2010, biospheric emission estimates from LPX-Bern v1.0, biomass-burning estimates from GFED v3.1, termite emission estimates from Ito and Inatomi (2012), and ocean emission estimates from Tsuruta *et al.* (2016). The inversions suggest that most of the increase was in anthropogenic rather than biospheric emission estimates. However, it must be noted that we could not confirm whether the increase in the south American temperate region, the Asian temperate region and Asian tropics are in anthropogenic or biosphere estimates from this study. The inversion has a tendency to optimize the major source in a region rather than the minor source, and anthropogenic emission estimates were often dominant over biospheric emissions in those regions. Further investigations with other prior emissions of different spatial distribution or isotope observations could be useful in the future studies to better distinguish between microbial, thermogenic and pyrogenic emission sources. Furthermore, the results showed that posterior emissions are generally lower than the prior emissions in the high northern hemisphere (north American boreal region, Europe and Eurasian boreal region), whereas the posterior emissions are higher than the prior emissions in Africa and the Southern Hemisphere (SH) (north Africa, south Africa, south American temperate and Australia). For the Tropics (south American Tropics and Asian Tropics), posterior emissions are similar or slightly lower than the prior emissions. This was consistent in all inversions. The inversion results suggest that the distribution in the prior emissions, probably of anthropogenic sources, may need to be revised with less emissions in the mid-latitude NH and more emissions in the temperate regions in the SH.

The study specifically focused on Europe by dividing it into 4 mTC regions: south east, south west, north east, and north west Europe. Neither prior nor posterior emissions showed significant trend in anthropogenic or biospheric emission estimates in Europe as a whole. However, the posterior anthropogenic emissions were higher than the estimates in EDGAR v4.2 FT2010 inventory for southern Europe, while it was lower in northern Europe. Also, the posterior biospheric emission estimates have different interannual variability than those from the LPX-Bern vegetation model, such that CTE-$CH_4$ estimates agreed better with methane emissions measured at some wetland sites. Furthermore, applying different scaling factors to regions divided by land-ecosystem type was an improvement. Posterior emissions in inversions v1.0 and v1.1 were similar such that regardless of whether we were optimizing only anthropogenic, only biospheric, or both per region, total emissions were similar and the ratio of anthropogenic to biospheric estimates did not change much from the prior. However, v1.0 was found to be more consistent with observations, and it produced more reasonable estimates in the Asian temperate region and Asian tropics, but v1.1 was better where both anthropogenic and biospheric emissions are high and the land-ecosystem map is badly defined, such as Australia. This approach could be useful to better understand the dependence of methane emissions on meteorological parameters for different ecosystem types, and we will continue developing it further.



Evaluations with in-situ observations showed that the inversions successfully reduced bias between observed and estimated mole fractions from the prior to the posterior. The comparison with model-independent observations of $XCH_4$ from TCCON and GOSAT showed that posterior $XCH_4$ agrees better with observations than the prior. Agreement in posterior $XCH_4$ is especially good in the NH. However, negative biases were found in the SH for both $XCH_4$ comparisons in all inversions, although the seasonal cycle at the TCCON sites was well captured. This suggests that there are some missing emissions that are not optimized well by CTE-CH$_4$, although we cannot ignore the possible effects of the vertical distribution or influence of stratospheric air with low CH$_4$ defined in the transport model. The evaluation also revealed that TM5 with the Gregory *et al.* (2000) convection scheme produces higher emission estimates in the NH and lower emissions in the SH compared to when the Tiedtke (1989) convection scheme. With the Gregory *et al.* (2000) convection scheme, the transport from the NH to the SH is faster, leading to smaller inferred SH emissions and larger NH emissions. This means that the changes in the emission estimates from the prior were smaller using Gregory *et al.* (2000) in the SH, and higher in the NH, except for the north American boreal region and north east Europe. Also the simulated mole fractions with posterior emissions agree slightly better with observations when using the Gregory *et al.* (2000) convection scheme. Furthermore, the evaluation with GOSAT $XCH_4$ revealed that the spring peaks in $XCH_4$ in the tropics were poorly captured in inversions M1 and M2, in which Tiedtke (1989) convection scheme was used. This feature was best captured in the M3 inversion, in which the Gregory *et al.* (2000) convection scheme was used, and the NH winter time emission estimates were higher than the M1 and M2 inversions.

**Code and data availability**

The source code of CTE-CH$_4$ and data presented in this paper will be provided on request from the corresponding author (Aki Tsuruta: Aki.Tsuruta@fmi.fi).

*Acknowledgements.* We thank the Nessling foundation, NCoE DEFROST, NCoE eSTICC and the Finnish Academy project CARB-ARC (285630) for their financial support. We thank Dr. Akihiko Ito for providing prior emissions of termites, and Dr. Lori Bruhwiler for the valuable discussion that greatly assisted this work. We are grateful for Swiss Federal Laboratories for Materials Science and Technology (EMPA), Environment Canada (EC), Meteorological Research Institute (MRI), Laboratoire des Sciences du Climat et de l'Environnement (LSCE), the National Institute of Water and Atmospheric Research Ltd. (NIWA), the Environment Division Global Environment and Marine Department Japan Meteorological Agency (JMA), National Institute for Environmental Studies (NIES), Umweltbundesamt Germany/Federal Environmental Agency (UBA), Umweltbundesamt Austria/Environment Agency Austria (EAA) as the data provider for Sonnblick, the South African Weather Service (SAWS), the Main Geophysical Observatory (MGO), the Korea Meteorological Administration (KMA), Meteorology, Climatology, and Geophysics Agency Indonesia (BMKG), University of Bristol (UNIVBRIS), National Institute of Environmental Research (NIER), Centre for Environmental Monitoring (RIVM) for performing high-quality CH$_4$




measurements at global sites and making them available through the GAW-WDCGG. The in situ methane measurements at Lauder, Baring Head, and Arrival Heights are conducted as part of NIWA's government-funded, core research from New Zealand's ministry of Business, Innovation and Employment. The observations by JMA is a part of the GAW program of the WMO. For aircraft measurements of the IMECC project we acknowledge the support of the European Commission within the 6th Framework Program through the Integrated Infrastructure Initiative IMECC (Infrastructure for Measurement of the European Carbon Cycle), and the Max Planck Society for funding additional flight hours onboard the Lear Jet. For regular aircraft measurements at Bialystok we thank the gas lab at the Max Planck Institute for Biogeochemistry at Jena for analysis of the flask samples, and the Max Planck Society for funding. We also acknowledge California Institute of Technology, University of Wollongong, Institute of Environmental Physics, University of Bremen, NIWA, NIES, Karlsruhe Institute of Technology, IMK-IFU, Japan Aerospace Exploration Agency (JAXA), Los Alamos National Laboratory, University of Toronto, NASA Ames Research Center, Max Planck Institute for Biogeochemistry for their XCH4 retrievals. This work was also supported by EU-FP7 InGOS project (no. 284274), ICOS Carbon Portal (ICOS-ERIC, no. 281250) and Academy of Finland Center of Excellence (no. 272041).

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



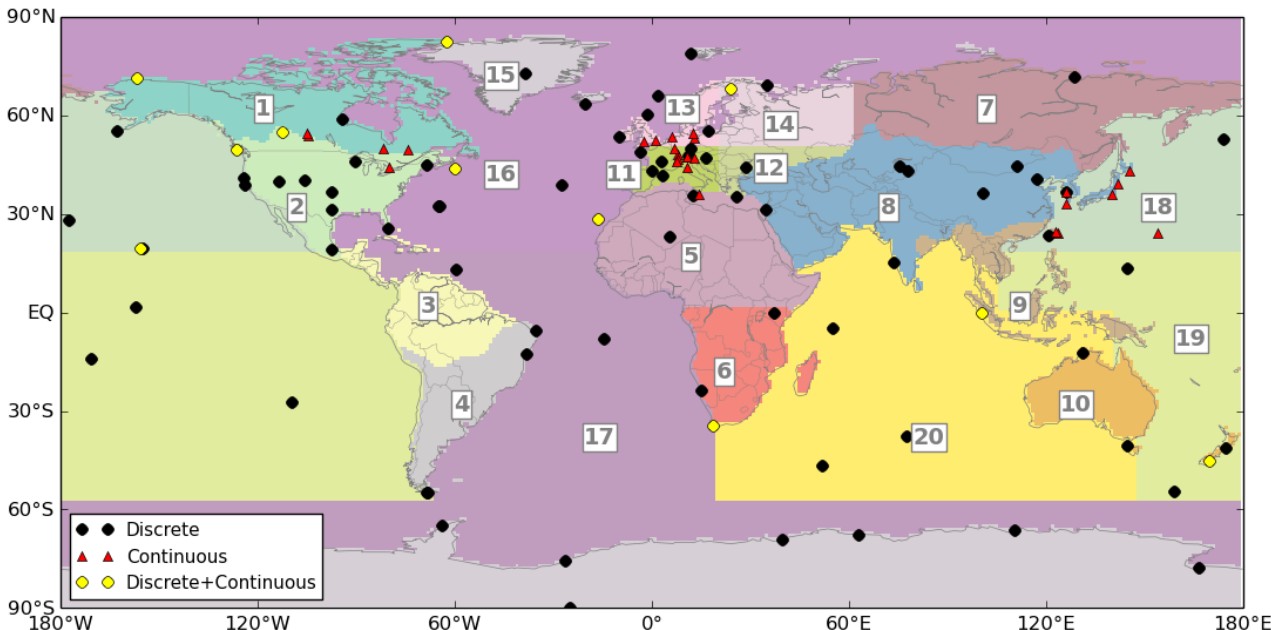

**Figure 1.** Modified TransCom (mTC) regions illustrated in numbers and colours and locations of sites with observations assimilated in the inversions.





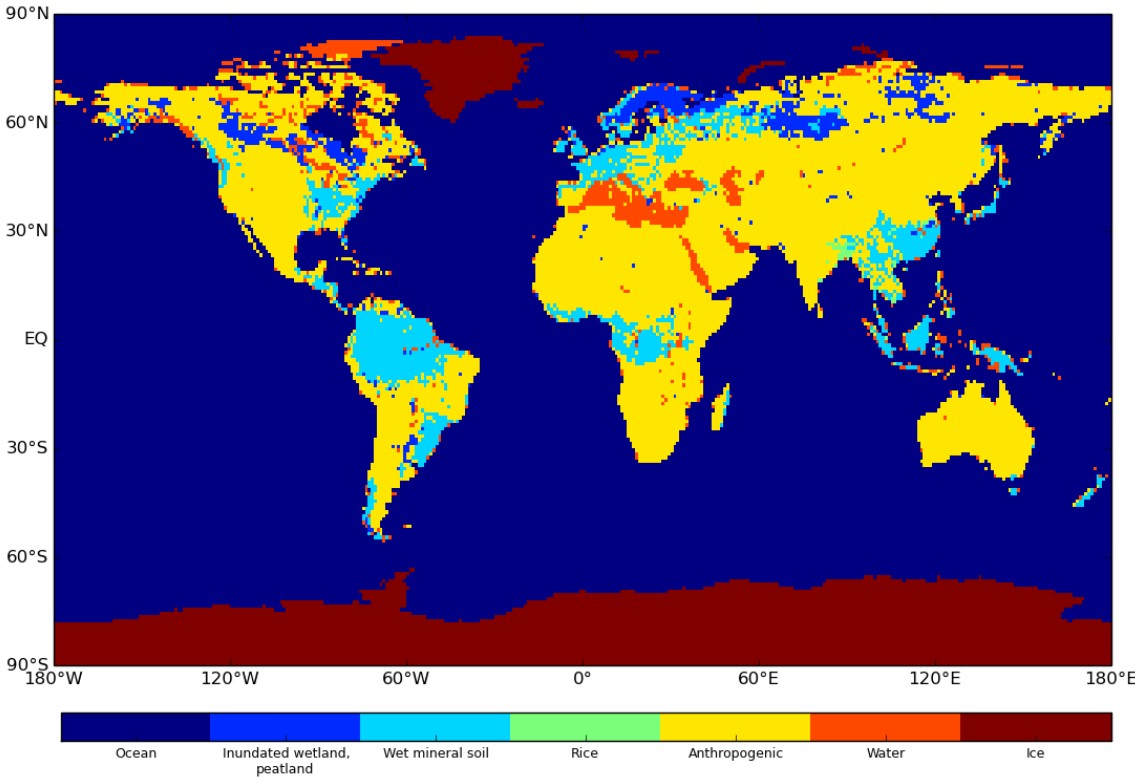

**Figure 2.** Land-ecosystem map used as regional definition in the optimisation.





**Figure 3.** Differences between the estimated and observed mole fractions from the inversions M1, M2 and M3. For comparisons with the prior, all observations were used, while for posterior only assimilated observations were shown.





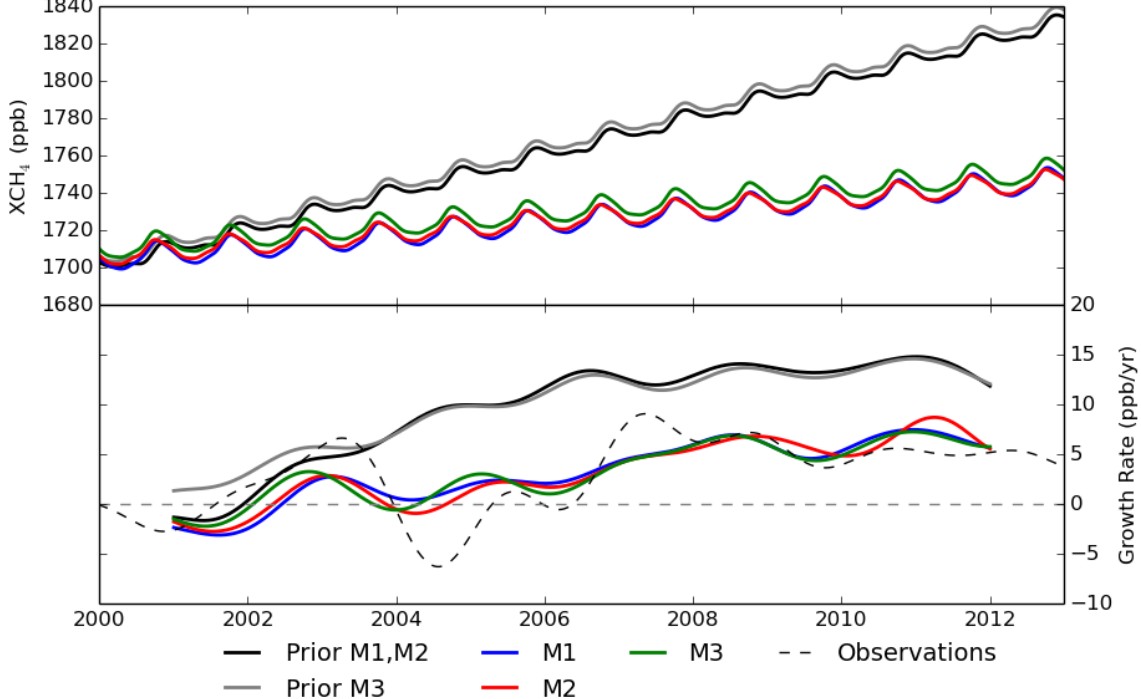

**Figure 4.** Smoothed global mean total column atmospheric mole fractions (XCH₄) and their growth rates from inversions M1, M2 and M3. The growth rate of the observations are computed from NOAA marine boundary later observations. The growth rate was calculated using the methods by Thoning *et al.* (1989).







**Figure 5.** Vertical profiles of atmospheric methane mole fraction from aircraft and posterior estimates. For each site, the medians were calculated and plotted for both observations and posterior estimates for each altitude band.



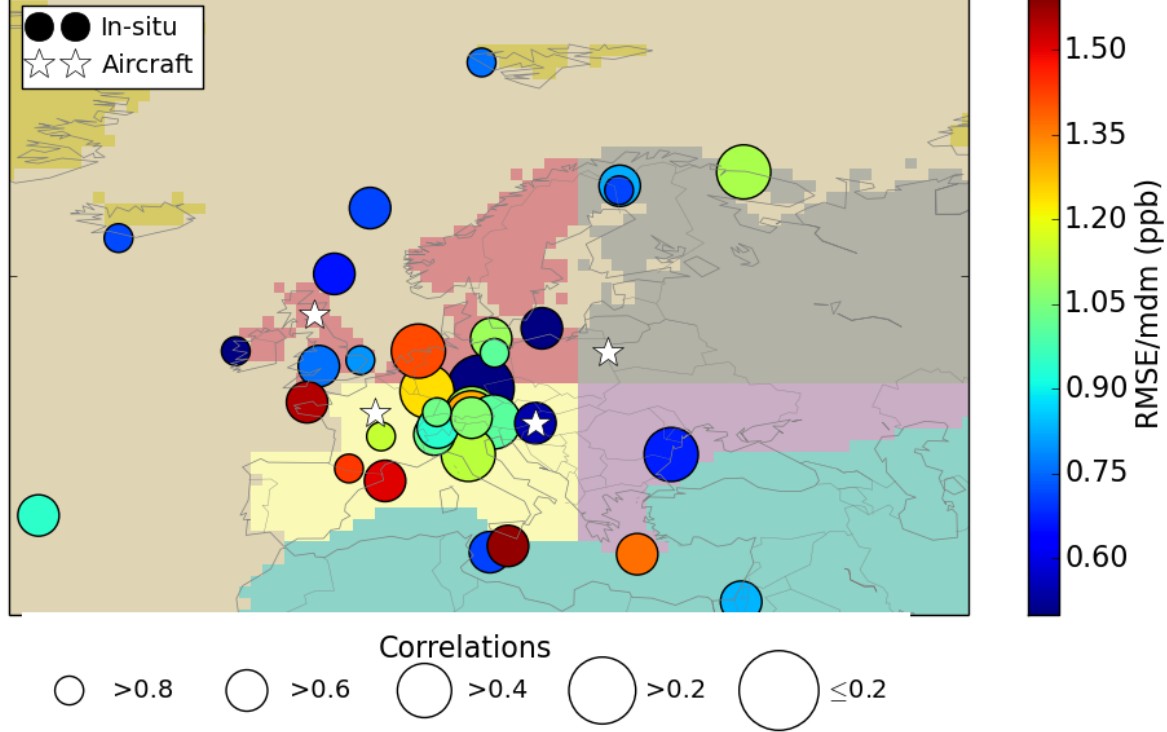

**Figure 6.** Performance of inversion M3 at European in-situ observation sites, whose data were assimilated in the model, and at the location of four aircraft campaigns. The campaign locations are marked with stars. Aircraft observations were used for validation. The colour of the marker for the in-situ observation site is determined by the RMSE of the observations and posterior mole fractions divided by the pre-defined mdm. The radius of each circle provides the correlation between observations and posterior mole fractions, where a larger radius corresponds to a smaller correlation. Background colours identify the mTC regions.





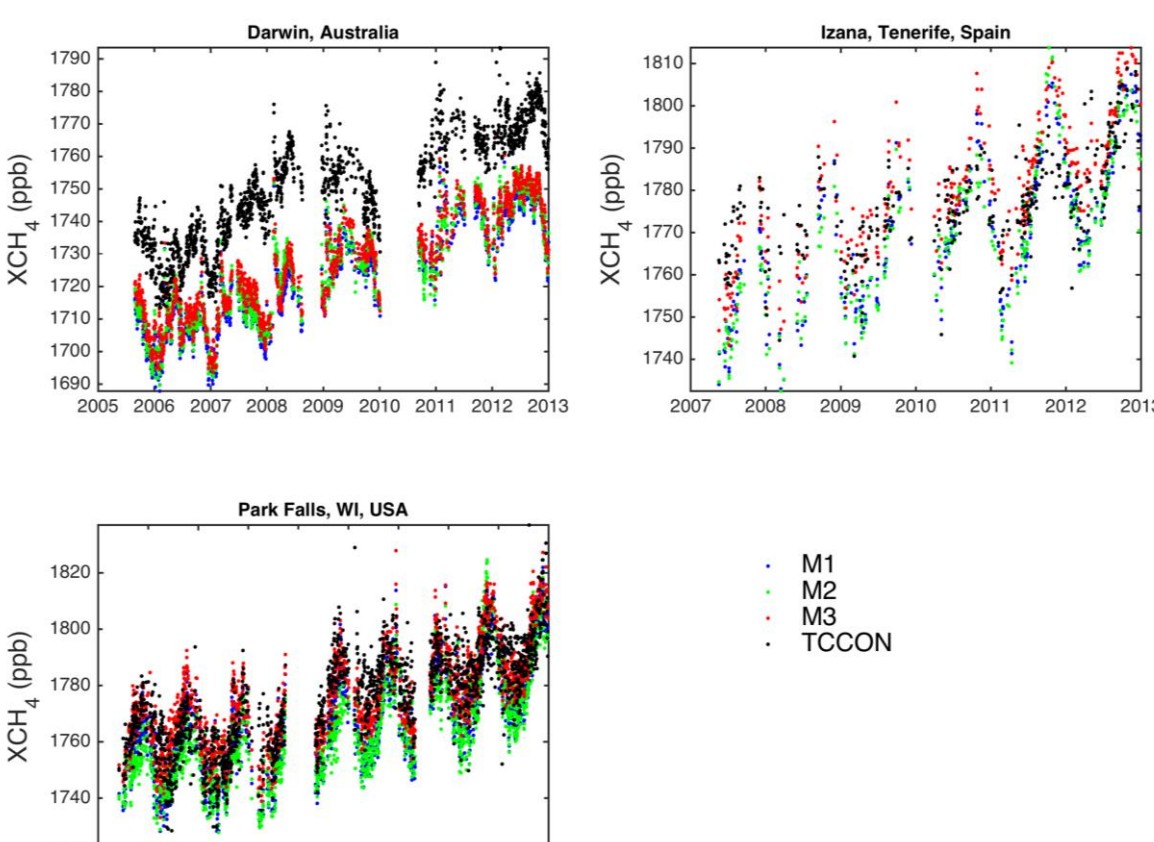

**Figure 7.** Observed and estimated daily mean XCH4 at TCCON sites.



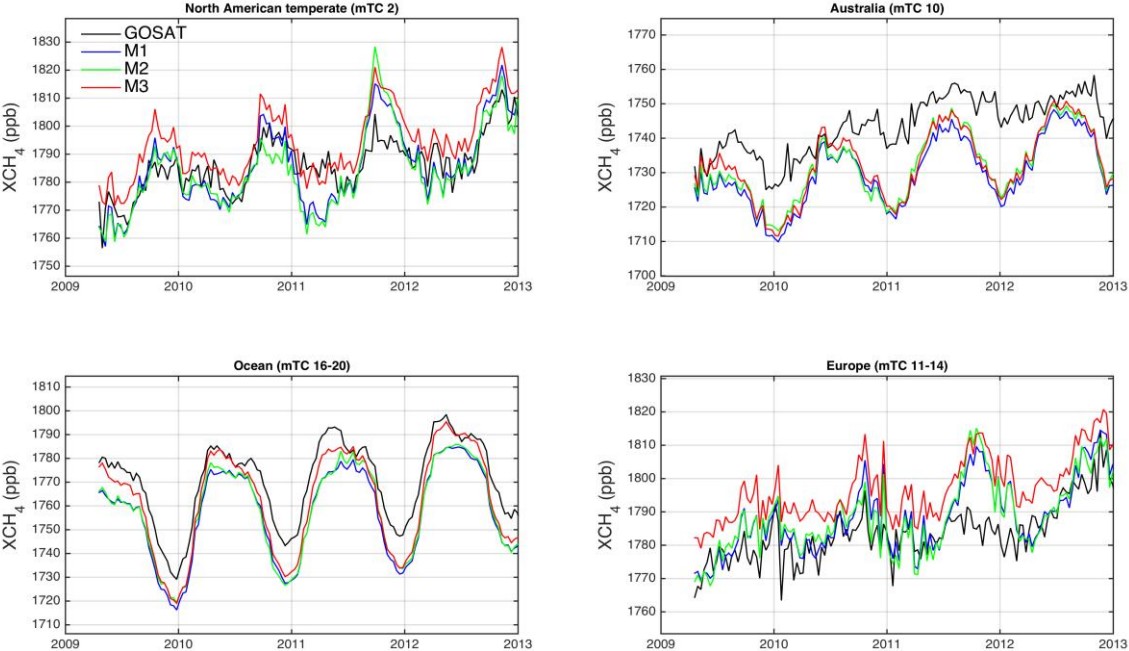

**Figure 8.** GOSAT and estimated regional 10-day mean XCH$_4$ for the mTC regions.




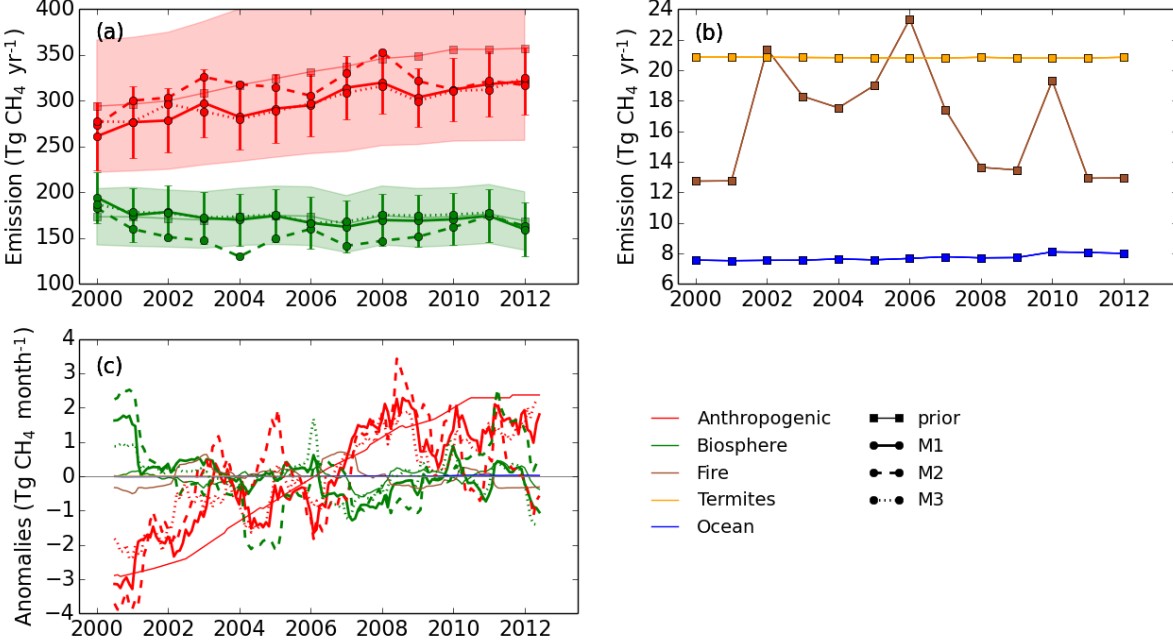

**Figure 9.** (a), (b) Global annual emission estimates by source category. Shaded areas are the prior uncertainties, and vertical bars illustrate M1 posterior uncertainties. Note that ocean emissions are only from natural sources, i.e. anthropogenic emissions over the ocean are included in anthropogenic emission. (c) The anomalies of the emission estimates. 12-month moving averages from monthly means were calculated and plotted. The zero level shown by a grey line is the mean of the moving averages during 2000-2012.




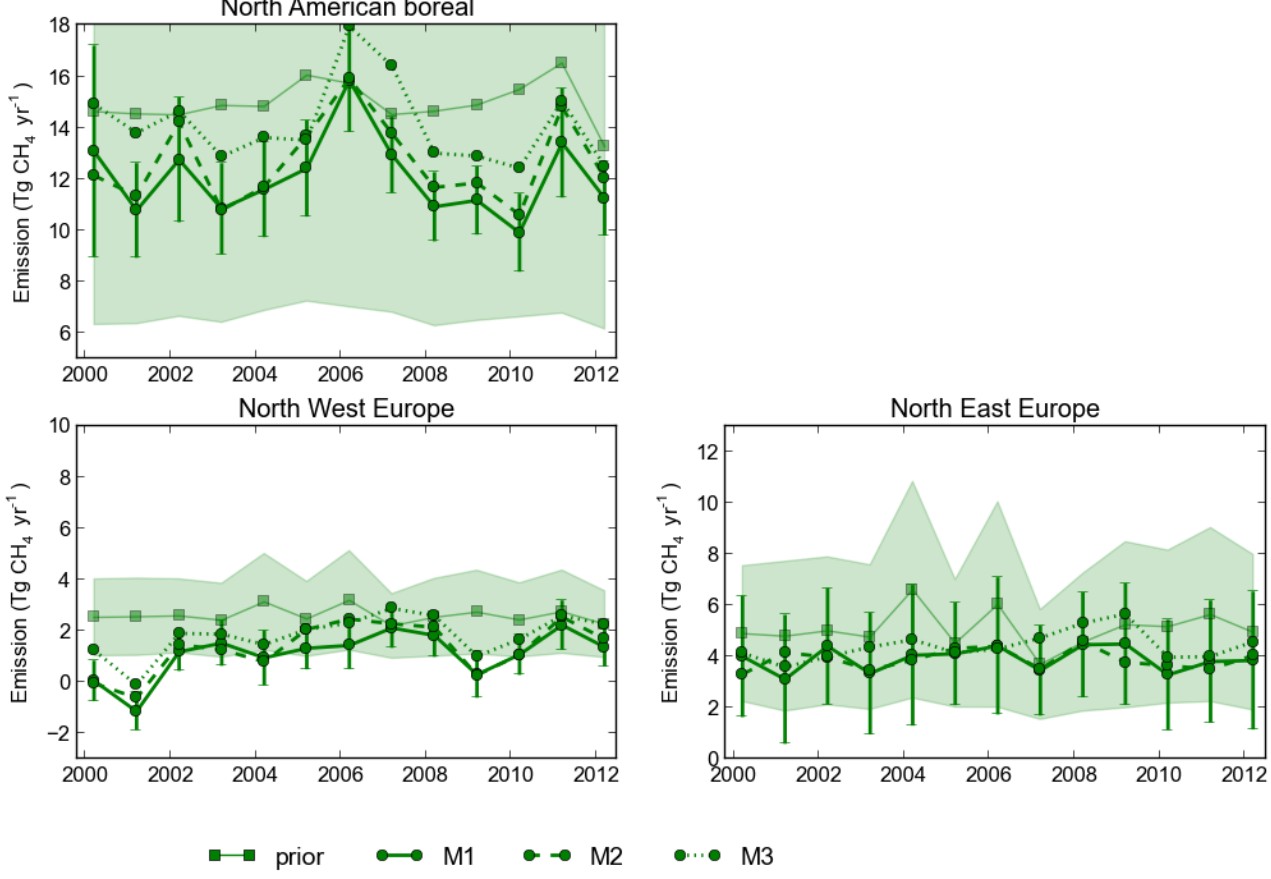

**Figure 10.** Prior and posterior regional annual biospheric emission estimates for the northern Boreal regions. Shaded areas are prior uncertainties, and vertical bars illustrate M1 posterior uncertainties. Note different ranges on the y-axis.




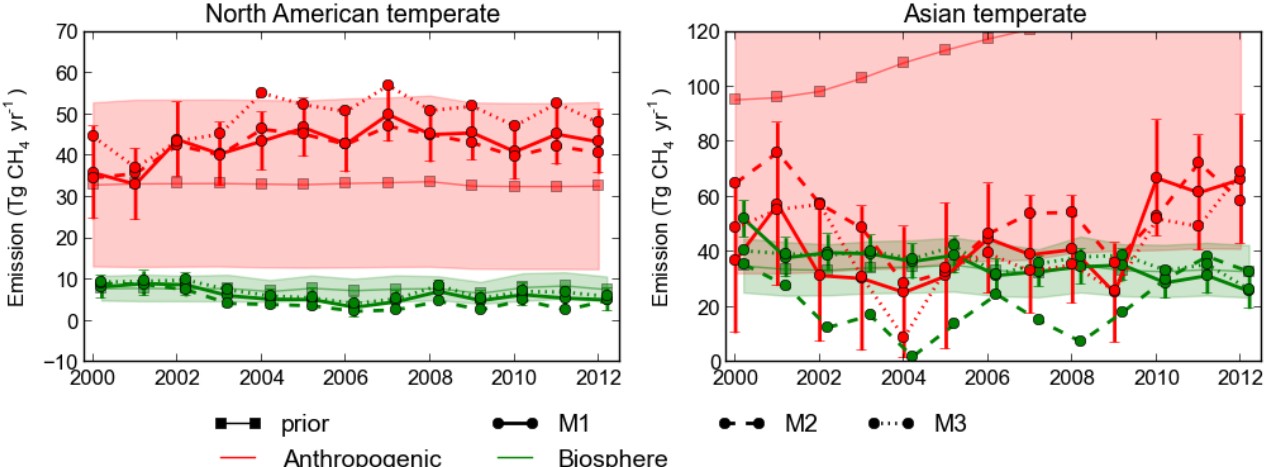

**Figure 11.** Prior and posterior regional annual emission estimates for the north American temperate and Asian temperate regions. Shaded areas are prior uncertainties, and vertical bars illustrate M1 posterior uncertainties. Note different ranges on the y-axis.

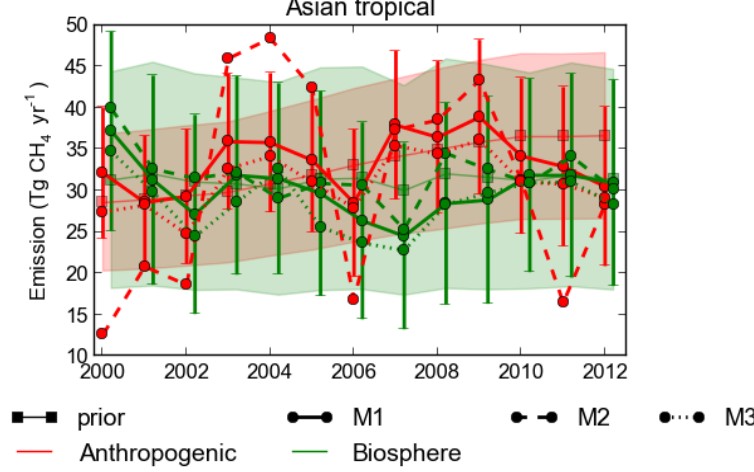

**Figure 12.** Prior and posterior regional annual emission estimates for Asian tropics. Shaded areas are prior uncertainties, and vertical bars illustrate M1 posterior uncertainties.




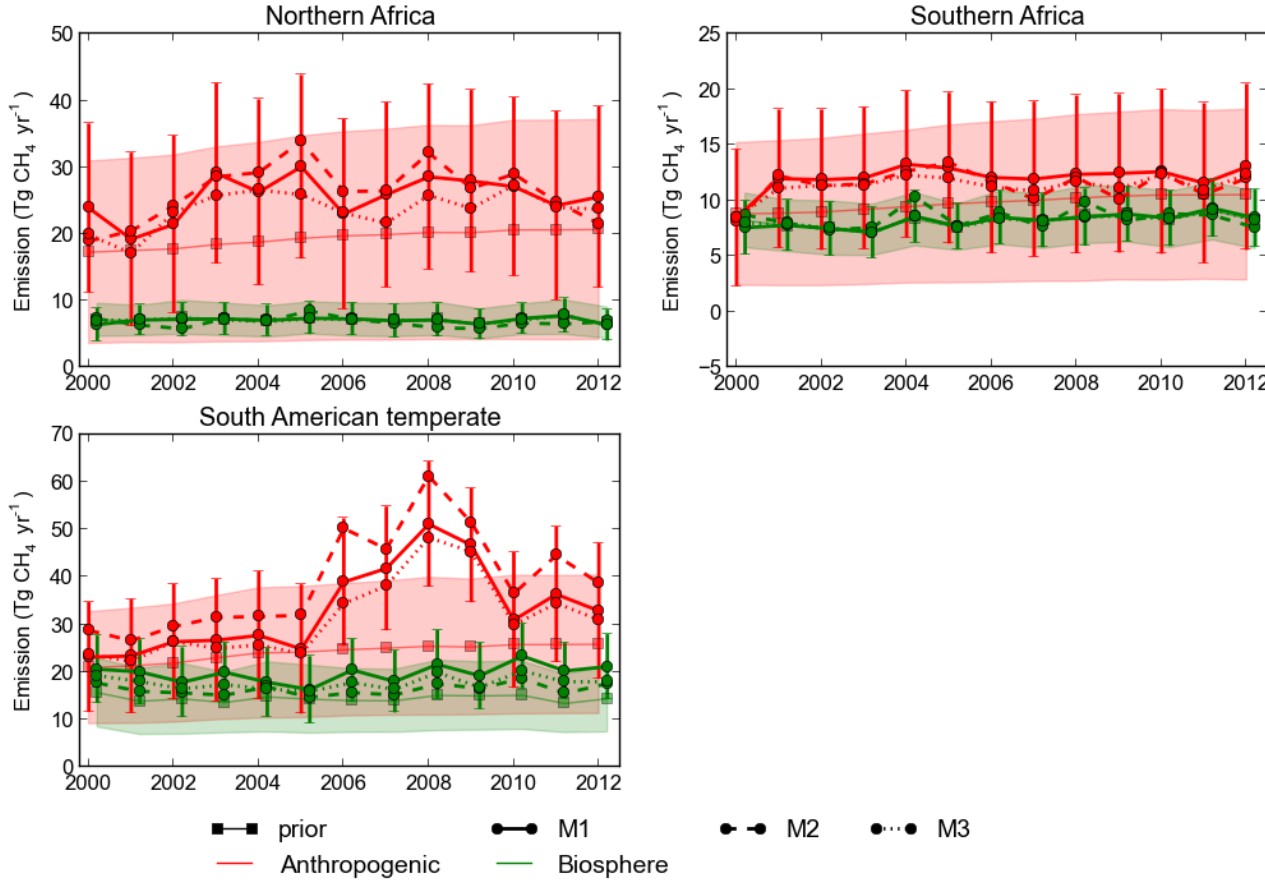

**Figure 13.** Prior and posterior regional annual emission estimates for north Africa, south Africa and south American temperate region. Shaded areas are prior uncertainties, and vertical bars illustrate M1 posterior uncertainties. Note different ranges on the y-axis.





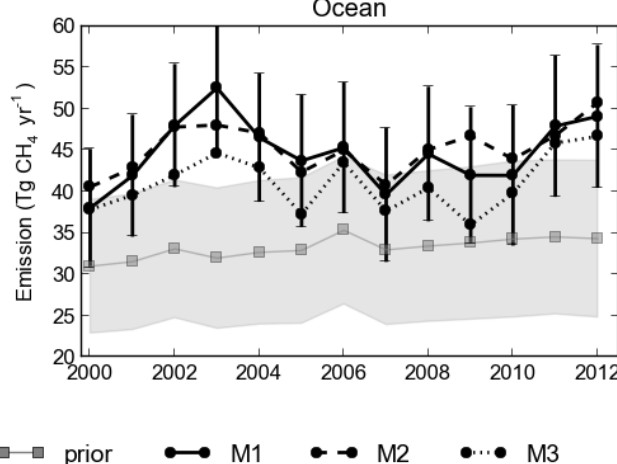

**Figure 14.** Prior and posterior regional annual total emission estimates for global ocean. Shaded areas are prior uncertainties, and vertical bars illustrate M1 posterior uncertainties. Note different ranges on the y-axis. Note that anthropogenic and biospheric sources are included as well as natural ocean emissions.





**Table 1.** List of inversion set-ups.

| Inversion | CTE-CH$_4$ | TM5 convection |
|---|---|---|
| M1 | v1.0 | Tiedtke (1989) |
| M2 | v1.1 | Tiedtke (1989) |
| M3 | v1.0 | Gregory *et al.* (2000) |

**Table 2.** List of surface in-situ observation sites used in inversions. Model-data-mismatch (mdm) is used in the observation covariance matrix, and defining rejection threshold of the observations. Data type is categorized into two: discrete (D) and continuous (C) measurements.

| Site Code | Station Name | Country/Territory | Contributor | Latitude | Longitude | Elevation (m a.s.l.) | mdm (ppb) | Data type (d/c) |
|---|---|---|---|---|---|---|---|---|
| ABP | Arembepe | Brazil | NOAA/ESRL | 12.77°S | 38.17°W | 1 | 4.5 | D |
| ALT | Alert | Canada | NOAA/ESRL | 82.45°N | 62.52°W | 210 | 15.0 | D |
| ALT | Alert | Canada | EC | 82.45°N | 62.52°W | 210 | 15.0 | C |
| AMS | Amsterdam Island | France | LSCE | 37.8°S | 77.53°E | 55 | 4.5 | D |
| AMT | Argyle | USA | NOAA/ESRL | 45.03°N | 68.68°W | 53 | 30.0 | D |
| AMY | Anmyeon-do | Republic of Korea | KMA | 36.53°N | 126.32°E | 86 | 15.0 | C |
| ARH | Arrival Heights | New Zealand | NIWA | 77.80°S | 166.67°E | 189 | 4.5 | D |
| ASC | Ascension Island | UK | NOAA/ESRL | 7.92°S | 14.42°W | 54 | 4.5 | D |
| ASK | Assekrem | Algeria | NOAA/ESRL | 23.18°N | 5.42°E | 2728 | 25.0 | D |
| AZR | Terceira Island | Portugal | NOAA/ESRL | 38.77°N | 27.38°W | 40 | 15.0 | D |
| BAL | Baltic Sea | Poland | NOAA/ESRL | 55.35°N | 17.22°E | 28 | 75.0 | D |
| BGU | Begur | Spain | LSCE | 41.83°N | 3.33°E | 30 | 15.0 | D |
| BHD | Baring Head | New Zealand | NOAA/ESRL | 41.41°S | 174.87°E | 80 | 4.5 | D |
| BKT | Bukit Koto Tabang | Indonesia | NOAA/ESRL | 0.20°S | 100.32°E | 865 | 75.0 | D |
| BKT | Bukit Koto Tabang | Indonesia | BMG_EMPA | 0.20°S | 100.32°E | 896.5 | 75.0 | C |
| BME | St. David's Head | UK | NOAA/ESRL | 32.37°N | 64.65°W | 30 | 15.0 | D |
| BMW | Tudor Hill | UK | NOAA/ESRL | 32.27°N | 64.88°W | 30 | 15.0 | D |
| BRW | Barrow | USA | NOAA/ESRL | 71.32°N | 156.60°W | 11 | 15.0 | C |
| BRW | Barrow | USA | NOAA/ESRL | 71.32°N | 156.60°W | 11 | 15.0 | D |
| BSC | Black Sea | Romania | NOAA/ESRL | 44.17°N | 28.68°E | 3 | 75.0 | D |
| CBA | Cold Bay | USA | NOAA/ESRL | 55.20°N | 162.72°W | 25 | 15.0 | D |
| CDL | Candle Lake | Canada | EC | 53.87°N | 104.65°W | 630 | 25.0 | C |
| CGO | Cape Grim | Australia | NOAA/ESRL | 40.68°S | 144.68°E | 94 | 4.5 | D |
| CHL | Churchill | Canada | EC | 58.75°N | 94.07°W | 76 | 15.0 | D |
| CHM | Chibougamau | Canada | EC | 49.68°N | 74.34°W | 393 | 15.0 | C |
| CHR | Christmas Island | Kiribati | NOAA/ESRL | 1.70°N | 157.17°W | 3 | 4.5 | D |
| CMN | Monte Cimone | Italy | UNIURB/ISAC | 44.18°N | 10.70°E | 2172 | 15.0 | C |
| COI | Cape Ochi-ishi | Japan | NIES | 43.15°N | 145.50°E | 100 | 4.5 | C |
| CPT | Cape Point | South Africa | NOAA/ESRL | 34.35°S | 18.49°E | 230 | 25.0 | D |





| | | | | | | | | |
|---|---|---|---|---|---|---|---|---|
| CPT | Cape Point | South Africa | SAWS | 34.35°S | 18.49°E | 260 | 15.0 | C |
| CRI | Cape Rama | India | CSIRO | 15.08°N | 73.83°E | 60 | 75.0 | D |
| CRZ | Crozet | France | NOAA/ESRL | 46.45°S | 51.85°E | 120 | 4.5 | D |
| CYA | Casey Station | Australia | CSIRO | 66.28°S | 110.52°E | 2 | 4.5 | D |
| DEU | Deuselbach | Germany | UBA | 49.77°N | 7.05°E | 480 | 15.0 | C |
| EGB | Egbert | Canada | EC | 44.23°N | 79.78°W | 226 | 75.0 | C |
| EIC | Easter Island | Chile | NOAA/ESRL | 27.15°S | 109.45°W | 50 | 4.5 | D |
| ESP | Estevan Point | Canada | CSIRO | 49.38°N | 126.55°W | 39 | 25.0 | D |
| ESP | Estevan Point | Canada | EC | 49.38°N | 126.55°W | 39 | 25.0 | C |
| ETL | East Trout Lake | Canada | EC | 54.35°N | 104.98°W | 492 | 25.0 | C |
| FIK | Finokalia | Greece | LSCE | 35.34°N | 25.67°E | 150 | 15.0 | D |
| FSD | Fraserdale | Canada | EC | 49.88°N | 81.57°W | 210 | 15.0 | C |
| GLH | Giordan Lighthouse | Malta | UMLT | 36.07°N | 14.22°E | 167 | 15.0 | C |
| GMI | Guam | US Territory | NOAA/ESRL | 13.43°N | 144.78°E | 2 | 15.0 | D |
| GPA | Gunn Point | Australia | CSIRO | 12.25°S | 131.05°E | 37 | 75.0 | D |
| GSN | Gosan | Republic of Korea | GERC | 33.15°N | 126.12°E | 144 | 15.0 | C |
| HAT | Hateruma | Japan | NIES | 24.05°N | 123.80°E | 47 | 15.0 | C |
| HBA | Halley Bay | UK | NOAA/ESRL | 75.58°S | 26.50°W | 30 | 4.5 | D |
| HPB | Hohenpeissenberg | Germany | NOAA/ESRL | 47.80°N | 11.01°E | 985 | 25.0 | D |
| HUN | Hegyhatsal | Hungary | NOAA/ESRL | 46.95°N | 16.65°E | 344 | 75.0 | D |
| ICE | Heimaey | Iceland | NOAA/ESRL | 63.34°N | 20.29°W | 118 | 15.0 | D |
| IZO | Izaña (Tenerife) | Spain | NOAA/ESRL | 28.30°N | 16.48°W | 2360 | 15.0 | D |
| IZO | Izaña (Tenerife) | Spain | AEMET | 28.30°N | 16.48°W | 2360 | 15.0 | C |
| JFJ | Jungfraujoch | Switzerland | EMPA | 46.55°N | 7.99°E | 3583 | 15.0 | C |
| KEY | Key Biscayne | USA | NOAA/ESRL | 25.67°N | 80.20°W | 3 | 25.0 | D |
| KMW | Kollumerwaard | Netherlands | RIVM | 53.33°N | 6.28°E | 0 | 15.0 | C |
| KUM | Cape Kumukahi | USA | NOAA/ESRL | 19.52°N | 154.82°W | 3 | 4.5 | D |
| KZD | Sary Taukum | Kazakhstan | NOAA/ESRL | 44.45°N | 75.57°E | 412 | 75.0 | D |
| KZM | Plateau Assy | Kazakhstan | NOAA/ESRL | 43.25°N | 77.88°E | 2519 | 25.0 | D |
| LAU | Lauder | New Zealand | NIWA | 45.03°S | 169.67°E | 370 | 15.0 | C |
| LAU | Lauder | New Zealand | NIWA | 45.03°S | 169.67°E | 370 | 15.0 | D |
| LEF | Park Falls | USA | NOAA/ESRL | 45.93°N | 90.27°W | 868 | 30.0 | D |
| LLB | Lac La Biche | Canada | NOAA/ESRL | 54.95°N | 112.45°W | 540 | 75.0 | D |
| LLB | Lac La Biche (Alberta) | Canada | EC | 54.95°N | 112.45°W | 540 | 75.0 | C |
| LLN | Lulin | China | NOAA/ESRL | 23.47°N | 120.87°E | 2862 | 25.0 | D |
| LMP | Lampedusa | Italy | NOAA/ESRL | 35.52°N | 12.62°E | 45 | 25.0 | D |
| LPO | Ile Grande | France | LSCE | 48.80°N | 3.58°W | 20 | 15.0 | D |
| MAA | Mawson | Australia | CSIRO | 67.62°S | 62.87°E | 32 | 4.5 | D |
| MEX | Pico de Orizaba | Mexico | NOAA/ESRL | 18.98°N | 97.31°W | 4464 | 15.0 | D |
| MHD | Mace Head | Ireland | NOAA/ESRL | 53.33°N | 9.90°W | 25 | 25.0 | D |
| MID | Sand Island | US Territory | NOAA/ESRL | 28.21°N | 177.38°W | 4 | 15.0 | D |
| MKN | Mt. Kenya | Kenya | NOAA/ESRL | 0.05°S | 37.30°E | 3897 | 25.0 | D |
| MLO | Mauna Loa | USA | NOAA/ESRL | 19.53°N | 155.58°W | 3397 | 15.0 | C |




| MLO | Mauna Loa | USA | NOAA/ESRL | 19.53°N | 155.58°W | 3397 | 15.0 | D |
|-----|-----------|-----|-----------|---------|----------|------|------|---|
| MNM | Minamitorishima | Japan | JMA | 24.30°N | 153.97°E | 8 | 15.0 | C |
| MQA | Macquarie Island | Australia | CSIRO | 54.48°S | 158.97°E | 12 | 4.5 | D |
| NAT | Natal | Brazil | NOAA/ESRL | 5.51°S | 35.26°W | 15 | 15.0 | D |
| NGL | Neuglobsow | Germany | UBA | 53.17°N | 13.03°E | 68.4 | 15.0 | C |
| NMB | Gobabeb | Namibia | NOAA/ESRL | 23.58°S | 15.03°E | 456 | 25.0 | D |
| NWR | Niwot Ridge (T-van) | USA | NOAA/ESRL | 40.05°N | 105.58°W | 3523 | 15.0 | D |
| OXK | Ochsenkopf | Germany | NOAA/ESRL | 50.03°N | 11.80°E | 1009 | 75.0 | D |
| PAL | Pallas-Sammaltunturi | Finland | NOAA/ESRL | 67.97°N | 24.12°E | 560 | 15.0 | D |
| PAL | Pallas-Sammaltunturi | Finland | FMI | 67.58°N | 24.06°E | 572 | 15.0 | C |
| PDM | Pic du Midi | France | LSCE | 42.93°N | 0.13°E | 2877 | 15.0 | D |
| PRS | Plateau Rosa | Italy | RSE | 45.93°N | 7.70°E | 3490 | 15.0 | C |
| PSA | Palmer Station | USA | NOAA/ESRL | 64.92°S | 64.00°W | 10 | 4.5 | D |
| PTA | Point Arena | USA | NOAA/ESRL | 38.95°N | 123.73°W | 17 | 25.0 | D |
| PUY | Puy de Dome | France | LSCE | 45.77°N | 2.97°E | 1465 | 15.0 | D |
| RGL | Ridge Hill | UK | UNIVBRIS | 52.00°N | 2.54°W | 294 | 25.0 | C |
| RPB | Ragged Point | Barbados | NOAA/ESRL | 13.17°N | 59.43°W | 45 | 15.0 | D |
| RYO | Ryori | Japan | JMA | 39.03°N | 141.83°E | 260 | 15.0 | C |
| SDZ | Shangdianzi | China | CMA_NOAA/ESRL | 40.65°N | 117.11°E | 293 | 15.0 | D |
| SEY | Mahe Island | Seychelles | NOAA/ESRL | 4.67°S | 55.17°E | 3 | 4.5 | D |
| SGP | Southern Great Plains | USA | NOAA/ESRL | 36.60°N | 97.49°W | 314 | 75.0 | D |
| SHM | Shemya Island | USA | NOAA/ESRL | 52.72°N | 174.10°E | 40 | 25.0 | D |
| SIS | Shetland | UK | CSIRO | 60.17°N | 1.17°W | 30 | 15.0 | D |
| SMO | Tutuila (Cape Matatula) | US Territory | NOAA/ESRL | 14.24°S | 170.57°W | 42 | 4.5 | D |
| SNB | Sonnblick | Austria | EAA | 47.05°N | 12.95°E | 3111 | 15.0 | C |
| SPO | South Pole | USA | NOAA/ESRL | 89.98°S | 24.80°W | 2810 | 4.5 | D |
| SSL | Schauinsland | Germany | UBA | 47.92°N | 7.92°E | 1205 | 15.0 | C |
| STM | Ocean Station "M" | Norway | NOAA/ESRL | 66.00°N | 2.00°E | 5 | 15.0 | D |
| SUM | Summit | Denmark | NOAA/ESRL | 72.58°N | 38.48°W | 3238 | 15.0 | D |
| SYO | Syowa Station | Japan | NOAA/ESRL | 69.00°S | 39.58°E | 11 | 4.5 | D |
| TAC | Tacolneston Tall Tower | UK | UNIVBRIS | 52.52°N | 1.14°E | 156 | 25.0 | C |
| TAP | Tae-ahn Peninsula | Republic of Korea | NOAA/ESRL | 36.73°N | 126.13°E | 20 | 75.0 | D |
| TDF | Tierra del Fuego | Argentina | NOAA/ESRL | 54.87°S | 68.48°W | 20 | 4.5 | D |
| TER | Teriberka | Russian Federation | MGO | 69.20°N | 35.10°E | 42 | 15.0 | D |
| THD | Trinidad Head | USA | NOAA/ESRL | 41.05°N | 124.15°W | 107 | 25.0 | D |
| TIK | Tiksi | Russian Federation | NOAA/ESRL | 71.59°N | 128.89°E | 31 | 15.0 | D |
| TKB | Tsukuba | Japan | MRI | 36.05°N | 140.13°E | 26 | 15.0 | C |
| USH | Ushuaia | Argentina | NOAA/ESRL | 54.85°S | 68.31°W | 12 | 4.5 | D |



| UTA | Wendover | USA | NOAA/ESRL | 39.90°N | 113.72°W | 1320 | 25.0 | D |
|---|---|---|---|---|---|---|---|---|
| UUM | Ulaan Uul | Mongolia | NOAA/ESRL | 44.45°N | 111.10°E | 914 | 25.0 | D |
| WIS | Sede Boker | Israel | NOAA/ESRL | 31.13°N | 34.88°E | 400 | 25.0 | D |
| WKT | Moody | USA | NOAA/ESRL | 31.31°N | 97.33°W | 251 | 30.0 | D |
| WLG | Mt. Waliguan | China | CMA_NOAA | 36.28°N | 100.90°E | 3810 | 15.0 | D |
| WSA | Sable Island | Canada | EC | 43.93°N | 60.02°W | 5 | 25.0 | C |
| WSA | Sable Island | Canada | EC | 43.93°N | 60.02°W | 5 | 25.0 | D |
| YON | Yonagunijima | Japan | JMA | 24.47°N | 123.02°E | 30 | 15.0 | C |
| ZEP | Zeppelinfjellet (Ny-Alesund) | Norway | NOAA/ESRL | 78.90°N | 11.88°E | 475 | 15.0 | D |
| ZGT | Zingst | Germany | UBA | 54.43°N | 12.73°E | 1 | 15.0 | C |
| ZSF | Zugspitze / Schneefernerhaus | Germany | UBA | 47.42°N | 10.98°E | 2673.5 | 15.0 | C |
| ZUG | Zugspitze | Germany | UBA | 47.42°N | 10.98°E | 2965.5 | 15.0 | C |

**Table 3.** List of aircraft profile measurement sites. *The observations from IMECC campaign contain samples from several sites and routes, i.e. the location is not site specific.

| Site Code | Station Name | Country | Project | Sampling heights (m) | | Data range (year) |
|---|---|---|---|---|---|---|
| | | | | [min] | [max] | |
| ORL | Orléans | France | CarboEurope | 100.0 | 3200.0 | 2006-2012 |
| BIK | Bialystok | Poland | CarboEurope | 223.8 | 3025.9 | 2007-2011 |
| HNG | Hegyhatsal | Hungary | CarboEurope | 300.0 | 3250.0 | 2006-2009 |
| GRI | Griffin | UK | CarboEurope | 550.0 | 3100.0 | 2006-2010 |
| IMECC* | | | IMECC | 19.5 | 13240.0 | 2009 |





**Table 4.** Mean emission estimates and their uncertainties before and after 2007 (Tg CH$_4$ yr$^{-1}$). The prior uncertainties are from inversion M1 and M3, i.e. of CTE-CH$_4$ v1.0. M2 (CTE-CH$_4$ v1.1) has higher prior uncertainties in all regions due to its set-up. For other regions, see Supplementary material. Emission estimates after 2007 that are more than 1 Tg CH$_4$ yr$^{-1}$ larger than those before 2007 are marked in bold.

| Region (mTC) | Total | | Anthropogenic | | Biosphere | |
|---|---|---|---|---|---|---|
| | Before 2007 | After 2007 | Before 2007 | After 2007 | Before 2007 | After 2007 |
| Global (1-20) | | | | | | |
| Prior | 532.9±86.7 | **566.0±102.6** | 313.0±80.7 | **350.5±97.5** | 172.8±31.6 | 171.8+/-31.8 |
| M1 | 507.0±45.1 | **526.3± 43.7** | 287.0±36.4 | **314.9±34.5** | 172.8±28.7 | 167.7+/-28.7 |
| M2 | 508.2±62.0 | **526.3± 60.9** | 311.4±50.2 | **326.0±49.7** | 149.7±45.1 | 156.6+/-44.1 |
| M3 | 509.1±45.9 | **527.6± 44.0** | 287.9±37.4 | **312.2±34.8** | 174.1±28.8 | 171.7+/-28.9 |
| Europe (11-14) | | | | | | |
| Prior | 56.2±14.2 | 55.0±14.5 | 45.4±13.6 | 45.0±14.1 | 9.8±3.9 | 9.0+/-3.5 |
| M1 | 54.2±10.4 | 51.5±10.5 | 46.8±10.3 | 43.8±10.5 | 6.4±2.7 | 6.8+/-2.5 |
| M2 | 53.3±13.3 | 53.3±13.3 | 45.1±13.4 | 45.1±13.5 | 7.2±3.6 | 7.1+/-3.4 |
| M3 | 59.7±10.6 | 58.5±10.7 | 50.9±10.6 | 49.1±10.7 | 7.7±2.7 | 8.4+/-2.5 |
| South American temperate (4) | | | | | | |
| Prior | 40.0±14.9 | **42.8±16.0** | 23.2±13.1 | **25.5±14.4** | 14.2±7.0 | 14.5+/-6.9 |
| M1 | 49.4±14.6 | **63.3±14.9** | 28.0±12.9 | **39.9±13.5** | 18.8±6.9 | 20.6+/-6.7 |
| M2 | 51.9±24.6 | **66.0±24.7** | 33.6±22.5 | **46.4±23.0** | 15.7±9.8 | 16.9+/-9.9 |
| M3 | 46.0±14.6 | **58.8±15.0** | 26.3±12.9 | **37.9±13.5** | 17.0±6.9 | 18.2+/-6.8 |
| Asian temperate (8) | | | | | | |
| Prior | 142.4±72.7 | **164.7±89.8** | 106.2±72.1 | **129.3±89.3** | 34.2± 9.6 | 33.4+/-9.5 |
| M1 | 76.3±24.2 | **83.7±20.1** | 36.9±25.0 | **50.1±20.7** | 37.4± 6.5 | 31.5+/-6.1 |
| M2 | 66.8±28.7 | **80.6±24.2** | 48.4±26.6 | **54.8±23.2** | 16.4±24.7 | 23.8+/-22.5 |
| M3 | 78.2±25.2 | **81.0±19.9** | 37.8±26.1 | **44.2±20.6** | 38.5± 6.9 | 34.8+/-6.4 |
| Asian tropical (9) | | | | | | |
| Prior | 67.7±15.8 | **70.8±16.6** | 30.6± 8.7 | **35.7± 9.8** | 31.1±13.2 | 31.3+/-13.3 |
| M1 | 67.5±14.3 | 68.3±14.7 | 32.0± 8.4 | **35.1± 9.3** | 29.6±12.1 | 29.4+/-12.1 |
| M2 | 69.2±27.8 | 67.5±28.8 | 32.2±23.0 | 32.5±24.7 | 31.1±19.6 | 31.3+/-19.7 |
| M3 | 63.2±14.3 | **65.1±14.8** | 29.8± 8.4 | **32.8± 9.4** | 27.4±12.2 | 28.5+/-12.2 |



**Table 5.** Root mean square error (RMSE) between TCCON and model XCH$_4$ with averaging kernel applied (ppb). The inversion with the smallest RMSE is marked in bold.

*1 = California Institute of Technology, 2012

*2 = Jet Propulsion Laboratory, 2007-2008

5     *3 = Jet Propulsion Laboratory, 2011-2012

| Site names | Coordinates | | Inversion | | |
|---|---|---|---|---|---|
| | Latitude | Longitude | M1 | M2 | M3 |
| Ascension Island | 7.92°S | 14.33°W | 26.8 | 26.2 | **21.7** |
| Bialystok, Poland | 53.23°N | 23.03°E | 17.2 | 17.4 | **10.4** |
| Darwin, Australia | 12.42°S | 130.89°E | 28.3 | 26.9 | **25.4** |
| Eureka, Canada | 80.05°N | 86.42°W | 13.6 | 13.9 | **8.8** |
| Garmisch, Germany | 47.48°N | 11.06°E | **11.7** | 12.1 | 15.3 |
| Indianapolis, IN, USA | 39.86°N | 86.00°W | 11.9 | 13.6 | **8.7** |
| Izana, Tenerife, Spain | 28.30°N | 16.50°W | 11.9 | 12.8 | **10.0** |
| Karlsruhe, Germany | 49.10°N | 8.44°E | 12.7 | 13.4 | **11.2** |
| Lauder, New Zealand (120HR) | 45.04°S | 169.68°E | 23.6 | 21.4 | **20.2** |
| Lauder, New Zealand (125HR) | 45.04°S | 169.68°E | 23.4 | 21.2 | **20.7** |
| Lamont, OK, USA | 36.60°N | 97.49°W | 17.0 | 19.6 | **12.4** |
| Park Falls, WI, USA | 45.95°N | 90.27°W | 13.9 | 15.7 | **10.6** |
| Pasadena, CA, USA (Caltech*1) | 34.14°N | 118.13°W | 14.3 | 16.6 | **11.0** |
| Pasadena, CA, USA (JPL*2) | 34.12°N | 118.18°W | 26.6 | 27.9 | **17.9** |
| Pasadena, CA, USA (JPL*3) | 34.12°N | 118.18°W | 24.1 | 25.4 | **16.3** |
| Reunion Island, France | 20.90°S | 55.49°E | 27.1 | 25.5 | **24.7** |
| Saga, Japan | 33.24°N | 130.29°E | 26.2 | 26.8 | **18.6** |
| Sodankylä, Finland | 67.37°N | 26.63°E | 13.3 | 13.2 | **11.3** |
| Wollongong, Australia | 34.41°S | 150.88°E | 36.6 | 34.4 | **34.0** |



**Table 6.** Root mean squared error (RMSE) between GOSAT and model XCH$_4$ with averaging kernel applied (ppb). The inversions with the smallest RMSE are marked in bold.

| Region (mTC) | Inversion | | |
|---|---|---|---|
| | M1 | M2 | M3 |
| Global (1-20) | 9.5 | 9.7 | **5.1** |
| Europe (11-14) | **11.5** | 12.1 | 16.3 |
| North American boreal (1) | **11.2** | 11.7 | 15.3 |
| North American temperate (2) | **10.1** | 11.3 | 11.7 |
| South American tropical (3) | 23.0 | 22.7 | **19.8** |
| South American temperate (4) | 17.4 | **15.9** | 16.0 |
| Northern Africa (5) | **7.8** | 9.8 | 8.9 |
| Southern Africa (6) | 18.2 | 17.3 | **16.3** |
| Eurasian boreal (7) | **12.2** | 12.9 | 17.5 |
| Asian temperate (8) | 10.5 | 12.2 | **10.2** |
| Asian tropical (9) | 22.7 | 23.9 | **17.3** |
| Australia (10) | 15.4 | 13.7 | **13.4** |
| South West Europe (11) | **12.5** | 12.9 | 15.8 |
| South East Europe (12) | **13.8** | 14.7 | 18.7 |
| North West Europe (13) | **15.0** | 16.0 | 19.1 |
| North East Europe (14) | **12.6** | 13.5 | 17.5 |
| Ocean (16-20) | 13.7 | 13.0 | **9.3** |