# Peer review of "Global methane emission estimates for 2000-2012 from CarbonTracker Europe-CH4 v1.0"

_Geoscientific Model Development, 2016_

## Short Comment (SC1) · 22 Aug 2016

Dear authors,

In my role as Executive editor of GMD, I would like to bring to your attention our Editorial version 1.1:

http://www.geosci-model-dev.net/8/3487/2015/gmd-8-3487-2015.html

This highlights some requirements of papers published in GMD, which is also available on the GMD website in the 'Manuscript Types' section:

http://www.geoscientific-model-development.net/submission/manuscript_types.html

In particular, please note that for your paper, the following requirement has not been met in the Discussions paper:

- "The main paper must give the model name and version number (or other unique identifier) in the title."

For a model evaluation it is important to know, which model version exactly was evaluated. Therefore, please add a version number for your model in the title upon your revised submission to GMD.

Yours,

Astrid Kerkweg

―――――――――――――――――――――

---

## Referee Comment (RC1) · Anonymous Referee #1 · 29 Sep 2016

This paper presents the results of a study assimilating ground-based in-situ methane concentrations into the CarbonTracker Europe-CH4 model using an Ensemble Kalman filter. The authors compare the posterior model with independent observations, and discuss the differences between the prior and posterior fluxes.

In general I find the paper well-presented and worthy of publication after the following comments are addressed.

**General comments**

A lot of figures are relegated to the supplemental information, while I would prefer to see them in the main paper. I appreciate this can be a matter of preference, but I have

made the case for their promotion in my comments below.

In most of the figures/tables showing either the model run with posterior emissions, only the posterior values are plotted. I would like to see the priors on the figures as well, in order to assess the improvement made by assimilating the data.

**Specific comments**

Pg. 5, first paragraph: I found it a bit confusing throughout the paper that the modified Transcom regions and the regions to be optimized were both referred to solely as "regions".

Pg. 5, line 12-13: There is a discrepancy here with respect to the companion paper Tsuruta et al., 2016: that paper has a different number of regions optimized.

Pg. 5, line 16: What was the variance for the ice region?

Pg. 6, line 30: I believe that many of these observations are not available for the whole 2000-2012 time period and you might consider giving the date range in Table 2. Does the assimilation system show any effects of observations turning "off" and "on"?

Pg. 7, line 3: In the companion paper (Tsuruta et al., 2016) the minimum mdm is 7.5 ppb. Is this a difference in the approach between the two papers? I'd also suggest you include which category of observation has this value.

Pg. 9, second paragraph and Figure 3: I really cannot make most of these observations out on Figure 3 because of the large range of values used. In fact, I had to stare at the posterior differences for quite a long time before I was sure the three sub-figures were different. I suggest plotting the posterior panels with a different colour scale so the differences can be made out.

Pg. 9, third paragraph: A2007 and B2007 are used here (and throughout the paper), but I don't believe they are formally defined.

Pg. 10, third paragraph: I think you've missed a chance to draw some conclusions
about the model due to the fact that the agreement with TCCON tends to be better before the averaging kernels are applied. As noted in the paper, applying the kernels does make the comparison more valid, as the effects of the prior used in the TCCON analysis and the instruments' vertical sensitivity are taken into account. The fact that this makes the agreement worse could indicate that there's some compensating effect in the total column, which I think you can see by looking at the comparisons with IMECC in Figure 5.

Pg. 10, line 15-16: Why not show the prior?

Pg. 10, line 24-25: I'm not sure I understand what you're getting at here: why would missing spring observations in one year affect the trend over the study period?

Pg. 10, line 32: Can you not be sure that the data was not used by the inversion?

Pg. 11, line 1: I know observations are few and far between in Australia, but Cape Grim is in a much different environment than Wollongong! Wollongong is a hard site to model in general – see for example Fraser et al., 2011.

Pg. 11, line 6: I think this is a typo? Darwin is closer to the Equator than Wollongong.

Pg. 11, last paragraph: The text occasionally refers to Figure 8 when it should be Figure S3 and vice versa.

Pg. 12, line 31: Differences are also surely due to the different time periods studied.

Pg. 14, line 8: Fig. Sx? I think this figure was dropped from the submitted manuscript.

Pg. 15, line 28: Should this be a reference to Fig. 11?

Pg. 16, line 22: Should this be Fig. S4?

Pg. 19, line 15: Is there something missing after i.e.? I don't understand the reference to i here.

Pg. 22, line 7: I don't understand what is meant here by the extra degree of freedom.

Pg. 23, line 10: Here is an example of where I found the term "region" to be confusing, as I thought at first you were referring to the optimized regions here, where in most of your inversions only one of anthropogenic or biogenic emissions were optimized.

Figure 1: Please add a reference here to Table 6 for the names of the regions.

Figure 2: I find the colour scheme a little difficult – very hard to make out the green regions in amongst the lighter blue! You could also consider blocking out the modified Transcom regions with black lines.

Figure 3 (again): For the comparisons with the prior I assume it's all *surface* observations that are used? (As opposed to TCCON etc...) Can you also plot the prior difference using only the assimilated observations? It would make it more straight-forward to assess the performance of the inversion.

Figure 4: Do the observations appear on the top panel? Can you also plot the differences between the observations and the prior and posterior (for XCH4 only)? You could consider adding a vertical line at 2007 in both panels.

Figure 5: I would use Figure S1 here since it shows the prior concentrations too. It would also be useful to give the prior RMSEs.

Figure 7: I would use Figure S2 here since you discuss many of the sites in that figure in the main paper. (And you don't discuss Park Falls, but it appears here.) I'd also like to see the prior concentrations included.

Figure 8: Again, I'd suggest using Figure S3 here and showing all the regions in the main paper. And I'd include the priors on these figures too.

Figure 12: This figure is impossible for me to decipher. I'd replace this with the bottom panels from Figure S4. (Or even move S4 to the main paper and remove Asian temperate from Figure 11.)

Figure 13: Since you have the space, you could include Australia in the bottom right.

[Figure]

Tables 5 and 6: Please include the prior RMSEs as well.

**Typographical comments**

Throughout the manuscript, "North" and "South" should be capitalized when referring to the continents (e.g. "North America") but not when referring to regions (e.g. "south Africa").

Pg. 1, line 31: . . . and (b) the parametrization. . ." (remove "on").

Pg. 2, line 31: . . .changes **in** emission sources. . .

Pg. 7, line 4: . . .20 times **mdm** because...

Pg. 9, line 4: . . .emissions and **if it** is actually. . .

Pg. 9, line 23: . . .change much **in** B2007. . .

Pg. 9, line 2: . . .Fig. 4 **are** calculated. . .

Pg. 19, line 19: . . .optimized in M1 **and** M3. . .

Pg. 19, line 32: . . .around the **coast** were not. . .

Pg. 22, line 12: . . .variation or **lack** of . . .

Pg. 22., line 31: . . .CTE-CH4 **differ** by the . . .

**References**

Tsuruta, A. et al.: Development of CarbonTracker Europe-CH4 –Part 1: system set-up and sensitivity analysis, 2016. (submitted to Geosci. Model Dev. Discuss.)

Fraser, A. et al.: The Australian methane budget: Interpreting surface and train-borne measurements using a chemistry transport model, J. Geophys. Res., 116, D20306, doi:10.1029/2011JD015964, 2011.

---

## Referee Comment (RC2) · Anonymous Referee #2 · 6 Oct 2016

Overview:

This paper presents an evaluation and interpretation of CarbonTracker Europe-CH4 (CTE-CH4). CTE-CH4 estimates global methane emissions by assimilating in-situ observations of methane mole fractions. It estimates scaling factors to prior emissions on the regional scale (order of 1000 km). It covers the time period 2000-2012, which includes a period of fairly constant global mean methane concentrations (2000-2006) and a period of growing global mean methane concentrations (2007-2012).

The model evaluation includes a comparison of the model with two different state spaces (sizes and shapes of scaling factor regions), as well as two different vertical mixing schemes. The work reported good agreement between predictions made by

the posterior model and independent data from aircraft profiles, TCCON, and GOSAT - with the best agreement found using a vertical mixing scheme from Gregory et al., 2000 and a state space with fewer elements. However, there were biases in the comparison to some of the independent data, as well as in the model residuals.

The model interpretation focused on trends in emissions from each of the optimized regions and the sensitivity of the estimates to the choice of state space and vertical mixing scheme. The authors attribute the renewed growth of global mean methane concentrations to increased anthropogenic emissions in the South American temperate region, Asian temperate region.

I think that there is some scientific potential in this work, but the article requires major revisions in order to realize that potential. I found the presentation to be below publication quality, and I had a few moderate but material concerns about the scientific quality.

I had to work far too hard to understand the work and its importance, which I attribute to a combination of poor communication of the message, a generally low quality of writing, and flaws in the figures.

My scientific concerns are:

1. that the model assumed a fixed lifetime for CH4 even though the authors explicitly acknowledge that this assumption is unlikely to hold. It is very important that the authors qualify any reported results from this model with this assumption.

2. that the inversion violates the assumptions that form its foundation in a way that likely aliased biases in the posterior emissions estimates.

3. that the prior and model-data mismatch uncertainty estimates appear to be arbitrary, and that no tests (e.g., reduced chi-squared statistic) were given to demonstrate that they accurately reflect the actual uncertainty distribution.

4. that the model evaluation examined only the maximum a posteriori estimate of the

inverse model and did not give an assessment of the uncertainty estimates (similar to point 3). This paper evaluates a model that generates a statistical distribution as output – that distribution should be evaluated in its entirety.

I think that point 1 could be dealt with by adjusting the language of the paper and more fully acknowledging the shortcomings, and that points 2, 3, and 4 could be dealt with by revising the uncertainty characterization and model evaluation analysis.

Detailed Comments:

Scientific Significance:

This article presents an evaluation and interpretation of an inverse modeling system designed to attribute the renewed growth in the global mean methane concentration since 2007. This is an important problem, and the literature on the subject is quickly evolving.

This work makes an incremental advancement by testing CTE-CH4. Due to the scientific concerns I raise in the Overview and Scientific Quality sections of this review, I would not believe that the findings of this work uncover the cause of the renewed growth with scientific rigor. However, if CTE-CH4 is further developed it might be a useful tool in contributing to the solution of the problem.

Scientific Quality:

I have reservations about the scientific quality of this work, but most of them could be addressed by refining the language to be more forthright about the limitations of the model and by revising the uncertainty characterization and model evaluation metrics.

My issues are as follows:

1. The authors assumed a lifetime of CH4 that varied seasonally, based on a one-year climatology of hydroxyl concentrations. They assumed that the CH4 lifetime had no inter-annual variability. This assumption is not safe, and the authors acknowledge it -

citing Rigby et al., 2008, Ghosh et al., 2015, and Dalsoren et al., 2016 (lines 21–31 on page 5). The atmospheric lifetime of CH4 likely changed during the study period, and error in the lifetime of CH4 would be directly aliased onto the retrieved emissions.

Additional evidence for changing OH concentrations and therefore changing atmospheric lifetimes of CH4 can be found in:

- McNorton J et al. (2016) Role of OH variability in the stalling of the global atmospheric ch4 growth rate from 1999 to 2006. Atmospheric Chemistry and Physics 16(12):7943–7956.

- Montzka SA et al. (2011) Small interannual variability of global atmospheric hydroxyl. Science 331(6013):67–9.

- Prather MJ, Holmes CD, Hsu J (2012) Reactive greenhouse gas scenarios: Systematic exploration of uncertainties and the role of atmospheric chemistry. Geophysical Research Letters 39.

- Rigby M et al. (2013) Re-evaluation of the lifetimes of the major CFCs and CH3CCl3 using atmospheric trends. Atmospheric Chemistry and Physics 13(5):2691–2702.

One would not expect the authors to implement a changing atmospheric lifetime at this point, but the issue should be treated very seriously and highlighted as a future research need. The caveat of a fixed methane lifetime should appear in the abstract.

2. The inversion setup violates one of the fundamental assumptions from which it is derived in a material way that leads me to doubt the validity of the conclusions. An inversion of this sort assumes that the error in the prior is a second-order (a.k.a. weak-sense) stationary Gaussian random process with zero mean.

The authors use the EDGAR 4.2 FT2010 emissions inventory as a prior anthropogenic emissions field. This is a high-resolution (0.1x0.1 degree) inventory that is known to be (and demonstrated in the paper to be) biased in its spatial distribution over a broad spectrum of scales and also biased in its temporal trend.

[Figure]

They set the prior error variance for the total emissions from any region to 0.8. The assignment is arbitrary and very likely too high. The authors effectively eliminate the bias by over-estimating the random error in the prior. Still, the posterior estimate is at the extreme of the error bounds, and so the bias in the prior affects the posterior estimate. This is visible in Fig. 3, where biases in the latitudinal distribution and seasonal cycles are visible in all posteriors.

Additionally, the scaling factors are resolved at spatial scales of thousands of kilometers. The error in the prior varies at scales much smaller than this – producing a severe representation error. The problem is therefore likely under-parameterized (equivalent to having covariance lengths that are too long, or regions too large), and so adequate scaling factors cannot be derived that permit unbiased residuals at individual sites. As a result, the authors find strong biases in the residuals, and even throw out some sites.

An example of such sites are given in the paper, and the authors remove them:

"Strong negative bias as found in Bukit Koto Tabang, Indonesia (BKT) (-25 to -27 ppb) and Mt. Kenya, Kenya (MKN) (-18 to -23 ppb), such that the posterior mole fractions were especially low during June-October. This suggests that the measurements at those latitudes are not representative of large regions optimized in the model."

To solve this problem, the authors would need to perform the inversions at high resolution using covariance length scales constrained as part of an objective error characterization.

3. There is no objective justification given for the uncertainty used in the study. The authors state: "Variance of the scaling factors was set to 0.8 for all regions, except for 'Ice' region (Fig. 2)" (sic).

It is important that the error statistics of the combined prior and model-data mismatch error match the error statistics of the actual mismatch between the prior model and the observations.

For more information on this problem, please see:

Michalak AM et al. (2005) Maximum likelihood estimation of covariance parameters for Bayesian atmospheric trace gas surface flux inversions. J. Geophys. Res. 110:D24107.

4. The model evaluation examined only the maximum a posteriori estimate of the inverse model and did not give an assessment of the uncertainty estimates (similar to point 3). This paper evaluates a model that generates a statistical distribution as output – that distribution should be evaluated in its entirety.

The analysis of residuals and independent data should test whether the posterior model-data mismatches follow the expected distributions.

Scientific Reproducibility:

This work is reproducible – in fact there have already been other studies that have conducted similar experiments and come to similar conclusions.

Presentation Quality:

This work is very poorly presented. My reservations about the presentation fall into three broad categories: 1) messaging, 2) writing, and 3) figures.

1. Messaging

This work is presented as model evaluation and interpretation. The paper goes into great detail about the variations in every region, giving their trends, comparisons to other regions, and comparisons to other papers. The work needs to be boiled down to a set of key messages. My understanding is that the main messages are those described in the Overview section of this review.

The body of the paper needs to be focused on providing the scientific justification for the given messages.

2. Writing

The paper requires extensive revision by an English language editor. Problems include:

- incorrectly cased letters (e.g., "south America" should be "South America"). - inconsistent tenses and active vs. passive voice (e.g., in the abstract, line 29 "We use three configurations. . .", then line 32 "The posterior estimates were evaluated. . ."). - broken sentences (e.g., page 3, line 22 "To estimate biospheric emissions, information from an underlying ecosystem distribution map is useful, which defines the location of the sources and can help distribute larger regions over which the atmospheric signals integrate."). - truisms (e.g., page 9, line 20 "The growth rate (GR) of atmospheric methane mole fractions showed that the posterior estimates are closer to the observations than the prior, as expected"). - paragraphs that are incredibly long and rambling (e.g., page 14, line 7 to page 15, line 6; page 16, line 13 to page 17, line 5; and page 18, lines 5 - 30).

An exhaustive list of problematic sentences would be too long for this review.

3. Figures

The figures in this paper have a number of issues. Points a-c absolutely must be addressed in order for the paper to be publishable.

a) Figs. 9, 10, 11, 14, S4, and S5 include data and/or error bars that run off of the figure.

b) Many of the figures include error bars but do not specify their meaning (1 standard deviation? 95% credible interval?)

c) Figs. 4, 5, 6, 7, 8, 9, 12, 13, S1, S2, S3, and S6 are not colorblind safe.

d) Figs. 6 and 7 are difficult to read because of closely placed points.

e) Fig. 4 (top panel) should include observations.

---

## Author Comment (AC1) · 13 Dec 2016

Authors' response to anonymous referee #1

In the following, referee's comments are in *italic*, authors' responses in normal font, and references (page, line, figure, and table number) to the revised manuscript in **bold**. Please note that this paper is merged with the accompanying paper, following the referees' comments and with approval from the Topical Editor. A summary of the accompanying paper was included in the Supplementary Material of this paper.

In addition, figures and tables were revised substantially, and the following table summarizes the changes to figure and table numbers. In addition, please note that Fig. 2, Fig. 3, and Fig. 4 of the accompanying paper were moved to the Supplementary Material as Fig. S1, Fig. S2, Fig. S3.

| Revised | Original | Short description |
| --- | --- | --- |
| Fig. 1 | Fig. 1 | mTC regions + observation sites |
| Fig. 2 | Fig. 3 | Comparison with assimilated observations |
| Fig. 3 | Fig. 4 | Global mean $XCH_4$ and growth rates |
| Fig. 4 | Fig. 5, S1 | Comparison with aircraft observations |
| Fig. 5 | Fig. 6 | Model performance in Europe at assimilated sites |
| Fig. 6 | Fig. 7 | Comparison with TCCON observations |
| Fig. 7 | Fig. 8 | Comparison with GOSAT observations |
| Fig. 8 | Fig. 11, 12, S4 | Emission estimates of global and Asian temperate and tropical mTC regions. |
| Fig. 9 | Fig. 9 | Growth rates of global emission estimates |
| Fig. S4 | Fig. 2 | Land-ecosystem map |
| Fig. S5 | Fig. S2 | Comparison with TCCON observations |
| Fig. S6 | Fig. S3 | Comparison with GOSAT observations |
| Fig. S7 | Fig. S6 | Monthly mean of total emission estimates at latitudinal bands |
| Fig. S8 | Fig. 8, 10, 13 | Emission estimates of land mTC regions |
| Fig. S9 | Fig. 14, S5 | Emission estimates of ocean mTC regions. |
| Table 1 | Table 1 | Inversion set-up |
| Table 2 | Table 2 | List of observation sites used in the inversions |
| Table 3 | Table 3 | List of aircraft observation sites |
| Table 4 | Table 5 | RMSE with TCCON observations |
| Table 5 | Table 6 | RMSE with GOSAT observations |
| Table 6 | Table 4 | Global and regional emission estimates |

*General comments*
*A lot of figures are relegated to the supplemental information, while I would prefer to see them in the main paper. I appreciate this can be a matter of preference, but I have made the case for their promotion in my comments below.*

*In most of the figures/tables showing either the model run with posterior emissions, only the posterior values are plotted. I would like to see the priors on the figures as well, in order to assess the improvement made by assimilating the data.*

We appreciate the reviewer's suggestions, and the figures and tables were revised extensively following the specific comments below. We agree with the reviewer that the figures and tables became more complete by including the prior. The prior is added in the figures and tables in the revised manuscript. Please see below for responses and changes in more detail.

*Specific comments*
*Pg. 5, first paragraph: I found it a bit confusing throughout the paper that the modified Transcom regions and the regions to be optimized were both referred to solely as "regions".*

We apologize for the confusion. We agree with the reviewer that the definition of 'regions' was confusing in

many sentences. We have tried to make this clear in the revised manuscript by referring to to modified TransCom regions as mTCs and adding the word 'optimization' for the optimization regions where the definition is unclear.

*Pg. 5, line 16: What was the variance for the ice region?*

The variance for the ice region was $1 \times 10^{-8}$. It was stated in the accompanying paper, but we agree with the reviewer that it was not clear only from this paper. In the revised manuscript, the number is explicitly added.

**Text is revised: Pg. 6, line 18.**

*Pg. 6, line 30: I believe that many of these observations are not available for the whole 2000-2012 time period and you might consider giving the date range in Table 2. Does the assimilation system show any effects of observations turning "off" and "on"?*

This an excellent and very interesting point. We indeed acknowledge the sensitivity of the inversions to the set of observations assimilated, and the emission estimates could vary by turning "off" and "on" some of the observations. We surmise that the effect would be mostly in regional emission estimates, and less in continental scale estimates. This is because many "background" sites that constrain large-scale emissions have long-term observations, and the number of "new" background observations is low. The effect was especially prominent in regions that were already poorly constrained by the current background network: sites in South America had a significant effect on the increase in regional emission estimates (see Section 3.4.5). Although we did not test this further for all sites, we also discussed the effect for other observations (discrete vs discrete+continuous (see Supplementary Material), and GSN, AMY in Section 3.4.3), and we feel that this is sufficient for the paper at this point. Finally, we followed the reviewers suggestion and the date range is added to Table 2.

**The date range is added to Table 2.**

*Pg. 7, line 3: In the companion paper, the minimum mdm is 7.5 ppb. Is this a difference in the approach between the two papers? I'd also suggest you include which category of observation has this value.*

Firstly, we thank the reviewer for also carefully reading the accompanying paper. As pointed out by the reviewer, the minimum mdm was different in the two papers. We found that the sites with low mdm (i.e. marine boundary layer or high southern latitude sites) were particularly important to remove the effect of initial 3D mole fraction fields and to constrain background concentrations well. In addition, these sites were important to correct a bias in the latitudinal gradient of the prior emissions. In the long-term inversion we noticed that the constraints introduced by these sites was not strong enough (not shown or published), and therefore, we gave greater weight (=lower mdm) for those sites for the long-term experiment. In the revised paper, the table and description of the accompanying paper regarding mdm was omitted. The main finding of the sensitivity tests in the Supplementary Material does not change due to differences in the mdm of those sties, and therefore, we hope the reviewer agrees that the alternative description of the mdm was not necessary to be included in the Supplementary Material.

*Pg. 9, second paragraph and Figure 3: I really cannot make most of these observations out on Figure 3 because of the large range of values used. In fact, I had to stare at the posterior differences for quite a long time before I was sure the three sub-figures were different. I suggest plotting the posterior panels with a different colour scale so the differences can be made out.*

We apologize for difficulties that arose by the poor illustration in the figure. This is an excellent suggestion and we revised the figure by plotting posteriors in a different scale.

**Fig. 2 (original Fig. 3) is revised.**

*Pg. 9, third paragraph: A2007 and B2007 are used here (and throughout the paper), but I don't believe they are formally defined.*

We acknowledge that the words were not descriptive enough. It was referred to only once in a much earlier

section. To make this clear, the text was revised in the new manuscript, where A2007 is simply written as "after 2007", and B2007 as "before 2007".

*Pg. 10, third paragraph: I think you've missed a chance to draw some conclusions about the model due to the fact that the agreement with TCCON tends to be better before the averaging kernels are applied. As noted in the paper, applying the kernels does make the comparison more valid, as the effects of the prior used in the TCCON analysis and the instruments' vertical sensitivity are taken into account. The fact that this makes the agreement worse could indicate that there's some compensating effect in the total column, which I think you can see by looking at the comparisons with IMECC in Figure 5.*

We agree with the reviewer that the point was missing in the text. This is a very interesting point in regard to the application of averaging kernels. As pointed out by the reviewer, the comparison with IMECC showed that model $CH_4$ was lower than the observations and agreement was worse at lower altitudes. The weights at lower altitudes were greater than those at higher altitudes when the averaging kernel was applied. Therefore, model $XCH_4$ decreased and agreement with retrieved $XCH_4$ became worse when the averaging kernel was applied. Although we acknowledge the effect of the averaging kernel, we decided to remove the text, as all comparisons were done with the averaging kernel applied.

*Pg. 10, line 15-16: Why not show the prior?*

We acknowledge the importance of also showing the improvements by the inversions for $XCH_4$ estimates. In the original figures, the prior was not included, because it was already clear from the comparison with the in situ observations that the prior $XCH_4$ would also be much higher than the observations. Since prior $XCH_4$ also increases significantly more than the observations and the posterior, the differences between the posterior estimates and that with the observations were less clear when prior estimates were plotted. However, in the revised figures, we included the priors.

**Fig 6 (original Fig. 7) is revised.**

*Pg. 10, line 24-25: I'm not sure I understand what you're getting at here: why would missing spring observations in one year affect the trend over the study period?*

We appreciate the reviewer for noticing this inconsistency. We agree with the reviewer that the missing spring observations were not the reason for the mismatch in trends because the model trend was calculated from the same date and location of the observations, and using the averaging kernel.

We looked at the trend further, and found that the model seems to overestimate the trend (see figure below). $XCH_4$ before 2010 was lower and after 2010 was higher than the observations. A similar feature was found at other sites, e.g. at Park Falls and Lamont, such that the increasing trend for 2007-2013 was stronger in the model than in the observations. This could be caused by underestimation of emissions in northern temperate regions before 2010 and a significant increase after 2010. This may indicate that our inversion does not estimate the start of the emission increase in temperate regions correctly. The discussion is added in the revised manuscript.

**Text is revised: Pg. 12, line 9-11.**

[Figure]

[Figure]

*Pg. 10, line 32: Can you not be sure that the data was not used by the inversion?*

Here, we tried to address that the inversions probably did not learn much from GPA observations because mdm was set large (75 ppb). We apologize that the sentence was misleading. We are certain the data were used and assimilated in the inversions, but the high mdm limited the effect of the observations on the emissions to some extent. In theory, the effect could be checked by running inversions without the data, although such sensitivity test was not done as it would have been an expensive (CPU time) extra test. The sentence in the new manuscript is revised to address that the data were assimilated in the system.

**Text is revised: Pg 12, line 23-24.**

*Pg. 11, line 1: I know observations are few and far between in Australia, but Cape Grim is in a much different environment than Wollongong! Wollongong is a hard site to model in general – see for example Fraser et al., 2011.*

This is a good point. We agree with the reviewer that the $XCH_4$ cannot simply be compared because the sites are near. As pointed out by the reviewer, Cape Grim is located on an island far from cities and has little local emissions, whereas Wollongong is located in the city of Wollongong, where local emissions could have a large influence. We have added this to the discussion in the revised manuscript.

**Text is revised: Pg. 12, line 17-19.**

*Pg. 11, line 6: I think this is a typo? Darwin is closer to the Equator than Wollongong.*

We apologize for the incorrect sentence. The reviewer is correct. Darwin is closer to the Equator than Wollongong. The argument therefore needed to be rephrased. Considering its surrounding environment (see also the above comment by the reviewer), we are now more certain that the model bias in the Wollongong was likely due to an error in local emission estimates rather than long-range transport. We have revised the text in the new manuscript.

**Text is revised: Pg. 12, line 17-24.**

*Pg. 11, last paragraph: The text occasionally refers to Figure 8 when it should be Figure S3 and vice versa.*
*Pg. 14, line 8: Fig. Sx? I think this figure was dropped from the submitted manuscript.*
*Pg. 15, line 28: Should this be a reference to Fig. 11?*
*Pg. 16, line 22: Should this be Fig. S4?*

> We apologize for the confusion that arose by inconsistencies in the figure references. We revised the references in the new manuscript.

*Pg. 12, line 31: Differences are also surely due to the different time periods studied.*

> We agree with the reviewer that the differences in the increasing rate are also due to the differences in time periods compared. We added further discussion in the revised manuscript.

> **Text is revised: Pg. 14, line 2-3.**

*Pg. 19, line 15: Is there something missing after i.e.? I don't understand the reference to i here.*

> We apologize for the poor formulation in this sentence. The sentence was meant to say that it was not possible to conclude from this study which estimates better capture actual emissions, because the emission estimates for Africa were not well constrained by observations in either study. We revised the new manuscript accordingly.

> **Text is revised: Pg. 20 line 5-6.**

*Pg. 22, line 7: I don't understand what is meant here by the extra degree of freedom.*

> The sentence was meant to explain that estimates of Eurasian boreal were sensitive to estimates of other regions. Due to very sparse observations in boreal Eurasia, the scaling factors of in this mTC region acted more as a compensating effect. We revised the sentence in the new manuscript.

> **Text is revised: Pg. 22 line 27-29.**

*Pg. 23, line 10: Here is an example of where I found the term "region" to be confusing, as I thought at first you were referring to the optimized regions here, where in most of your inversions only one of anthropogenic or biogenic emissions were optimized.*

> Here, the first 'region' was meant to be an optimization region, and the second one to be the mTCs. Both anthropogenic and biospheric emissions were optimized in all mTC regions over land, and the system tends to change optimization regions that have large emissions near the observation sites. In many mTC regions, anthropogenic emissions were greater than biospheric emissions near the observation sites, and therefore, the inversion tends to scale anthropogenic emissions more. This then led to larger changes in the anthropogenic emission estimates for optimization regions and also in mTC regions. We revised the sentence in the new manuscript.

> **Text is revised: Pg. 24 line 1-2.**

*Figure 1: Please add a reference here to Table 6 for the names of the regions.*

> Following the reviewer's suggestion, the caption of Figure 1 is modified by adding a reference to Table 5 (original Table 6).

> **Caption of Fig. 1 is modified.**

*Figure 2: I find the colour scheme a little difficult – very hard to make out the green regions in amongst the lighter*

*blue! You could also consider blocking out the modified Transcom regions with black lines.*

We follow the reviewer's suggestion and changed the colour scale of Fig. S4 (original Fig. 2), and mTC and land ecosystem region borders were added.

**Fig. S4 (original Fig. 2) is revised.**

*Figure 3 (again): For the comparisons with the prior I assume it's all surface observations that are used? (As opposed to TCCON etc. . .) Can you also plot the prior difference using only the assimilated observations? It would make it more straight-forward to assess the performance of the inversion.*

We revised the figure by plotting the prior differences between assimilated observations. In addition, the colour scale of the posterior comparison changed following the reviewer's earlier comment.

**Fig. 2 (original Fig. 3) is revised.**

*Figure 4: Do the observations appear on the top panel? Can you also plot the differences between the observations and the prior and posterior (for $XCH_4$ only)? You could consider adding a vertical line at 2007 in both panels.*

Observations were not plotted in the figure because NOAA global averages are for the surface, unlike $XCH_4$. However, we agree with the reviewer that the figure becomes more comprehensive by adding the observations in the top panel. We followed the reviewer's suggestion and added NOAA surface global mean $CH_4$ with a second y-axis. In addition, a vertical line at 2007 is added to both panels.

We did not include the differences between the NOAA global mean and the simulated $XCH_4$, as additional complications arise due to sampling. The top panel illustrates global mean $XCH_4$ from the model, but no satellite products have sufficient global coverage throughout the years to make a fair comparison with it. In Fig. 7 comparison with GOSAT $XCH_4$ is included, where model $XCH_4$ was sampled at the locations and times of the observations. We hope the reviewer agrees that the comparison in Fig. 7 is satisfying.

**Fig. 3 (original Fig. 4) is revised.**

*Figure 5: I would use Figure S1 here since it shows the prior concentrations too. It would also be useful to give the prior RMSEs.*

We acknowledge the importance of showing the improvements by the inversions for $XCH_4$ estimates. Following the reviewer's suggestion, we added the prior in Fig. 4 (original Fig. 5), and removed original Fig. S1.

In addition, we agree with the reviewer that it is also important to show the prior RMSE. For that, we found a table was more appropriate and therefore, both prior and posterior RMSE are summarized in Table 3 in the revised manuscript.

**Fig. 4 (original Fig. 5) and Table 3 are revised.**

*Figure 7: I would use Figure S2 here since you discuss many of the sites in that figure in the main paper. (And you don't discuss Park Falls, but it appears here.) I'd also like to see the prior concentrations included.*

*Figure 8: Again, I'd suggest using Figure S3 here and showing all the regions in the main paper. And I'd include the priors on these figures too.*

Response to the comment on Fig. 7 and Fig. 8:
We acknowledge that not only a selection of the sites and regions but all sites and regions are important to illustrate the results. We tried to make satisfactory figures with all sites and regions, but could not find a better illustration. Therefore, we decided to retain the figures, but selecting the sites and regions more carefully based on the results discussed in the text. We hope the reviewer agrees that the new figures illustrate the findings better.

**Fig. 6 (original Fig. 7) and Fig. 7 (original Fig. 8) are revised.**

*Figure 12: This figure is impossible for me to decipher. I'd replace this with the bottom panels from Figure S4. (Or even move S4 to the main paper and remove Asian temperate from Figure 11.)*

We agree with the reviewer that the figure in the Supplementary Material illustrates our results and discussion better. Following the suggestion, original Fig. S4 is moved to the main paper with modifications as Fig. 8. The original Fig. 11 is removed.

**Fig. 8 (original Fig. S4) is revised.**

*Figure 13: Since you have the space, you could include Australia in the bottom right.*

We agree with the reviewer that the figure is more complete when emission estimates for Australia are added. Following the reviewer's suggestion, the estimates are illustrated in Fig. S8 of the revised manuscript, together with other land mTC regions. We hope the reviewer agrees that the SH emission estimates could be represented in the Supplementary Material. This was done to better balance the manuscript by reducing the number of figures.

**Fig. S8 (original Fig. 13) is revised.**

*Tables 5 and 6: Please include the prior RMSEs as well.*

We agree with the reviewer that the tables are more complete when prior RMSE is added. Following the suggestion, prior RMSE is added in both tables.

**Table 4 (original Table 5) and Table 5 (original Table 6) are revised.**

*Typographical comments*
*Throughout the manuscript, "North" and "South" should be capitalized when referring*
*to the continents (e.g. "North America") but not when referring to regions (e.g. "south*
*Africa").*
*Pg. 1, line 31: . . . and (b) the parametrization. . ." (remove "on").*
*Pg. 2, line 31: . . .changes in emission sources. . .*
*Pg. 7, line 4: . . .20 times mdm because...*
*Pg. 9, line 4: . . .emissions and if it is actually. . .*
*Pg. 9, line 23: . . .change much in B2007. . .*
*Pg. 9, line 2: . . .Fig. 4 are calculated. . .*
*Pg. 19, line 19: . . .optimized in M1 and M3. . .*
*Pg. 19, line 32: . . .around the coast were not. . .*
*Pg. 22, line 12: . . .variation or lack of . . .*
*Pg. 22., line 31: . . .CTE-CH$_4$ differ by the . . .*

We appreciate the reviewer for the language correction. The manuscript is revised taking into account all the suggested changes.

---

## Author Comment (AC2) · 13 Dec 2016

In the following, referee's comments are in *italic*, authors' responses in normal font, and references (page, line, figure, and table number) to the revised manuscript in **bold**. Please note that this paper is merged with the accompanying paper, following the referees' comments and with approval from the Topical Editor. A summary of the accompanying paper was included in the Supplementary Material of this paper.

In addition, figures and tables were revised substantially, and the following table summarizes the changes to figure and table numbers. In addition, please note that Fig. 2, Fig. 3, and Fig. 4 of the accompanying paper were moved to the Supplementary Material as Fig. S1, Fig. S2, Fig. S3.

| Revised | Original | Short description |
|---------|----------|-------------------|
| Fig. 1 | Fig. 1 | mTC regions + observation sites |
| Fig. 2 | Fig. 3 | Comparison with assimilated observations |
| Fig. 3 | Fig. 4 | Global mean $XCH_4$ and growth rates |
| Fig. 4 | Fig. 5, S1 | Comparison with aircraft observations |
| Fig. 5 | Fig. 6 | Model performance in Europe at assimilated sites |
| Fig. 6 | Fig. 7 | Comparison with TCCON observations |
| Fig. 7 | Fig. 8 | Comparison with GOSAT observations |
| Fig. 8 | Fig. 11, 12, S4 | Emission estimates of global and Asian temperate and tropical mTC regions. |
| Fig. 9 | Fig. 9 | Growth rates of global emission estimates |
| Fig. S4 | Fig. 2 | Land-ecosystem map |
| Fig. S5 | Fig. S2 | Comparison with TCCON observations |
| Fig. S6 | Fig. S3 | Comparison with GOSAT observations |
| Fig. S7 | Fig. S6 | Monthly mean of total emission estimates at latitudinal bands |
| Fig. S8 | Fig. 8, 10, 13 | Emission estimates of land mTC regions |
| Fig. S9 | Fig. 14, S5 | Emission estimates of ocean mTC regions. |
| Table 1 | Table 1 | Inversion set-up |
| Table 2 | Table 2 | List of observation sites used in the inversions |
| Table 3 | Table 3 | List of aircraft observation sites |
| Table 4 | Table 5 | RMSE with TCCON observations |
| Table 5 | Table 6 | RMSE with GOSAT observations |
| Table 6 | Table 4 | Global and regional emission estimates |

*Overview:*

*Scientific concerns:*

*1. that the model assumed a fixed lifetime for CH₄ even though the authors explicitly acknowledge that this assumption is unlikely to hold. It is very important that the authors qualify any reported results from this model with this assumption.*

This is an excellent point. As the reviewer points out, our results depend on the assumption of a fixed $CH_4$ lifetime, but we agree with the reviewer that the assumption is unlikely hold. Montzka et al. (2011) found an increase in OH concentrations in the beginning of the 21st century, followed by a decrease in OH concentrations after 2004-2005. Similarly, Ghosh et al. (2015) and Dalsøren et al. (2016) also obtained a decrease in the $CH_4$ lifetime in their simulations. In addition, McNorton et al. (2015) showed that although interannual variability of OH may be small, small changes in OH concentrations could lead to significant changes in $CH_4$ concentrations. We did not carry additional sensitivity test on $CH_4$ lifetime since the uncertainty in changes in OH concentrations and its relation to the $CH_4$ burden is still high, as discussed by Prather et al. (2012). We hope the reviewer agrees that further discussion added in the revised manuscript

based on suggested studies is satisfactory. We have also stated this assumption in the abstract and conclusions.

**Text is revised: see e.g. Pg. 3 line 8-15.**

*2. that the inversion violates the assumptions that form its foundation in a way that likely aliased biases in the posterior emissions estimates.*

This is a very interesting point. We acknowledge that the assumptions such as the prior emission estimates and the representativeness of the atmospheric observations affected the inversion results. As pointed out by the reviewer (see also the comments below, Detailed Comments 2), the fundamental assumption that the prior is normally distribute with mean 1 may not hold, and the bias of prior spatial distribution remains in the posterior to a certain extent. Although the posterior atmospheric $CH_4$ values in the Northern Hemisphere agreed fairly well with the observations, we find negative bias in the posterior $CH_4$ values in the Southern Hemisphere. Although we could not eliminate the bias completely nor find the exact cause, we hope the reviewer agrees that the findings are meaningful and this point is to be examined continuously in future developments. We added further discussion on this issue in the revised manuscript. Please also see the responses to Detailed Comments 2.

*3. that the prior and model-data mismatch uncertainty estimates appear to be arbitrary, and that no tests (e.g., reduced chi-squared statistic) were given to demonstrate that they accurately reflect the actual uncertainty distribution.*

This is again an excellent point. We agree with the reviewer that the prior and model-data mismatch uncertainty were indeed somewhat arbitrary. The values could not be chosen based on a theory or numerical method, because such a theory or method is not yet developed to estimate the covariance structure exactly. However, as the reviewer pointed out, the assumptions can be examined by the method presented by Michalak *et al.* (2005). We have now examined the Chi-squared statistics ($\chi^2$), following the reviewer's suggestion.

Most of $\chi^2$ for the in situ observation sites ranged between $0 < \chi^2 < 2$ (Fig. R1), indicating that the chosen mdm were in range of the expected value. However, $\chi^2$ for most marine boundary layer sites are greater than one (Fig. R1), which indicates that the chosen prior mdm uncertainties were low. The $\chi^2$ of these sites were high probably due to model errors rather than observational errors. The $\chi^2$ for these sites were high for $L^{62}T$ especially because of the negative bias found around 2002, which was the most prominent among the inversions. The negative bias in $L^{62}G$ was not as strong as in $L^{62}T$, and $\chi^2$ was closer to one in $L^{62}G$ than $L^{62}T$ for the mbl sites (Fig. R1).

On the other hand, some sites have low $\chi^2$, indicating that the chosen mdm was larger than the expected uncertainty. However, mdm uncertainties for the continental sites should be assigned carefully because spatial representativity of the measurements may be low. Since the system optimizes emission estimates region-wise in this study, assigning mdm that are too small could lead to larger influence of the observations to the regional estimates than the observations would represent. The posterior ensemble distribution of $\chi^2$ statistics followed normal distributions for all the sites based on normality tests (see Fig. R2 for an example), indicating that the normality assumption in the prior holds.

Regional $\chi^2$ statistics were also distributed around 1 (Fig. R3). However, region mTC8 had high $\chi^2$, indicating that the prior uncertainty was lower than expected. This suggests that higher prior uncertainty or better prior emission estimates for the Asian temperate region was needed. On the other hand, regions such as mTC3, 5, and 6 have low $\chi^2$. For these regions, smaller prior uncertainties could have been used since the inversions did not retrieve much information from the observations. However, smaller prior uncertainties would lead to smaller posterior uncertainties, which may mislead the credibility of the emission estimates because having smaller posterior uncertainty does not necessary mean the estimates are reliable. The $\chi^2$ statistics of $L^{62}T$ were closer to 1 compared to $L^{78}T$, whose covariance matrix was diagonal (Fig. R3). This indicates that the assumed correlations between the scaling factors were probably appropriate to a certain extent.

The values in covariance matrices could be adjusted further. However we should note that the resulting $\chi^2$ depends on e.g. the choice of prior emission and observation data sets, and an arbitrary combination of these may or may not be better than some other, as noted by Michalak *et al.* (2005).

**Text is revised: see e.g. Pg.11 line 9-11, and Pg. 22, line 22-25.**

[Figure]

**Figure R1.** Chi-squared statistics from inversion $L^{62}T$ (top) and $L^{62}G$ (bottom) at the assimilated sites. The red triangles indicate marine boundary layer (mbl) sites. The sites with Chi-squared statistic larger than 2ppb or smaller than 0.2 are marked with three-letter site code.

[Figure]

**Figure R2.** Example distribution of Chi-squared statistic at an assimilate site (CHR: Christmas Island). Skewness (skw) and Kurtosis (kur) are shown to indicate normality of the distribution. Note that variance of the distribution was often very small.

[Figure]

**Figure R3.** Chi-squared statistic of regional estimates per mTC region: (left) $L^{62}T$, (right) $L^{78}T$.

*4. that the model evaluation examined only the maximum a posteriori estimate of the inverse model and did not give an assessment of the uncertainty estimates (similar to point 3). This paper evaluates a model that generates a statistical distribution as output – that distribution should be evaluated in its entirety.*

We agree with the reviewer that the analysis based on not only the optimum (mean) posterior mole fractions, but also the ensemble distributions are important. Following the suggestion, we extended the analysis and its discussion is included in the revised manuscript.

Distribution of ensemble mole fractions at assimilated in situ sites showed that the ensemble variation in $CH_4$ was small in general (<5ppb; Fig. R4). However, Black Sea, Constanta (BSC) has the exceptionally high standard deviation (std) of the ensemble, which indicates the difficulty in the inversions to close the emission budgets nearby. The observation network around BSC was very sparse, and emission estimates around it have large uncertainty. The posterior std was also high at the sites in west and central Asia (KZD, UUM, WIS), suggesting that the emissions there were not well constrained. Small differences were found in the ensemble std at in situ sites between inversions (not shown).

Distribution of ensemble $XCH_4$ showed that the standard deviation were less than 3 ppb globally and less than 1 ppb at TCCON sites (Fig. R5, Table R1). Largest deviation was found in South American tropical region and around north west and south east Asia, again addressing the difficulty of the inversion to close budget in those regions. The results also support the finding at in situ sites. This is expected, as deviation at lower altitude affect the $XCH_4$ deviation the most.

**Text is revised: see e.g. Pg 10, line 28-30, and Pg. 12 line 33 – Pg. 13 line 1.**

[Figure]

**Figure R4.** Average standard deviation (std) of ensembles per site. Red triangles illustrate marine boundary layer (mbl) sites. The sites with the std higher than 5 ppb are marked with three-letter site code.

[Figure]

**Figure R5:** Average standard deviation (ppb) of posterior XCH$_4$ ensemble.

**Table R1:** Average standard deviation (std) of posterior XCH$_4$ ensemble per TCCON site.

| Sites | std (ppb) |
|---|---|
| Ascension Island, Saint Helena, Ascension and Tristan da Cunha | 0.00 |
| Bialystok, Poland | 0.37 |
| Darwin, Australia | 0.01 |
| Eureka, Canada | 0.00 |
| Garmisch, Germany | 0.17 |
| Indianapolis, Indiana, USA | 0.28 |
| Izana, Tenerife, Spain | 0.02 |
| Saga, Japan | 0.06 |
| California Institute of Technology, Pasadena, California, USA | 0.18 |
| Karlsruhe, Germany | 0.17 |
| Lauder, New Zealand, 120HR | 0.04 |
| Lauder, New Zealand, 125HR | 0.04 |
| Lamont, Oklahoma, USA | 0.53 |
| Park Falls, Wisconsin, USA | 0.21 |
| Reunion Island, France | 0.02 |
| Sodankylä, Finland | 0.08 |
| Wollongong, Australia | 0.09 |

*Scientific Quality:*

For Point 1 of $CH_4$ lifetime, Point 3 of uncertainty assumption, and Point 4 of posterior distribution and analysis of posterior uncertainty, we have considered the reveiwer's comments together with the comments in the Overview section, and therefore, the response is included above.

*2. The inversion setup violates one of the fundamental assumptions from which it is derived in a material way that leads me to doubt the validity of the conclusions. An inversion of this sort assumes that the error in the prior is a second-order (a.k.a. weak-sense) stationary Gaussian random process with zero mean.*

*The authors use the EDGAR 4.2 FT2010 emissions inventory as a prior anthropogenic emissions field. This is a high-resolution (0.1x0.1 degree) inventory that is known to be (and demonstrated in the paper to be) biased in its spatial distribution over a broad spectrum of scales and also biased in its temporal trend.*

*They set the prior error variance for the total emissions from any region to 0.8. The assignment is arbitrary and very likely too high. The authors effectively eliminate the bias by over-estimating the random error in the prior. Still, the posterior estimate is at the extreme of the error bounds, and so the bias in the prior affects the posterior estimate. This is visible in Fig. 3, where biases in the latitudinal distribution and seasonal cycles are visible in all posteriors.*

This is a very interesting point. It is true that the bias in the spatial distribution and temporal trend of the EDGARv4.2 FT2010 inventory has been reported, and may have violated the prior assumption. We acknowledge that the bias in the prior is one of the reasons why large prior uncertainty was needed. However, the inventory has the advantages that it provides global estimates at high resolution and long temporal coverage, which the inversions benefit from. The reported biases were not taken into account, as these are also uncertain, and it would bring uncertainty in the prior estimates in some other way, such as model bias. We assumed the inversion could correct it, but we agree with the reviewer that some bias still remained, even with the high prior uncertainty. Therefore, we would like to address that the prior estimates also need improvement, especially for regions with sparse measurement coverage. In addition, it is important to note that the bias in atmospheric $CH_4$ seen in Fig. 2 (original Fig. 3) could also be due to atmospheric transport. For example, the observed seasonal cycle was better captured using the faster vertical mixing scheme in TM5. This indicates that not only the prior emission estimates, but also the slow vertical mixing was one cause for the mismatch in the atmospheric seasonal cycle. We hope the reviewer agrees that despite the remaining bias, the finding are important, and the bias in the inventory and process-model based estimates are to be investigated in future studies.

*Additionally, the scaling factors are resolved at spatial scales of thousands of kilometers. The error in the prior varies at scales much smaller than this – producing a severe representation error. The problem is therefore likely under-parameterized (equivalent to having covariance lengths that are too long, or regions too large), and so adequate scaling factors cannot be derived that permit unbiased residuals at individual sites. As a result, the authors find strong biases in the residuals, and even throw out some sites.*

*An example of such sites are given in the paper, and the authors remove them:*
*"Strong negative bias as found in Bukit Koto Tabang, Indonesia (BKT) (-25 to -27 ppb) and Mt. Kenya, Kenya (MKN) (-18 to -23 ppb), such that the posterior mole fractions were especially low during June-October. This suggests that the measurements at those latitudes are not representative of large regions optimized in the model."*

*To solve this problem, the authors would need to perform the inversions at high resolution using covariance length scales constrained as part of an objective error characterization.*

We agree with the reviewer that the representation errors of the observations were likely high in some regions, especially where the observation network is sparse. As pointed out by the reviewer, example regions were Asian tropical regions, where BKT is located, and south Africa where MKN is located. Although those sites were assimilated in the system, the emissions in those regions were not well constrained due to luck of observations and good prior information about the emissions and their uncertainties. We acknowledge that the bias in the posterior mole fractions remained partly due to underparametrization of the system, and resolving at higher spatial resolution with carefully chosen correlation lengths would reduce such bias. However, even with a high resolution model, we would not be able to reduce the uncertainty in the regional estimates unless further information becomes available. Although we did not develop and examine the emissions further with a higher resolution optimization scheme at this point, we hope the reviewer allows us to undertake such development in

a future study as well.

Based on the reviewer's comment, further discussion on the prior assumption and representativity of the observations are added in the revised manuscript.

**Text is revised: see e.g. Pg. 19, line 1-12.**

*Presentation Quality:*

*1. Messaging*

*This work is presented as model evaluation and interpretation. The paper goes into great detail about the variations in every region, giving their trends, comparisons to other regions, and comparisons to other papers. The work needs to be boiled down to a set of key messages. My understanding is that the main messages are those described in the Overview section of this review.*
*The body of the paper needs to be focused on providing the scientific justification for the given messages.*

We appreciate the reviewer for carefully reading the manuscript and trying to understand our messages. Following the reviewer's comment, we tried to more carefully phrase our text to better present the study and its key findings. The figures were revised substantially following the reviewers' suggestions (see also below), and the texts, including abstracts and conclusions, were revised to better highlight the key findings. Please also note that a summary of the key findings were added at the end of Summary and Conclusions.

*2. Writing*

*The paper requires extensive revision by an English language editor. Problems include: - incorrectly cased letters (e.g., "south America" should be "South America"). - inconsistent tenses and active vs. passive voice (e.g., in the abstract, line 29 "We use three configurations. . .", then line 32 "The posterior estimates were evaluated. . ."). - broken sentences (e.g., page 3, line 22 "To estimate biospheric emissions, information from an underlying ecosystem distribution map is useful, which defines the location of the sources and can help distribute larger regions over which the atmospheric signals integrate."). - truisms (e.g., page 9, line 20 "The growth rate (GR) of atmospheric methane mole fractions showed that the posterior estimates are closer to the observations than the prior, as expected"). - paragraphs that are incredibly long and rambling (e.g., page 14, line 7 to page 15, line 6; page 16, line 13 to page 17, line 5; and page 18, lines 5 - 30).*

We apologize for the inconsistencies that arose as a consequence of the weak formulation that existed in the manuscript. In this revision, we tried to more carefully phrase our text, and also had the full paper language edited by a native English speaker. Moreover, we tried to make our descriptions more clear using new labelling.

*3. Figures*

*The figures in this paper have a number of issues. Points a-c absolutely must be addressed in order for the paper to be publishable.*
*a) Figs. 9, 10, 11, 14, S4, and S5 include data and/or error bars that run off of the figure.*

We agree with the reviewer that the figures became more complete by showing the error bands fully. The y-axes of the figures were revised following the suggestion.

**Figs. 9, S8 (original 10), S9 (original 14 and S5) and 8 (original S4) are revised. Fig. 11 is removed.**

*b) Many of the figures include error bars but do not specify their meaning (1 standard deviation? 95% credible interval?)*

These were meant to be 1 standard deviation of the ensembles. We have now included the information in figure captions.

**The Figure captions (Fig. 8, S8, S9) are revised.**

*c) Figs. 4, 5, 6, 7, 8, 9, 12, 13, S1, S2, S3, and S6 are not colorblind safe.*

We apologize for the confusion that would have arose due to the choice of the colours. We tried to make the lines more distinguishable by changing the colours.

**The colours in the figures are revised.**

*d) Figs. 6 and 7 are difficult to read because of closely placed points.*

We have tried to make the points clear in the Fig. 5 (original Fig. 6) by adding a zoomed map of central Europe.

For Fig. 6 (original Fig. 7), we acknowledge that the points are close to each other, and it is difficult to distinguish each points. However, for some sites, the temporal coverage of the data was not good enough to present e.g. moving averages as lines. We tried to make points clear by changing the point sizes and shapes, but could not find a better way to present than in the revised manuscript. Therefore, we decided to retain the look and chose to illustrate using points. We hope the reviewer agrees that the intent of Fig. 6 was to give an overview of the agreement, rather than focusing on each point, and the current way of presenting is satisfactory for that.

**Fig. 5 (original Fig. 6) is revised.**

*e) Fig. 4 (top panel) should include observations.*

Observations were not plotted in the figure because NOAA global averages are for the surface, unlike $XCH_4$. However, we agree with the reviewer that the figure becomes more comprehensive by adding the observations in the top panel. We followed the reviewer's suggestion and added NOAA surface global mean $CH_4$ mole fractions with a second y-axis.

**Fig. 3 (original Fig. 4) is revised.**

---

## Author Comment (AC3) · 13 Dec 2016

This paper is merged with the accompanying paper, following the referees' comments and with approval from the Topical Editor, and we titled the new manuscript as "Global methane emission estimates for 2000-2012 from CarbonTracker Europe-CH$_4$ v1.0".

---

## Author Response (AR2)

**Response to an anonymous referee**

*- Note that Environment Canada (EC) has recently been re-branded Environment and Climate Change Canada (ECCC).*

Thank you for notifying the updated information. We have now revised the text and tables with new name.

*- pg 12, line 16-22: I interpret a "smaller bias" as being closer to zero. While the negative bias may be smaller in these comparisons, (-12 is smaller than -6), it seems like odd phrasing to me. Also, you talk about the bias at Cape Grim and Lauder being smaller, but I don't believe you give the bias at Wollongong.*

This is true that the bias for Cape Grim and Lauder were not particularly small, but better only relative to that of Wollongong, which had bias of about -35 ppb. We have now revised the text by adding bias of Wollongong to make this clear.

*- I hope that figures S5and S6 can be printed in landscape so that the individual subfigures are a bit larger.*

We have now rotated the figures to landscape.

*- Throughout the manuscript I came across a few typos and grammatical errors that I hope will be corrected in copy-editing.*

We apologies again for the errors. We have gone through the manuscript once again and tried to correct the mistakes.

[revised manuscript text omitted]

**1. Sensitivity experiments of CTE-CH4 for summer 2007**

Sensitivity experiments were performed for a test period between 29 May 2007 and 30 October 2007. Summer was chosen because biospheric methane (CH4) emissions are largest then in the Northern Hemisphere (NH), and our focus was on the northern boreal region and Europe.

**1.1 Experimental set-up**

**1.1.1 EnKF parameters' sensitivity experiments**

Two EnKF parameters (ensemble size and prior covariance matrix) were assessed using CTE-CH4, with only discrete air sample observations assimilated, and prior biosphere emission estimates from the LPX-Bern. EnKF allows a full posterior probability density function of the state (scaling factor in our case) to be represented exactly by an infinite ensemble of model states. A small ensemble size is computationally cheap to apply, but it may lead to a statistical misrepresentation of the posterior distribution. Choosing the suitable number of ensembles is often a question of finding a balance between ensemble size and computational cost. For the sensitivity experiments, we used ensemble sizes of 20 (E20) and 500 (E500) members, and in addition made a specific test for degrees of freedom (d.o.f.) related to five different ensemble sizes from 20 to 500 (i.e., 20, 60, 120, 240, and 500). The Finnish Meteorological Institute (FMI) has a computer facility with 20 nodes per processor. For E20, one processor was used, and for E500, 13 processors were used. To test sensitivities of the prior distribution of the states, we carried out four E20 simulations and three E500 simulations using random initial values sampled from a normal distribution; $N(0,1)$.

A model error covariance matrix $Q$ was used to create a prior state covariance matrix at the beginning of each time step:

$$P_b^{t+1} = P_a^t + Q, \tag{1}$$

where $P_b^{t+1}$ is the prior state covariance matrix at time $t + 1$, and $P_a^t$ is the posterior state covariance matrix at time $t$. Two matrices were examined in this study: identity ($Q1$), and $Q2$, which was based on Peters *et al.* (2005):

$$Q2 = \begin{pmatrix} A_{IWP} & A^{*1} & 0 & 0 & 0 \\ A^{*1} & A_{WMS} & 0 & 0 & 0 \\ 0 & 0 & A_{ANT} & A^{*2} & 0 \\ 0 & 0 & A^{*2} & A_{RIC} & 0 \\ 0 & 0 & 0 & 0 & \sigma_{ICE} \end{pmatrix},$$

$$A_{k_{ij}} = \begin{pmatrix} \sigma_k^2 & \sigma_k^2 \cdot e^{-d_{ij}/L} \\ \sigma_k^2 \cdot e^{-d_{ij}/L} & \sigma_k^2 \end{pmatrix} \quad \text{for } k = \text{IWP, WMS, ANT, RIC,}$$

where IWP (Inundated wetland and peatland), WMS (wet mineral soils), ANT (anthropogenic), and RIC (rice) are land-ecosystem types (Fig. 2 of main paper). ). It was assumed that $\lambda_{IWP}$, $\lambda_{WMS}$, $\lambda_{ANT}$, $\lambda_{RIC}$, $\lambda_{ICE}$ are uncorrelated, with each having a variance $\sigma_k^2 = 0.8$. Scaling factors of the same LET regions at different mTC regions (off diagonal of $A_{k_{ij}}$) were assumed to be correlated with $\sigma_k \cdot e^{-d_{ij}/L}$, where $d_{ij}$ is the distance between the centre of the regions $(i, j)$, and the correlation length $L =$

900km. For mTC3 (South American tropical), 7 (Eurasian boreal), and mTC9 (Asian tropical), between $\lambda_{\text{IWP}}$ and $\lambda_{\text{WMS}}$ ($A^{*1}$), and between $\lambda_{\text{ANT}}$ and $\lambda_{\text{RIC}}$ ($A^{*2}$) were assumed correlated with $\sigma_k^2 \cdot e^{-d_{ij}/L}$ to constrain the emissions in those regions better. The observation network within and around these regions is particularly sparse (only one or no site in the regions), which makes it difficult to constrain the emissions in the model. For $\lambda_{\text{ICE}}$, variance $\sigma_{ICE}^2$ was set to be $1e^{-8}$ for both $Q1$ and $Q2$, as

5   the emissions from this region are small, and we assumed that the prior estimates were already good.

**1.1.2 Other sensitivity experiments**

In the following experiments, CTE-CH4 with an ensemble size of 500, the same set of prior state distribution sampled from the same normal distribution, $N(0,1)$ (i.e. no random error due to sampling of prior state), and $Q2$ covariance were used. For sensitivity analysis, inversions were performed to examine the effects of: 1) the prior biosphere emissions by replacing the

10   LPX-Bern emissions with the LPJ-WHyME emission estimates, 2) the observation sets by removal of continuous observations, 3) the assimilation window length by increasing it to 12 weeks instead of 5 weeks. Finally, the effect of the Tiedtke (1989) and Gregory *et al.* (2000) convection schemes in both L$^{62}$ and L$^{78}$ configurations were examined.

**1.2 Results of sensitivity experiments**

**1.2.1 EnKF parameters' sensitivity experiments**

15   The results from the sensitivity runs (E20-E500) showed that the larger the ensemble size, the more stable the results were likely to be. With an ensemble size of 500, the mean estimates for the sum of biospheric and anthropogenic emissions aggregated over the test period differed by less than 0.5 Tg CH4 between the three E500 runs (217.9 ± 28.2, 217.7 ± 28.2, 217.4 ± 27.3 Tg CH4 per test period). However, with 20 ensemble members, mean estimates for the aggregated sum of biospheric and anthropogenic emissions differed by about 10 Tg CH4 (216.7 ± 25.3, 221.0 ± 24.9, 224.4 ± 24.3, 225.1 ± 24.6

20   Tg CH4). The smaller posterior uncertainties in the E20 experiments than in the E500 experiments were caused by underestimation of uncertainties due to the small ensemble size. The weekly sums also showed that there were more random variations in the estimates from the E20 experiments compared to the E500 experiments (Fig. S1). The stability also depended on the available observations. Regions with dense observational networks, e.g., North American boreal, showed less variation in the estimates than regions where the observation network was sparse, e.g., Asian tropical. This held for both E20 and E500.

25   The d.o.f. in the posterior ensembles (square of sum of singular values divided by sum of square of singular values) was small when the ensemble size was small as we cannot represent more d.o.f. than we have in the ensemble members. It increased significantly up to an ensemble size of 120, meaning the information added to the singular value decomposition matrix was significant, but the rate did not increase much after that, and slowly reached equilibrium (Fig. S2). Although we did not test larger ensemble sizes, the results suggest that 500 is large enough to represent the probability distribution well.

30   .

As expected, computational costs were higher for E500. With 13 processors of our computational system at FMI, the computational burden was about one hour of wall clock time per week of model time for E500. For E20, the burden was only about half an hour per week of model time with one processor. Note that the computational time of E500 could be as small as E20 if the number of nodes was increased to 500, i.e. using 25 processors in the FMI system. The observation operator was the most expensive, consuming about 80% of computational time for both cases.

The experiments using $Q1$ and $Q2$ prior covariance showed that the posterior mean emissions and their uncertainty estimates did not differ very much at a global scale. The posterior emissions that used $Q1$ and $Q2$ were $91 \pm 14$ and $91 \pm 13$ Tg $CH_4$ for biosphere emissions, and $126 \pm 27$ and $127 \pm 26$ Tg $CH_4$ for anthropogenic emissions (the numbers were aggregated over the entire run of 154 days), respectively. However, the regional uncertainty estimates were clearly smaller when $Q2$ was used rather than $Q1$, especially in the Eurasian boreal and Asian tropical regions, and showed the effect of correlations between the nearby regions and within the region (Fig. S3). Although reduction of uncertainty does not necessarily mean the estimates were better, the experiment showed the advantage of using the more informative covariance matrix, in which logical choices for spatial error correlations are made.

**1.2.2 Other sensitivity experiments**

Atmospheric $CH_4$ mole fractions were compared to assimilated NOAA discrete air sample observations. Globally, agreement with the observations did not differ much between the inversions, i.e., CTE-$CH_4$ successfully optimized emissions consistent with the average global observations regardless of the setups. For European sites, variability in the posterior mole fractions was less than in the observaions. For Asia temperate region, posterior mole fractions matched the observations noticeably better when the Gregory *et al.* (2000) convection scheme was used rather than the Tiedtke (1998) scheme.

Global biospheric emission estimates of LPJ-WHyME were 8 Tg $CH_4$ lower than those of LPX-Bern, and posterior emissions were also lower by 15 Tg $CH_4$ when LPJ-WHyME was used. The LPJ-WHyME estimates for Asian temperate and tropical regions were much lower than the LPX-Bern estimates, which remained the same in the posterior. In contrast, the LPJ-WHyME estimates in Eurasian boreal and northern Europe were more than twice as large as the LPX-Bern estimates, but were reduced to a level similar to the LPX-Bern estimates by inversion. The uncertainty estimates for those regions that used LPX-Bern were about a factor of three smaller, i.e., the system favoured the LPX-Bern estimates. For northern Europe, the difference in the posterior was 0.3 Tg $CH_4$, i.e., the inversion was not significantly sensitive to the prior estimates. For the Eurasian boreal region, the differences still remained by about 2 Tg $CH_4$ in the posterior, and additional observations would be needed to better constrain the estimates. The effect was also seen in the anthropogenic emissions; the posterior anthropogenic emissions were 10 Tg $CH_4$ greater when LPJ-WHyME was used as prior biospheric emissions. This was an effect of the inversion trying to compensate for low biosphere emissions by increasing anthropogenic emissions.

Removal of continuous observations decreased mean posterior anthropogenic emissions by about 70% in temperate North America and in southwest and east Europe. The decrease was partially compensated by an increase in biospheric emissions; for the North American temperate region, posterior biospheric emissions were about 100% larger without assimilating continuous observations, and the estimates were similar to the prior. Furthermore, the decrease was also compensated by >50%

5    increase Asian tropic emission estimates. However, differences in biospheric emissions in the Asian temperate region were small. The reason could be that the discrete observations may have had little effect on the biospheric emissions, as the observations were located near anthropogenic sources. Therefore, the inversion less sensitive to biospheric emissions when continuous measurements are not assimilated. The effect of removing continuous observations was also significant in the uncertainty estimates, which were larger for anthropogenic emissions than for biospheric emissions. The posterior uncertainty

10   for global anthropogenic emissions was about two times larger in the inversion not assimilating continuous observations, and the largest differences were found in the North American temperate and Asian temperate regions, and in southwest Europe. The posterior biospheric emission uncertainty was about three times larger in North American boreal, about twice as large in Asian temperate, and about 20% larger in North American temperate, Eurasian boreal, and Asian tropical regions than the estimates using continuous observations. These results indicate that improving prior estimates is important, especially for

15   regions where observations are sparse.

When a longer assimilation window length was used, effects of observations on emission estimates extend further in time, which could be an advantage in regions where the observation network is sparse. However, the longer travel time between sources and observations also increased the transport error, and correlates transport errors across the observation network,

20   making them less informative. Despite that, the mean and uncertainty estimates were not significantly different for both anthropogenic and biospheric emissions regardless of assimilation window length; i.e.. the expected differences were not seen in regions such as the Tropics. One reason for this would be the short test period examined, as the correlation between tropical and extratropical fluxes became significant only after several months of transport time. Simulations with longer time periods may also reveal the impacts in our model, especially in the tropics, but it may have a negative influence in other regions

25   (Babenhauserheide *et al.*, 2015).

Total global posterior mean biospheric and anthropogenic emissions were similar regardless of the convection schemes, but the sum of the posterior mean emissions in the SH was about 10 Tg $CH_4$ smaller, and that in the NH was larger when the Gregory *et al.* (2000) convection scheme was used. Due to faster vertical mixing in the NH in the Gregory *et al.* (2000)

30   convection scheme, the simulated atmospheric $CH_4$ mole fractions in the troposphere were lower compared to the Tiedtke (1989) convection scheme. Therefore, CTE-$CH_4$ produced larger emission estimates in the NH when the Gregory *et al.* (2000) convection scheme was used.

The effect of convection was generally larger when using $L^{78}$ than $L^{62}$ configurations. With $L^{78}$, posterior anthropogenic emissions differed by more than 10% in 12 mTCs due to convection, whereas the posterior anthropogenic emissions differed by more than 10% in only two mTCs with $L^{62}$. For biospheric emissions, the number of regions affected was similar in both models, but the magnitude of the differences was generally larger in $L^{78}$T. The extreme cases were seen in temperate Asia and northwest Europe, where posterior mean biosphere emissions in temperate Asia were more than 70% smaller using the Gregory *et al*. (2000) scheme than using Tiedtke (1989), and posterior mean anthropogenic emissions in northwest Europe were about 45% larger when Gregory *et al*. (2000) was used. The estimates differed by about 1% and 8% in $L^{62}$ in those regions. One reason that $L^{78}$ had a larger influence on the convection schemes was the increase in the number of optimization regions. If a large prior biospheric emission remains in "anthropogenic regions" (RIC, ANT, WTR), the effect of convection in biospheric emission estimates in $L^{78}$ would be larger than in $L^{62}$, because biospheric emissions in those regions were not optimized in $L^{62}$. This was the case for the Asian temperate region; prior biospheric emissions in the anthropogenic regions were about 20 Tg $CH_4$ (nearly 75% of the regional prior biospheric emissions). Similarly for northwest Europe, prior anthropogenic emissions in biosphere regions (IWP and WMS) were about 74% of regional total prior anthropogenic emissions.

[Figure]

**Figure S1:** Weekly sum of posterior mean biospheric and anthropogenic emissions from six inversions with ensemble sizes of 500 and 20 members (three inversions for both sizes). For each line, the initial prior state vectors were sampled randomly from a normal distribution.

[Figure]

**Figure S2:** Number of degrees of freedom (d.o.f.) in the posterior ensemble as a function of number of ensemble.

[Figure]

**Figure S3:** Relative differences in average uncertainty estimates (U) between two runs, applying covariance matrices **Q1** and **Q2**, over the test period (1-U$_{Q2}$/U$_{Q1}$), for (a) anthropogenic and (b) biospheric emissions.

**2. Additional materials**

[Figure]

**Figure S4.** Land-ecosystem map used as regional definition in the optimisation. White lines illustrate mTC borders.

[Figure]

**Figure S5.** Observed and estimated daily mean X**CH₄** at TCCON sites

[Figure]

**Figure S6**. Global and open ocean GOSAT and simulated regional 10-day mean XCH₄.

[Figure]

**Figure S7.** Monthly mean total emission estimates for different latitudinal bands, averaged over 2000-2012.

[Figure]

**Figure S8.** Regional emission estimates for land mTCs.

[Figure]

**Figure S9.** Regional total emission estimates for ocean mTCs.

**Table S1.** Mean emission estimates and their uncertainties before and after 2007 (Tg CH$_4$ yr$^{-1}$). The prior uncertainties are of L$^{62}$T and L$^{62}$G. L$^{78}$T has higher prior uncertainties in all regions due to a model feature. Region names and modified TransCom (mTC) region numbers are indicated.

| Region (mTC) | Total | | Anthropogenic | | Biosphere | |
|---|---|---|---|---|---|---|
| | Before 2007 | After 2007 | Before 2007 | After 2007 | Before 2007 | After 2007 |
| Global | | | | | | |
| Prior | 532.9±86.7 | 566.0±102.6 | 313.0±80.7 | 350.5±97.5 | 172.8±31.6 | 171.8±31.8 |
| L$^{62}$T | 507.0±45.1 | 526.3±43.7 | 287.0±36.4 | 314.9±34.5 | 172.8±28.7 | 167.7±28.7 |
| L$^{78}$T | 508.2±62.0 | 526.3±60.9 | 311.4±50.2 | 326.0±49.7 | 149.7±45.1 | 156.6±44.1 |
| L$^{62}$G | 509.1±45.9 | 527.6±44.0 | 287.9±37.4 | 312.2±34.8 | 174.1±28.8 | 171.7±28.9 |
| Europe (11-14) | | | | | | |
| Prior | 56.2±14.2 | 55.0±14.5 | 45.4±13.6 | 45.0±14.1 | 9.8±3.9 | 9.0±3.5 |
| L$^{62}$T | 54.2±10.4 | 51.5±10.5 | 46.8±10.3 | 43.8±10.5 | 6.4±2.7 | 6.8±2.5 |
| L$^{78}$T | 53.3±13.3 | 53.3±13.3 | 45.1±13.4 | 45.1±13.5 | 7.2±3.6 | 7.1±3.4 |
| L$^{62}$G | 59.7±10.6 | 58.5±10.7 | 50.9±10.6 | 49.1±10.7 | 7.7±2.7 | 8.4±2.5 |
| North American boreal (1) | | | | | | |
| Prior | 16.4±8.3 | 16.1±8.4 | 0.5±0.2 | 0.5±0.2 | 15.1±8.3 | 14.9±8.4 |
| L$^{62}$T | 13.7±2.0 | 12.8±1.5 | 0.5±0.2 | 0.5±0.2 | 12.4±2.0 | 11.6±1.5 |
| L$^{78}$T | 14.3±3.5 | 13.9±2.7 | 0.6±0.5 | 0.8±0.4 | 12.9±3.5 | 12.5±2.7 |
| L$^{62}$G | 15.7±2.1 | 14.9±1.6 | 0.5±0.2 | 0.5±0.2 | 14.4±2.1 | 13.7±1.6 |
| North American temperate (2) | | | | | | |
| Prior | 42.0±20.5 | 41.9±20.5 | 33.2±20.3 | 32.9±20.3 | 7.7±3.0 | 7.8±3.0 |
| L$^{62}$T | 49.2±7.7 | 51.9±6.8 | 41.8±7.7 | 45.1±7.0 | 6.3±2.7 | 5.7±2.6 |
| L$^{78}$T | 48.4±9.2 | 48.1±6.8 | 42.2±9.4 | 43.1±7.3 | 5.1±3.7 | 3.8±3.5 |
| L$^{62}$G | 55.6±8.4 | 59.1±7.5 | 47.4±8.4 | 51.3±7.7 | 7.2±2.7 | 6.6±2.7 |
| South American tropical (3) | | | | | | |
| Prior | 52.2±24.2 | 53.6±24.4 | 10.5±4.3 | 11.4±4.6 | 35.8±23.8 | 35.9±23.9 |
| L$^{62}$T | 53.6±23.9 | 55.1±24.1 | 11.0±4.3 | 11.7±4.5 | 36.7±23.5 | 37.1±23.6 |
| L$^{78}$T | 53.1±28.9 | 54.7±29.2 | 11.1±10.3 | 12.7±11.2 | 36.0±26.9 | 35.7±27.0 |
| L$^{62}$G | 53.3±23.9 | 54.3±24.1 | 10.7±4.3 | 11.4±4.5 | 36.7±23.5 | 36.6±23.7 |
| South American temperate (4) | | | | | | |
| Prior | 40.0±14.9 | 42.8±16.0 | 23.2±13.1 | 25.5±14.4 | 14.2±7.0 | 14.5±6.9 |
| L$^{62}$T | 49.4±14.6 | 63.3±14.9 | 28.0±12.9 | 39.9±13.5 | 18.8±6.9 | 20.6±6.7 |
| L$^{78}$T | 51.9±24.6 | 66.0±24.7 | 33.6±22.5 | 46.4±23.0 | 15.7±9.8 | 16.9±9.9 |
| L$^{62}$G | 46.0±14.6 | 58.8±15.0 | 26.3±12.9 | 37.9±13.5 | 17.0±6.9 | 18.2±6.8 |
| Northern Africa (5) | | | | | | |
| Prior | 32.2±14.9 | 33.4±16.4 | 18.6±14.7 | 20.4±16.2 | 7.2±2.4 | 7.1±2.4 |
| L$^{62}$T | 38.5±13.8 | 39.5±14.0 | 24.9±13.6 | 26.6±13.8 | 7.2±2.4 | 7.0±2.4 |
| L$^{78}$T | 40.6±19.5 | 39.2±19.0 | 27.2±16.9 | 26.8±16.6 | 7.0±9.8 | 6.4±9.4 |
| L$^{62}$G | 37.2±14.0 | 37.3±14.2 | 23.6±13.7 | 24.4±14.0 | 7.2±2.4 | 7.0±2.4 |
| Southern Africa (6) | | | | | | |
| Prior | 24.8±7.2 | 26.6±8.0 | 9.4±6.8 | 10.4±7.5 | 7.8±2.3 | 8.6±2.5 |
| L$^{62}$T | 27.9±6.9 | 28.6±7.6 | 12.4±6.5 | 12.3±7.2 | 7.9±2.3 | 8.6±2.5 |
| L$^{78}$T | 28.1±12.2 | 27.4±13.4 | 12.2±8.8 | 11.3±9.8 | 8.3±8.5 | 8.5±9.0 |
| L$^{62}$G | 27.1±7.0 | 27.7±7.7 | 11.6±6.6 | 11.6±7.3 | 7.9±2.3 | 8.5±2.5 |
| Eurasian boreal (7) | | | | | | |
| Prior | 18.8±7.4 | 20.0±8.7 | 9.5±6.8 | 11.5±8.2 | 7.1±3.0 | 6.7±2.9 |
| L$^{62}$T | 19.6±5.4 | 18.9±6.2 | 10.1±4.6 | 10.6±5.6 | 7.3±3.0 | 6.5±2.8 |

| | | | | | | |
|---|---|---|---|---|---|---|
| L[78]T | 20.6±9.2 | 18.4±9.5 | 12.1±7.7 | 10.2±8.6 | 6.4±5.9 | 6.4±5.4 |
| L[62]G | 22.0±5.5 | 21.6±6.2 | 12.5±4.7 | 13.2±5.6 | 7.3±3.0 | 6.6±2.8 |
| **Asian temperate (8)** | | | | | | |
| Prior | 142.4±72.7 | 164.7±89.8 | 106.2±72.1 | 129.3±89.3 | 34.2±9.6 | 33.4±9.5 |
| L[62]T | 76.3±24.2 | 83.7±20.1 | 36.9±25.0 | 50.1±20.7 | 37.4±6.5 | 31.5±6.1 |
| L[78]T | 66.8±28.7 | 80.6±24.2 | 48.4±26.6 | 54.8±23.2 | 16.4±24.7 | 23.8±22.5 |
| L[62]G | 78.2±25.2 | 81.0±19.9 | 37.8±26.1 | 44.2±20.6 | 38.5±6.9 | 34.8±6.4 |
| **Asian tropical (9)** | | | | | | |
| Prior | 67.7±15.8 | 70.8±16.6 | 30.6±8.7 | 35.7±9.8 | 31.1±13.2 | 31.3±13.3 |
| L[62]T | 67.5±14.3 | 68.3±14.7 | 32.0±8.4 | 35.1±9.3 | 29.6±12.1 | 29.4±12.1 |
| L[78]T | 69.2±27.8 | 67.5±28.8 | 32.2±23.0 | 32.5±24.7 | 31.1±19.6 | 31.3±19.7 |
| L[62]G | 63.2±14.3 | 65.1±14.8 | 29.8±8.4 | 32.8±9.4 | 27.4±12.2 | 28.5±12.2 |
| **Australia (10)** | | | | | | |
| Prior | 7.1±4.3 | 7.2±4.6 | 5.7±4.3 | 6.1±4.6 | -0.9±0.2 | -0.9±0.2 |
| L[62]T | 10.6±4.2 | 8.4±4.4 | 9.1±4.2 | 7.3±4.4 | -0.8±0.2 | -0.9±0.2 |
| L[78]T | 16.2±5.4 | 11.5±5.6 | 14.8±5.1 | 10.4±5.4 | -0.9±1.6 | -0.9±1.5 |
| L[62]G | 9.4±4.2 | 8.1±4.5 | 7.9±4.2 | 6.9±4.5 | -0.8±0.2 | -0.9±0.2 |
| **South West Europe (11)** | | | | | | |
| Prior | 13.0±4.9 | 12.6±4.7 | 11.4±4.9 | 11.0±4.7 | 1.4±0.8 | 1.3±0.7 |
| L[62]T | 14.4±2.3 | 12.8±2.2 | 13.0±2.4 | 11.4±2.3 | 1.2±0.6 | 1.1±0.5 |
| L[78]T | 14.6±2.0 | 13.6±2.0 | 12.8±2.2 | 12.0±2.2 | 1.5±1.0 | 1.3±0.9 |
| L[62]G | 16.5±2.5 | 13.9±2.4 | 14.7±2.6 | 12.4±2.5 | 1.6±0.6 | 1.2±0.6 |
| **South East Europe (12)** | | | | | | |
| Prior | 8.8±6.1 | 8.7±6.0 | 8.1±6.1 | 8.1±6.0 | 0.4±0.1 | 0.3±0.1 |
| L[62]T | 11.6±5.1 | 10.1±4.9 | 10.9±5.1 | 9.5±4.9 | 0.4±0.1 | 0.3±0.1 |
| L[78]T | 12.6±6.5 | 10.2±6.0 | 11.9±6.5 | 9.6±6.0 | 0.4±0.5 | 0.3±0.4 |
| L[62]G | 12.3±5.2 | 10.8±5.0 | 11.6±5.2 | 10.2±5.0 | 0.4±0.1 | 0.3±0.1 |
| **North West Europe (13)** | | | | | | |
| Prior | 13.5±2.2 | 12.2±2.1 | 10.7±1.6 | 9.6±1.5 | 2.7±1.6 | 2.5±1.5 |
| L[62]T | 11.7±1.0 | 11.3±1.1 | 10.7±0.8 | 9.8±0.9 | 0.9±0.9 | 1.5±0.8 |
| L[78]T | 11.0±1.3 | 11.4±1.6 | 9.7±1.6 | 9.7±1.9 | 1.2±1.4 | 1.7±1.3 |
| L[62]G | 13.1±1.0 | 12.7±1.1 | 11.4±0.8 | 10.4±1.0 | 1.6±0.9 | 2.2±0.9 |
| **North East Europe (14)** | | | | | | |
| Prior | 20.8±10.4 | 21.5±11.0 | 15.2±9.8 | 16.3±10.6 | 5.3±3.2 | 4.9±2.9 |
| L[62]T | 16.5±8.6 | 17.4±8.9 | 12.3±8.4 | 13.1±8.8 | 3.9±2.4 | 3.9±2.2 |
| L[78]T | 15.1±12.0 | 18.0±12.3 | 10.7±12.0 | 13.8±12.3 | 4.0±3.2 | 3.8±2.9 |
| L[62]G | 17.8±8.7 | 21.2±9.0 | 13.3±8.6 | 16.1±8.9 | 4.2±2.5 | 4.7±2.2 |
| **Ocean (16-20)** | | | | | | |
| Prior | 32.9±8.6 | 33.9±9.2 | 20.1±8.6 | 21.6±9.2 | 3.7±0.0 | 3.7±0.0 |
| L[62]T | 46.3±7.7 | 44.2±8.4 | 33.5±7.7 | 31.9±8.4 | 3.7±0.0 | 3.7±0.0 |
| L[78]T | 45.5±9.2 | 45.7±9.8 | 32.1±8.6 | 31.9±9.3 | 4.4±3.5 | 5.3±3.4 |
| L[62]G | 41.6±7.7 | 41.1±8.4 | 28.9±7.7 | 28.8±8.4 | 3.7±0.0 | 3.7±0.0 |
| **Ice (15)** | | | | | | |
| Prior | 0.1±0.0 | 0.1±0.0 | 0.1±0.0 | 0.1±0.0 | -0.0±0.0 | -0.0±0.0 |
| L[62]T | 0.1±0.0 | 0.1±0.0 | 0.1±0.0 | 0.1±0.0 | -0.0±0.0 | -0.0±0.0 |
| L[78]T | 0.1±0.1 | 0.1±0.1 | 0.1±0.1 | 0.1±0.1 | -0.0±0.0 | -0.0±0.0 |
| L[62]G | 0.1±0.0 | 0.1±0.0 | 0.1±0.0 | 0.1±0.0 | -0.0±0.0 | -0.0±0.0 |

**Table S2.** Root mean squared error (RMSE) between TCCON and posterior XCH4 without averaging kernel applied (ppb).

| Site | Latitude (°N) | Longitude (°E) | Posterior $L^{62}T$ | $L^{78}T$ | $L^{62}G$ |
|---|---|---|---|---|---|
| Eureka, Canada | 80.05 | -86.42 | 8.48 | 8.21 | 10.26 |
| Sodankylä, Finland | 67.37 | 26.63 | 13.59 | 14.20 | 17.92 |
| Bialystok, Poland | 53.23 | 23.03 | 10.12 | 10.94 | 14.77 |
| Karlsruhe, Germany | 49.10 | 8.44 | 11.17 | 12.32 | 10.89 |
| Garmisch, Germany | 47.48 | 11.06 | 9.62 | 10.61 | 14.13 |
| Park Falls, WI, USA | 45.95 | -90.27 | 11.07 | 11.52 | 14.96 |
| Indianapolis, IN, USA | 39.86 | -86.00 | 8.00 | 8.67 | 11.89 |
| Lamont, OK, USA | 36.60 | -97.49 | 14.37 | 16.69 | 11.11 |
| Pasadena, CA, USA (Caltech*1) | 34.14 | -118.13 | 16.78 | 20.14 | 12.33 |
| Pasadena, CA, USA (JPL*2) | 34.12 | -118.18 | 26.65 | 28.16 | 18.04 |
| Pasadena, CA, USA (JPL*3) | 34.12 | -118.18 | 23.77 | 24.86 | 16.17 |
| Saga, Japan | 33.24 | 130.29 | 18.25 | 18.94 | 13.33 |
| Izana, Tenerife, Spain | 28.30 | -16.50 | 10.84 | 10.87 | 16.62 |
| Ascension Island | -7.92 | -14.33 | 23.03 | 22.44 | 18.21 |
| Darwin, Australia | -12.42 | 130.89 | 23.49 | 21.89 | 20.95 |
| Reunion Island, France | -20.90 | 55.49 | 21.05 | 19.34 | 18.73 |
| Wollongong, Australia | -34.41 | 150.88 | 26.84 | 24.36 | 24.46 |
| Lauder, New Zealand (120HR) | -45.04 | 169.68 | 15.11 | 13.04 | 12.21 |
| Lauder, New Zealand (125HR) | -45.04 | 169.68 | 15.48 | 13.30 | 13.03 |

*1 = California Institute of Technology, 2012

*2 = Jet Propulsion Laboratory, 2007-2008

*3 = Jet Propulsion Laboratory, 2011-2012

**Table S3.** Root mean squared error (RMSE) between GOSAT and model XCH$_4$ without averaging kernel applied (ppb).

| Region (mTC) \Inversion | Posterior | | |
|---|---|---|---|
| | L$^{62}$T | L$^{78}$T | L$^{62}$G |
| Global (1-20) | 12.5 | 12.5 | 7.2 |
| EU (11-14) | 11.5 | 12.0 | 15.9 |
| North American boreal (1) | 11.2 | 11.7 | 15.1 |
| North American temperate (2) | 10.4 | 11.7 | 11.0 |
| South American tropical (3) | 26.9 | 26.6 | 23.5 |
| South American temperate (4) | 19.5 | 17.9 | 18.2 |
| Northern Africa (5) | 9.4 | 11.2 | 7.8 |
| Southern Africa (6) | 21.7 | 20.8 | 19.6 |
| Eurasian boreal (7) | 11.8 | 12.6 | 16.8 |
| Asian temperate (8) | 12.3 | 13.7 | 9.4 |
| Asian tropical (9) | 24.8 | 25.6 | 19.0 |
| Australia (10) | 18.8 | 17.0 | 16.6 |
| South West Europe (11) | 12.7 | 13.1 | 15.3 |
| South East Europe (12) | 13.7 | 14.5 | 18.0 |
| North West Europe (13) | 15.4 | 16.4 | 19.6 |
| North East Europe (14) | 12.7 | 13.5 | 17.5 |
| Ocean (16-20) | 17.0 | 16.2 | 12.3 |